# Reciprocal inhibition between TP63 and STAT1 regulates anti-tumor immune response through interferon-γ signaling in squamous cancer

Yuan Jiang[1,2,15], Yueyuan Zheng [3,15], Yuan-Wei Zhang[1,15], Shuai Kong[1,2,15], Jinxiu Dong[1], Fei Wang[1,2], Benjamin Ziman[4], Sigal Gery[5], Jia-Jie Hao[6], Dan Zhou[1,7], Jianian Zhou[1,2], Allen S. Ho [8], Uttam K. Sinha[9], Jian Chen[1], Shuo Zhang[1,2], Chuntong Yin[1,10], Dan-Dan Wei[1,2,11], Masaharu Hazawa[12], Huaguang Pan[10], Zhihao Lu[13], Wen-Qiang Wei [14], Ming-Rong Wang[6], H. Phillip Koeffler [5], De-Chen Lin [4] ✉ & Yan-Yi Jiang [1,2] ✉

Squamous cell carcinomas (SCCs) are common and aggressive malignancies. Immune check point blockade (ICB) therapy using PD-1/PD-L1 antibodies has been approved in several types of advanced SCCs. However, low response rate and treatment resistance are common. Improving the efficacy of ICB therapy requires better understanding of the mechanism of immune evasion. Here, we identify that the SCC-master transcription factor TP63 suppresses interferon-γ (IFNγ) signaling. TP63 inhibition leads to increased CD8⁺ T cell infiltration and heighten tumor killing in in vivo syngeneic mouse model and ex vivo co-culture system, respectively. Moreover, expression of TP63 is negatively correlated with CD8⁺ T cell infiltration and activation in patients with SCC. Silencing of TP63 enhances the anti-tumor efficacy of PD-1 blockade by promoting CD8⁺ T cell infiltration and functionality. Mechanistically, TP63 and STAT1 mutually suppress each other to regulate the IFNγ signaling by co-occupying and co-regulating their own promoters and enhancers. Together, our findings elucidate a tumor-extrinsic function of TP63 in promoting immune evasion of SCC cells. Over-expression of TP63 may serve as a biomarker predicting the outcome of SCC patients treated with ICB therapy, and targeting TP63/STAT/IFNγ axis may enhance the efficacy of ICB therapy for this deadly cancer.

Squamous cell carcinomas (SCCs) are collection of aggressive malignancies emerging from various epithelial tissues, such as esophagus, lung, and head and neck. SCCs cause more than 1 million cancer-related deaths worldwide each year[1,2] and there is a lack of effective targeted therapies for SCC patients.

In recent years, immunotherapy has shifted the paradigm of the clinical management of cancer patients. In particular, immune check-point blockade (ICB) therapy targeting the PD-1/PD-L1 pathway has achieved clinical success in many cancer types, including SCCs[3–12]. However, as with other cancer types, only a minority of SCC patients

(20–40%) exhibit a durable response. In particular, patients who display an "immune-cold" immunophenotype (limited intra-tumoral infiltration of immune cells) are often nonresponsive or resistant to anti-PD-1/PD-L1 agents[13–15]. Although several studies have analyzed the immune profiles and heterogeneity of SCC tumor microenvironment (TME) using mouse models and human patient samples[7,16–21], the underlying mechanism of immune evasion remains to be explored. Understanding the theoretical basis of immune evasion may identify predictive biomarkers and potential drug targets for the development of more effective immunotherapeutic strategies.

TP63 is a recognized master regulator transcription factor (TF) in SCCs. We and others have reported copy number alterations and overexpression of *TP63* in SCCs[22–26]. In addition, our previous work has established the fundamental role of TP63 in promoting SCC development by regulating hundreds of super-enhancers in an SCC-specific manner[27–29]. Nevertheless, most of prior work on TP63 has been focused on its tumor-intrinsic functions. Whether and how this TF regulates cancer biology in a non-tumor cell autonomous manner is unclear.

Here, our unbiased analyses of both SCC patient samples and cell lines identify the interferon-γ (IFNγ) signaling as the most significantly enriched pathway suppresses by TP63 uniquely in SCCs. Elevated infiltration of CD8$^+$ T cell and enhanced tumor killing effect are observed in both SCC patients and syngeneic murine SCC tumors with low expression of *TP63*. Given the prominent and indispensable role of IFNγ signaling for anti-tumor immunity[30,31], we hypothesize that over-expression of TP63 promotes resistance to ICB therapy. Indeed, TP63 suppression enhances the efficacy of immunotherapy against SCCs. Finally, we report a reciprocal inhibition between TP63 and STAT1 at the transcription level, which determines anti-tumor immune response of SCC cells by regulating the IFNγ signaling. This study reveals a central regulator of anti-tumor immunity in SCC tumors, provides a potential strategy to turn "immune-cold" SCC tumors into "hot" ones, and suggests TP63 as a candidate biomarker of immune-cold tumor for predicting immunotherapy outcome in SCC patients.

## Results

### TP63 suppresses the IFNγ signaling pathway in SCC tumors

Our previous studies have established TP63 as an SCC-specific master regulator TF[27,28,32], and the specific expression profile of *TP63* in SCCs was validated at the pan-cancer level using RNA-seq data of human tumor samples from The Cancer Genome Atlas (TCGA) (Supplementary Fig. 1A). To explore biological processes associated with *TP63* expression in SCC patient samples, we interrogated RNA-seq data of 1077 patients and 112 cell lines (data were from TCGA and The Cancer Cell Line Encyclopedia (CCLE), respectively) representing three common types of squamous cancers, including lung squamous cell carcinoma (LUSC), head and neck squamous cell carcinoma (HNSC) and esophageal squamous cell carcinoma (ESCC). These SCCs have unified pathological and molecular features specific to squamous cell lineage. Consistent with the oncogenic role of TP63 in SCC, genes positively correlated with *TP63* expression were enriched in cancer-promoting hallmark pathways, including E2F targets, G2M checkpoint, MYC targets and mTOR signaling pathways (Supplementary Fig. 1B). Strikingly and unexpectedly, almost all enriched hallmarks negatively correlated with *TP63* expression were related to immune response functions, particularly Interferon-γ (IFNγ) and Interferon-α (IFNα) responses, which were top ranked (Fig. 1A). The anti-correlation was also validated by our in-house and public RNA-seq data in the presence or absence of *TP63* knockdown in multiple different SCC cell lines (Fig. 1B). This concordant result suggests that TP63 not only is negatively associated with IFNγ and IFNα signaling, but also functionally suppresses these pathways. Moreover, such negative correlation was only found in SCCs but none other cancer types, suggesting its specificity (Supplementary Fig. 1C).

We next closely inspected enriched genes in both IFNγ and IFNα signatures that were negatively correlated with *TP63* expression, noting that almost half of them (46.9%, 23/49) were shared across three types of SCCs. These 23 IFN-stimulated genes (ISGs) included antigen processing and presentation factors (*B2M, CD74, PSMB9, LAP3*), immune response-related transcription regulators (*IRF1, IRF7, SP110, BATF2*), chemokines and cytokines (*CXCL10, CXCL11, IL15*) (Fig. 1C and Supplementary Fig. 1D, E). To mimic IFN conditions in the tumor microenvironment (TME), we added IFNγ exogenously in SCC cells since (i) IFNγ pathway is the most significant pathway negatively regulated by TP63, (ii) IFNγ receptors (IFNGR1, IFNGR2) have higher expression than IFNα receptors (IFNAR1, IFNAR2) in SCC cells. As anticipated, most of ISG genes were significantly elevated following *TP63* knockdown in the presence of IFNγ (Fig. 1D, 1E and Supplementary Fig. 1F). As a validation, protein upregulation of representative ISGs was also detected, such as B2M, BATF2 and IRF1 (Fig. 1F). These results suggest that TP63 suppresses the IFNγ signaling and the expression of ISGs in an SCC specific manner.

### TP63 inhibits CD8$^+$ T cell infiltration and activation in murine SCC models

Given the prominent and essential role of IFNγ signaling in anti-tumor immunity, we hypothesized that over-expression of TP63 facilitates immune evasion of SCC cells by suppressing the IFNγ signaling pathway. To test this hypothesis in vivo, we employed syngeneic murine SCC models and established allografts in immuno-competent C57BL/6 J mice. To analyze the abundance of immune cells in the TME, excised tumors were dissociated into single cells and CD45$^+$ cells were enriched using anti-CD45 antibody-coated microbeads, followed by the analyses using either single-cell RNA sequencing (scRNA-seq) or flow cytometry (Fig. 2A, B).

In the scRNA-seq of MOC22 samples, 854 and 1057 transcriptomes of single cells were respectively obtained from Scramble and sh*Trp63* tumors (Supplementary Fig. 2A). Using canonical markers, we identified major immune cell populations, including T cells, natural killer (NK) cells, B cells, dendritic cells (DCs), macrophages and neutrophils (Fig. 2C, D). Knockdown of *Trp63* altered the relative composition of immune subsets in murine SCC TME, with CD8$^+$ T cells showing the most prominent increase compared with the Scramble group (44.2% vs. 20.8%) (Fig. 2D, E and Supplementary Fig. 2B, C). *Trp63* silencing additionally expanded the populations of Th17, Macrophage 1, NK and DC, while decreasing those of CD4$^+$ T, NKT (NK-T cell), Macrophage 2, Macrophage 3, and Neutrophil. It was particularly notable that CD8$^+$ T cells exhibited the most striking change since they are not only the soldiers executing tumor-killing function but also the central target of ICB therapies. Thus, we next focused on analyzing further CD8$^+$ T cells (Supplementary Fig. 2D, E). Tumor-associated macrophages and neutrophils can promote tumor cell survival and immune evasion[33,34]. It is interesting to note that macrophage 2/3 and neutrophil populations were also decreased upon *TP63* knockdown. Further investigations on potential regulation on macrophages and neutrophils by TP63 are thus necessary to further understand the complexity of tumor microenvironment and immune response of SCCs.

Unsupervised clustering identified 4 subsets of CD8$^+$ T cells, designated as exhausted (CD8_Tex), effector (CD8_Tem), naïve (CD8_Tn), and CD8_MHCII (Fig. 2F and Supplementary Figs. 2F, G and 3A) based on the mean expression of published gene signatures[35–37]. Both effector and exhausted CD8$^+$ T cells displayed high expression of activation features (*Pdcd1, Ctla4, Lag3, Havcr2/Tim-3, Ifng, Tnfrsf9, Cd69*), cytotoxic markers (*Gzmb, Gzma and Prf1*) and IFN signature genes (*Isg15, Isg20 and Ifit1*). Between these two clusters, the effector population (CD8_Tem) showed higher activation features of *Cd69, Ifng, Jun* and chemokines *Ccl3* and *Ccl4*, suggesting an early activation state of T cells; the exhausted cluster (CD8_Tex) was characterized by the highest expression of exhausted and cytotoxic markers including

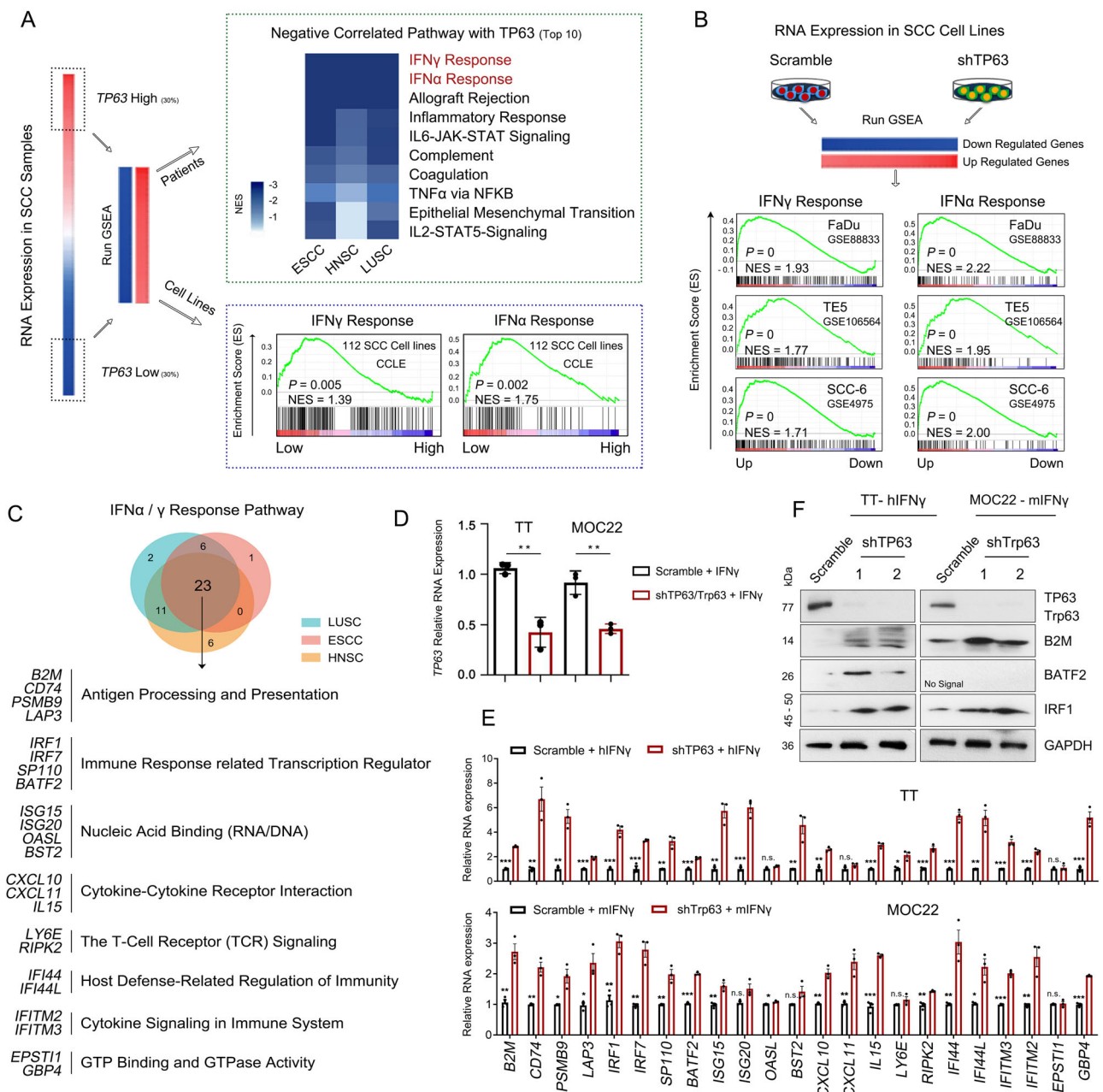

**Fig. 1 | TP63 suppresses IFNγ response signaling in SCC tumors. A** Scheme of RNA-seq analysis in SCC primary tumors and cell lines. Right upper: Hallmark pathway enrichment analysis showing the top 10 pathways that are negatively correlated with the expression of *TP63*. Right lower: gene set enrichment analysis (GSEA) plots revealing significant enrichment of IFNγ and IFNα response pathways in *TP63*-low expressed SCC cells. Data were from TCGA (*n* = 1077) and CCLE (*n* = 112), respectively. ESCC: esophageal squamous cell carcinoma; HNSC: head and neck squamous cell carcinoma; LUSC: lung squamous cell carcinoma. **B** GSEA revealing significant enrichment of upregulated genes in IFNγ and IFNα response pathways upon knockdown of *TP63* expression in 3 SCC cell lines. RNA-seq data were from in-house (GSE106564) and public datasets (GSE88833 and GSE4975).

**C** Venn diagram representing TP63 negatively regulated IFNα/γ response genes in three types of SCCs. 23 overlapped ISGs are listed below. **D, E** qRT-PCR analysis showing relative mRNA levels of *TP63* (**D**) and the 23 ISGs (**E**) in human (TT) and murine SCC (MOC22) cells expressing non-targeting control (Scramble) or *TP63*-targeting shRNA (sh*TP63*) pulsed with IFNγ (100 ng/mL) for 48 h. Data represent mean ± SD, *n* = 3 biologically independent experiments. **F** Western blotting analysis showing the protein levels of TP63 and representative ISGs of **E** in TT and MOC22 cells expressing non-targeting control (Scramble) or *TP63*-targeting shRNA (sh*TP63*) pulsed with IFNγ (100 ng/mL) for 48 h. The results were repeated with three biologically independent experiments in two cell lines. Source data and exact *P* values for Fig. 1D−F are provided as a Source Data file.

*Pdcd1* and *Ctla4* (Fig. 2G and Supplementary Figs. 2G and 3A). The naïve CD8⁺ T cells (CD8_Tn) expressed high levels of *Tcf7, Lef1, S1pr1* and *Pdlim1* while lacked cytotoxic and activation features, and the last subset was defined as CD8_MHCII[35,38] because of the prominent expression of MHCII signatures, including *H2-Ab1, H2-Aa, H2-Eb1*, and *Cd74* (Supplementary Figs. 2G and 3A). Notably, silencing of *Trp63* elevated the proportion of effector CD8⁺ T cell subset while reduced the frequency of naïve CD8⁺ T cells (Fig. 2H), suggesting that both the

number and activity of CD8⁺ T cells were enhanced by the perturbation of TP63.

Consistently, we detected significantly increased infiltration and activation of CD8⁺ T cells in the sh*Trp63* group compared with the Scramble group in HNM007 tumor samples, as measured by the levels of activation and cytotoxic protein markers (CD69, GZMB, IFNγ) using flow cytometry analysis. In AKR tumor samples, we also observed increased proportion of activated CD8⁺ T population (CD69 and IFNγ)

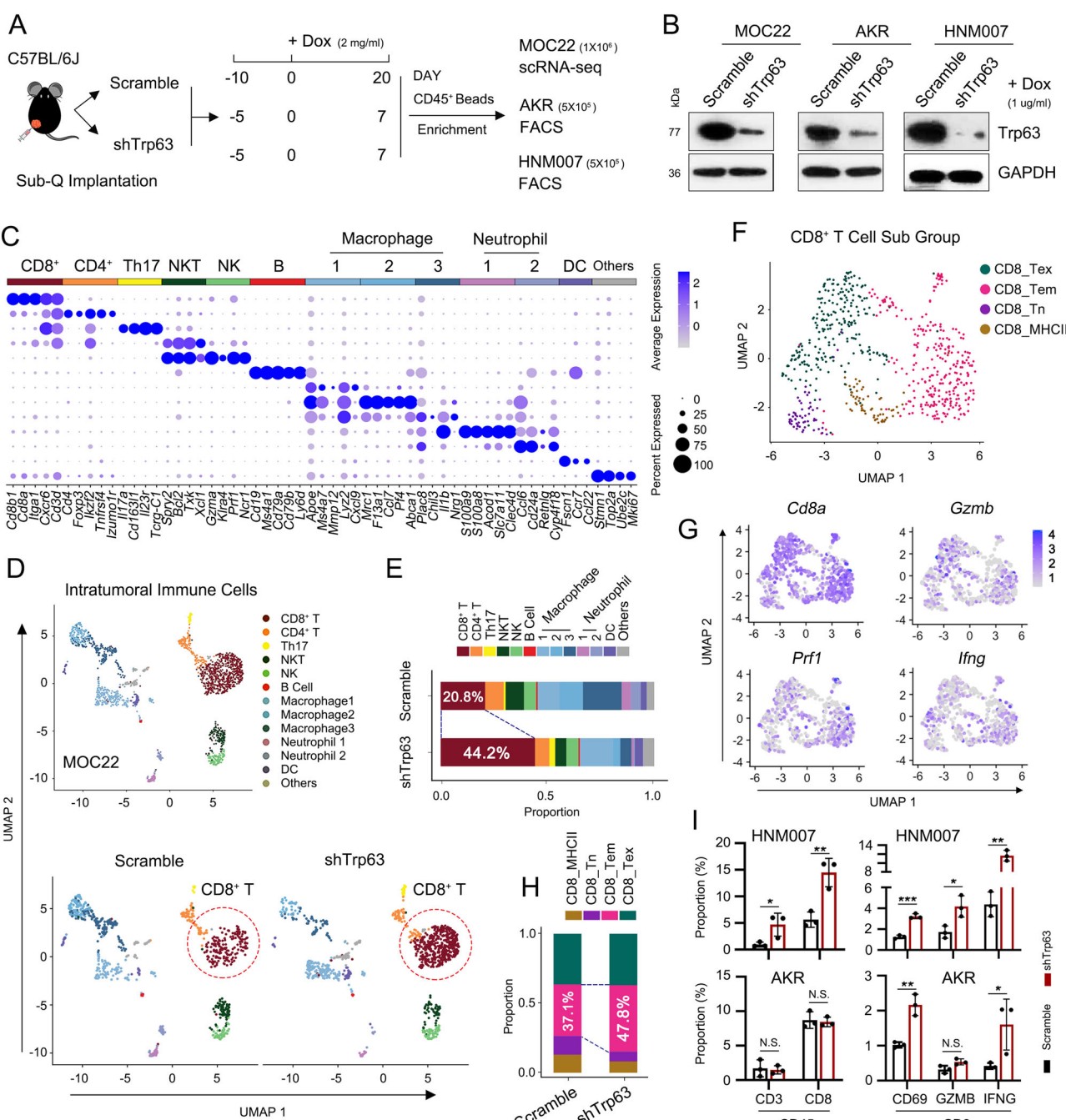

**Fig. 2 | TP63 suppresses CD8⁺ T cells infiltration and activation in immune-competent murine SCC models. A** Schematic graph of the in vivo syngeneic experiments. **B** TP63 expression assessed by western blotting in murine SCC cell lines transduced with Dox-inducible *Trp63* shRNA. The similar results were repeated in three biologically independent cells. **C** Dotplot showing the expression levels of representative marker genes across each immune cell type. For scRNA-seq experiment, each group included 4 tumors from 4 mice, which were combined and dissociated into single cells capture. A total of 1911 CD45⁺ immune cells were analyzed. **D** UMAP plots of the clustering of 1911 cells from 854 Scramble and 1,057 *Trp63* knockdown cells, showing all the intratumoral immune cells (upper) or the immune cells in either Scramble or sh*Trp63* MOC22 tumors (bottom). **E** The proportion of each immune cell type in the Scramble and *Trp63* knockdown tumors. **F** UMAP plots showing the subgroups of 645 CD8⁺ T cells from Scramble and

sh*Trp63* MOC22 tumors. **G** The expression levels of canonical markers for each cell cluster. **H** The proportion of CD8⁺ T cell subgroups in Scramble and sh*Trp63* MOC22 tumors. **I** The proportion of CD3⁺ and CD8⁺ in CD45⁺ immune cells and CD69⁺, GZMB⁺ and IFNγ⁺ in CD8⁺ T cells from sh*Trp63* or Scramble HNM007 and AKR allografts revealed by FACS analysis. Bars represent mean ± SD of three biologically independent experiments. For the comparison of each population between the Scramble and sh*TP63* group in HNM007 and AKR allografts: CD45⁺ CD3⁺ ($P = 0.0434/0.0792$), CD45⁺ CD8⁺ ($P = 0.0072/0.7600$), CD8⁺ CD69⁺ ($P = 0.0003/0.0033$), CD8⁺ GZMB⁺ ($P = 0.0229/0.0500$), CD8⁺ IFNG⁺ ($P = 0.0017/0.0495$). $P$ values were determined using a two-sided *t*-test. *$P < 0.05$, **$P < 0.01$, ***$P < 0.001$. The gating strategy are provided in Supplementary Fig. 3B. Source data and the exact cell number of each subgroup in Fig. 1C and E–H are provided as a Source Data file.

in *Trp63*-knockdown tumors (Fig. 2I and Supplementary Fig. 3B). Collectively, these results demonstrate that TP63 inhibits CD8[+] T cell infiltration and activation in murine SCC models.

### High expression of *TP63* negatively correlates with CD8[+] T cell infiltration and activation in the TME of human SCC

To extend the above findings to human cancers, we re-analyzed public scRNA-seq data (GSE160269) from 60 ESCC tumors and 4 adjacent normal tissues[20]. As expected, *TP63* was highly expressed in squamous epithelial cells, but was barely detectable in CD45[+] immune cells (Fig. 3A and Supplementary Fig. 4A). Importantly, among major immune subsets, the expression of *TP63* was specifically anti-correlated with the T cell fraction (Fig. 3B), which was not seen in either myeloid or B cell compartment (Supplementary Fig. 4B). We then further focused on intra-tumoral T cell populations, including effector, exhausted memory, naïve, regulatory T cells, etc. classified using established marker genes in Zhang et al.[20].

To determine the relationship between tumor-intrinsic *TP63* expression and T cell infiltration, we first stratified these 60 patient samples as either *TP63*-high or -low groups (top/bottom 15%) based on the expression level of *TP63* in 97,631 tumor cells (Supplementary Fig. 4C). A higher proportion of CD8[+] T cells was observed in *TP63*-low tumor samples (median = 52.29%) than *TP63*-high tumors (median = 40.07%) (Fig. 3C, D and Supplementary Fig. 4D). The negative correlation between expression of *TP63* and the abundance of CD8[+] T cells was further verified using both ESCC scRNA-seq and TCGA bulk RNA-seq data from LUSC and HNSC patient samples (Fig. 3E). It was also notable that almost no correlation existed in ESCC and HNSC tumor samples between *TP63* expression and the abundance of any other immune populations, including B cells, CD4[+] T cells, macrophages, neutrophils and DCs, except for memory CD4[+] T cells in ESCC scRNA-seq data. The negative correlation between those cells and *TP63* expression were observed in LUSC samples probably because of one exceptional case (Supplementary Fig. 4E–G).

By dissecting the altered subsets of CD8[+] T cells, we observed augmented infiltration of effector, exhausted, and memory CD8[+] T populations in *TP63*-low tumors (Supplementary Fig. 5A). The abundance of those CD8[+] T subsets also showed negative correlation with the expression of *TP63* (Supplementary Fig. 5B). In addition, inferred activity of those CD8[+] T cells was higher in *TP63*-low tumors, as characterized by the expression of cytotoxic markers, such as *GZMB, PRF1, GZMK and IFNγ* (Fig. 4A–C and Supplementary Fig. 5C, D). To further validate these findings, we examined the protein levels of TP63 and the enrichment of CD8[+] T cells in SCC TME by immunofluorescence (IF) staining using tissue microarrays from an independent cohort of 10 ESCC patient samples. A prominent accumulation of CD8[+] T cells was seen in SCC tumors with low/no expression of TP63. In contrast, tumors with high TP63 expression had only a few scattered CD8[+] T cells at the edge of cancer lesions (Fig. 4D). Statistical analysis confirmed the significant increase of the infiltration of CD8[+] T cells in patient tumors with low expression of TP63 (Fig. 4E). Taken together, these results revealed that over-expression of TP63 impaired CD8[+] T cell infiltration and activation, which may facilitate SCC cells to escape immunosurveillance and immune attack.

### Depletion of TP63 promotes SCC cell killing by CD8[+] T cell ex vivo

To further explore whether TP63 functionally impairs CD8[+] T cell functionality and T cell-mediated antigen-dependent cell killing, we established an ex vivo co-culture system by incubating SCC cancer cells with CD8[+] T cells derived from OT-I TCR mice[39] (Fig. 5A). In comparison with the Scramble group, *TP63*-knockdown strongly augmented tumor cell killing by CD8[+] T cells in both MOC22 and AKR cell lines, at different effector: target ratios (Fig. 5B, C and Supplementary Fig. 6A, B). As shown in Fig. 5D and Supplementary Fig. 6C (red arrows

and dotted circles), targeted SCC cells were surrounded by activated CD8[+] T cells during the attack. Consistently, we detected markedly increased proportions of GZMB[+] and IFNγ[+] CD8[+] T cells in co-culture with the *Trp63*-knockdown group (Fig. 5E, F and Supplementary Fig. 6D). More importantly, ectopic expression of TP63 rescued T-cell killing caused by *TP63*-knockdown (Supplementary Fig. 6E–G). These results confirm that silencing of *TP63* promotes the antigen-dependent cytotoxicity of CD8[+] T cells against SCC cells in vitro.

### TP63 inhibition enhances the efficacy of PD-1 mAb therapy in murine SCC models

Given the prominent function of TP63 in suppressing IFNγ pathway and impairing CD8[+] cell infiltration and activation in the SCC TME, we next asked whether depletion of *TP63* could enhance the efficacy of PD-1 mAb therapy, which reinvigorates CD8[+] T cells. Before addressing this question, we first analyzed the expression of *PD-L1* (*CD274*) in paired SCC tumors, finding significantly higher expression of *CD274* (*PD-L1*) in SCC tumors than matched normal tissues from 3 different cohorts (GSE53624, GSE53622, and TCGA datasets) (Supplementary Fig. 7A). We then inoculated either Scramble or sh*Trp63* SCC cells into syngeneic immuno-competent C57BL/6 J mice. These tumor-bearing mice were treated with either PD-1 mAb or IgG isotype (IgG2a) when the tumors grew to an appropriate size. No significant loss of body weight or other common toxic effects were observed in these mice during the treatment (Supplementary Fig. 7B).

In line with the tumor-intrinsic role of TP63, loss of *TP63* (sh*Trp63* + IgG2a) decreased the tumor volume compared with the control (Scrmable + IgG2a) group (Fig. 5G, H). PD-1 mAb treatment alone strongly reduced AKR tumor volume, but had no effect on HNM007 tumor growth (Fig. 5G, H). The heterogeneous responses of murine SCCs to ICB therapy mirror those seen in SCC patients[7,14]. Importantly, depletion of *Trp63* (sh*Trp63*) plus PD-1 mAb treatment had the most potent anti-tumor effect in both AKR and HNM007 models (Fig. 5G, H). Congruently, IF staining of tumor slices demonstrated that sh*Trp63* plus PD-1 mAb treatment maximized the infiltration of CD8[+] T cells in both AKR and HNM007 TME (Fig. 5I and Supplementary Fig. 7C). To ascertain the functional contribution of CD8[+] T cells, we utilized a CD8 blocking mAb to deplete CD8[+] T cells and repeated the assay. Notably, the anti-tumor effect exerted by sh*Trp63* and PD-1 mAb treatment was markedly diminished (Fig. 5J, K). We then performed additional mouse model experiments using sh*Trp63* SCC cells and treated these animals with either IgG isotype antibody, IFNγ blocking antibody or CD8 blocking mAb. Under the IFNγ blocking condition, tumors continued to grow and the tumor weight was significantly higher compared with that of the control group treated with IgG isotype antibody (Supplementary Fig. 7D), suggesting that the reduced tumor growth caused by knockdown of TP63 was mediated at least partially by the IFNγ signaling. In addition, depletion of CD8[+] T cell with CD8 mAb also resulted in continued tumor growth and elevated tumor weight (Supplementary Fig. 7E), strongly suggesting that inhibiting TP63 synergistically enhances the anti-tumor effect of PD-1 mAb by enhancing CD8[+] T cell infiltration and activity.

### TP63 suppresses IFNγ pathway by inhibiting STAT1 transcription

To elucidate the underlying mechanism of TP63 in the suppression of IFNγ signaling, we first interrogated TP63 occupancy at those 23 ISGs regulated by TP63 using ChIP-seq data from us and others[26–28]. While TP63 strongly bound its canonical targets such as *SOX2*, no binding peaks were observed at any of the 23 ISGs (Fig. 6A and Supplementary Fig. 8), indicating that TP63 indirectly regulates the transcription of these ISGs.

We next determined TP63-interacting proteins in SCCs by re-analyzing two public immunoprecipitation-mass spectrometry (IP-

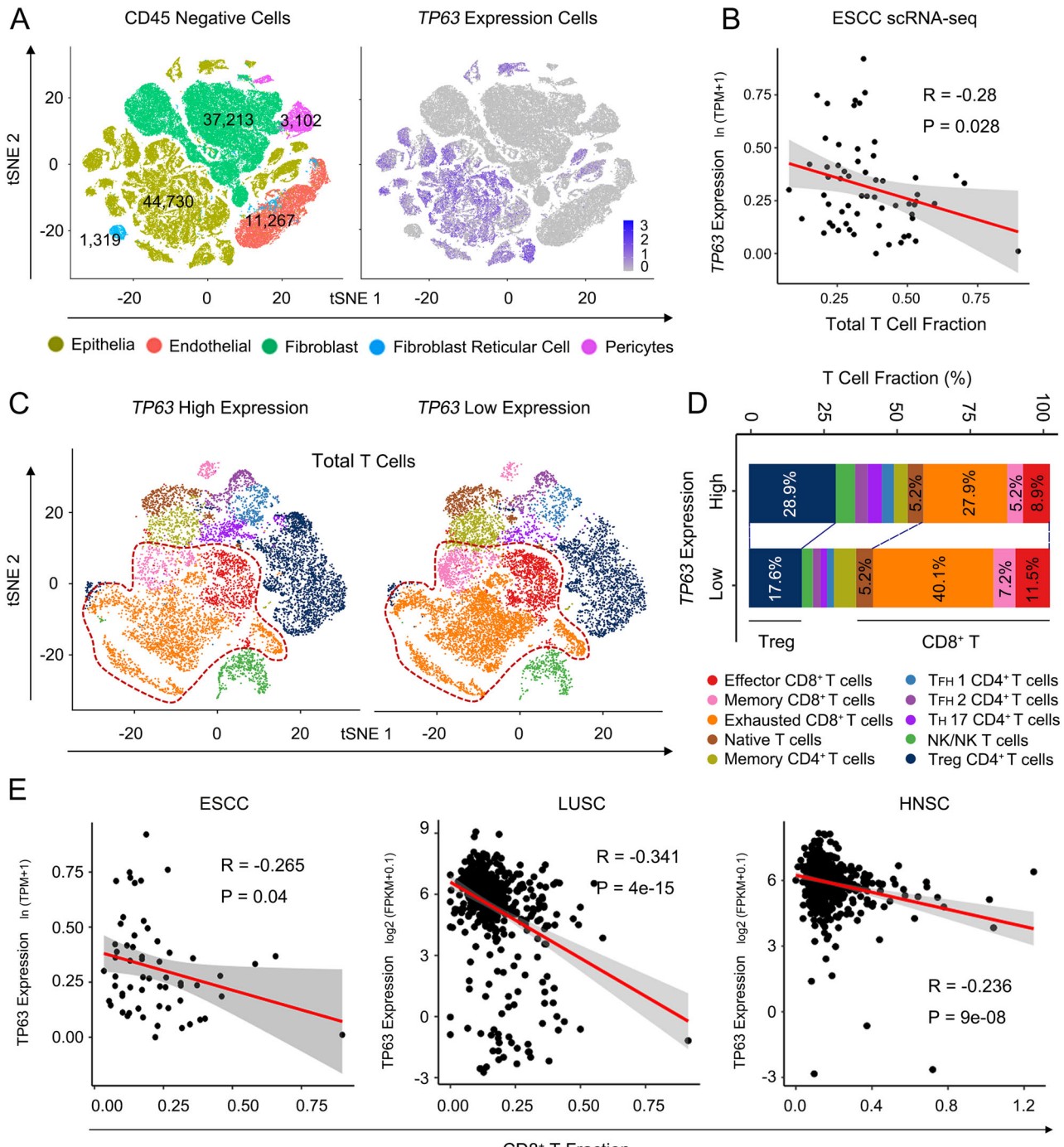

**Fig. 3 | *TP63* expression level is negatively correlated with CD8+ T cell infiltration in ESCC patient tumors. A** t-SNE plots of 97,631 CD45− cells in 60 human ESCC tumor and 4 adjacent normal tissue samples, colored by cell types (left) or *TP63* expression level (right). scRNA-seq dataset (GSE160269) and cell annotations were obtained from Zhang et al. [20]. **B** Scatter plot showing a negative correlation between total T cell fraction and *TP63* expression. The average *TP63* expression in CD45− cells in each tumor was calculated, n = 60 independent ESCC patients. **C** t-SNE plots of the clustering of T cells in 9 *TP63*-high vs. 9 *TP63*-low tumor samples. **D** The fraction of T cell subgroups in *TP63*-high vs. -low ESCC tumors. **E** Negative correlation between CD8+ T cell fraction and *TP63* expression in 76 ESCC (left), 501 LUSC (middle), and 500 HNSC (right) patient samples. The fraction in LUSC and HNSC samples were predicted by TIMER2 using TCGA expression datasets. Pearson Correlation Coefficient was calculated in **B** and **E**. The error bands show 95% confidence interval. R: Pearson's product-moment correlation; *P* value: two-sided *t*-test.

MS) datasets from us and others[26,28]. Among a total of 114 candidate TP63-interacting proteins, 12 were shared in both datasets, which notably included STAT1, a key mediator of the IFNγ signaling (Fig. 6B). Following co-immunoprecipitation (co-IP) assays confirmed the physical interaction between endogenous TP63 and p-STAT1 (particularly at Ser727 and weakly at Tyr701) proteins in both of human and murine SCC cells using either TP63 or p-STAT1

antibodies for reciprocal immunoprecipitation (Fig. 6C and Supplementary Fig. 9A). Furthermore, IF and western blot assays validated the co-localization of TP63 and p-STAT1 in the nucleus upon the stimulation of IFNγ, while unphosphorylated STAT1 primarily located in the cytoplasm (Fig. 6D and Supplementary Fig. 9B). These results were in agreement with previous reports that IFNγ stimulation results in phosphorylation of STAT1 (p-STAT1), which

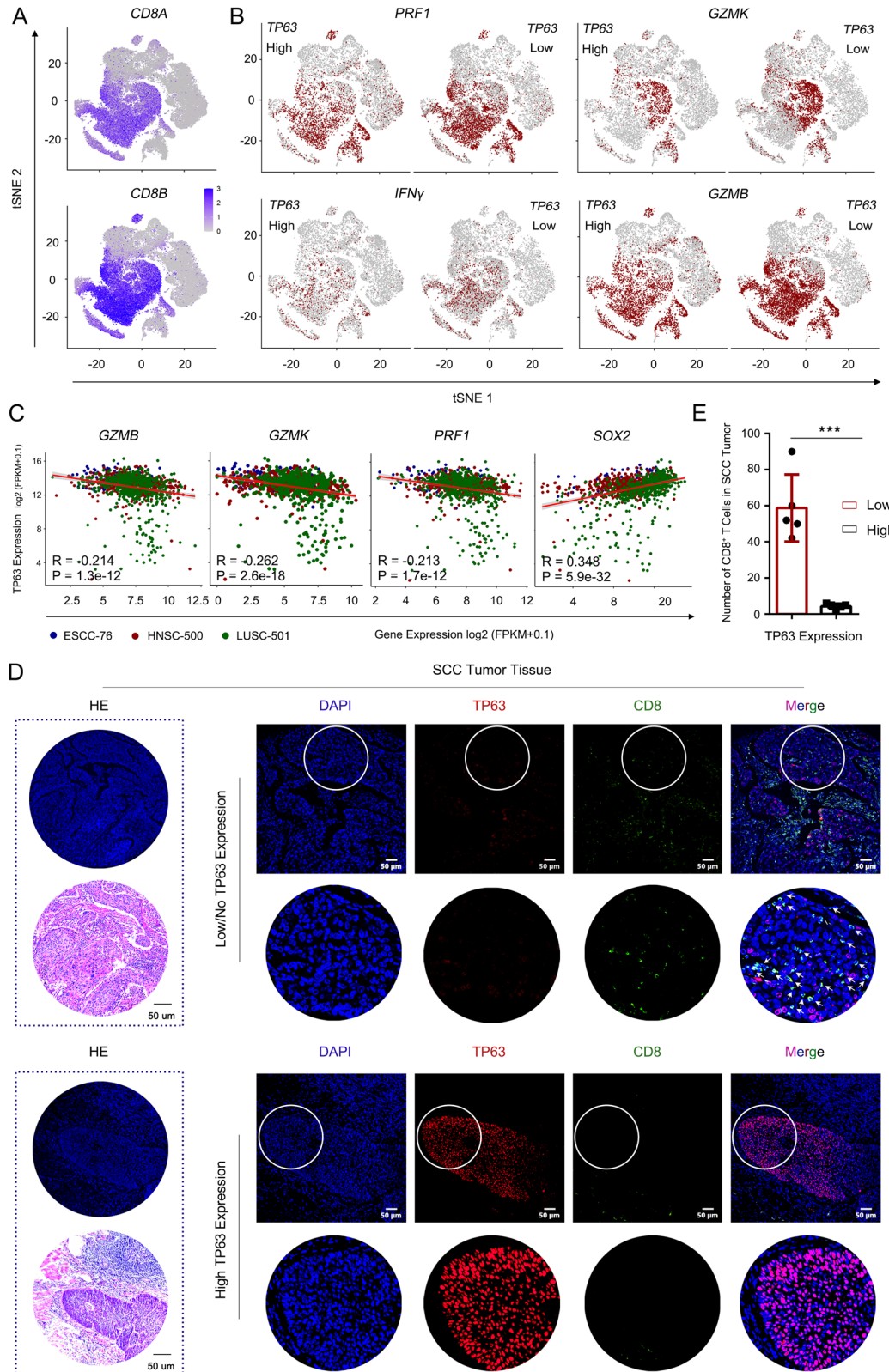

translocates into the nucleus[30,31,40,41]. We next examined the occupancy of STAT1 at those ISGs using published STAT1 ChIP-seq data, the specific binding of STAT1 was observed at the promoter and/or distal regulatory regions of many of those 23 ISGs, such as *IRF7, BATF2, B2M, IRF1, ISG20, IL15, IFI44* (Fig. 6A and Supplementary Fig. 8). These results indicate that the repression of ISGs by TP63 was possibly achieved via the interplay with STAT1.

To understand the relationship between TP63 and STAT1, we first assessed both the mRNA and protein levels of STAT1 after knockdown of *TP63*. Notably, *TP63* knockdown and IFNγ stimulation elevated both mRNA and protein levels of STAT1 as well as p-STAT1 (Fig. 6E, F and Supplementary Fig. 9C). Importantly, IFNγ-induced levels of both STAT1 and p-STAT1 were diminished by ectopic expression of *TP63* (Fig. 6G and Supplementary Fig. 9D), suggesting

**Fig. 4 | Over-expression of *TP63* impairs CD8⁺ T cells infiltration and activation.**
**A, B** t-SNE plots showing the expression levels of CD8 (**A**) and cytotoxic marker genes (**B**) of CD8⁺ T cells. Cells with the expression level of cytotoxic marker genes (TPM) high than 1 was labeled. A total of 69,278 T cells from scRNA-seq of 60 ESCC patients (GSE160269) were re-analyzed. *TP63* High: 9001 T cells; *TP63* Low: 11,012 T cells. **C** Scatter plots showing the significant negative correlation between cytotoxic marker genes and *TP63* expression in three types of SCC patient samples. SOX2 is shown as a positive control. The gene expression was extracted from TCGA bulk RNA-seq. *n* = 76 (ESCC), 500 (HNSC) and 501 (LUSC) independent patient samples, respectively. Pearson Correlation Coefficient was calculated. R: Pearson's product-moment correlation; *P* value: Two-sided *t*-test. **D, E** H&E and immuno-fluorescence (IF) staining of TP63 and CD8 in *TP63*-high or -low expressing ESCC patient samples. **D** Representative images. Zoom-in view of the area with white

circle as shown below at each right panel. White arrows denote CD8⁺ cells.
**E** Quantification of infiltrated CD8⁺ T cells. A total of 10 slides from 10 ESCC patient samples (one slide per patient) were analyzed (*TP63*-low patients, *n* = 5; *TP63*-high patients, *n* = 5). IF images were acquired at the same exposure time. The staining results were scored by two different researchers according to the fluorescence intensity of TP63; five cases with highest TP63 scores and five with the lowest scores were defined based on the median score. Five fields representing tumor regions were randomly selected for each slide, and the number of CD8⁺ T cell was counted under ×20 field of view which was then averaged across 5 fields. Statistical analysis was performed by comparing the average CD8⁺ T cell number of each slide from five *TP63*-high vs. five *TP63*-low patients. Data represent mean ± SD. *P* values were determined using a two-sided *t*-test. *P* = 0.0002. \*\*\**P* < 0.001.

that TP63 suppresses the IFNγ pathway and ISGs expression by inhibiting STAT1.

On the other hand, we unexpectedly observed strong and consistent downregulation of TP63 at both mRNA and protein levels upon IFNγ administration across multiple cell lines, in a time-dependent manner (Fig. 6H, I and Supplementary Fig. 9E and F). These results imply that TP63 and STAT1 negatively regulate the expression of each other. To test this, we utilized a STAT1-specific inhibitor, Fludarabine[42], which diminished both the mRNA and protein levels of STAT1 and p-STAT1 stimulated by IFNγ (Fig. 6J and Supplementary Fig. 10A, B). Importantly, IFNγ-induced downregulation of TP63 was fully reversed by the treatment with Fludarabine (Fig. 6J and Supplementary Fig. 10A, B). Consistent with the results obtained by the treatment of STAT1 inhibitor-Fludarabine, knockdown of *STAT1* significantly increased the protein levels of TP63 (Supplementary Fig. 10C). These results confirm the presence of a reciprocal inhibitory relationship between TP63 and STAT1.

### Antagonistic regulation between TP63 and STAT1 dictates the IFNγ signaling in immune response to SCCs

To probe the mechanistic basis of the reciprocal inhibition between TP63 and STAT1, we interrogated the genome-wide occupancy of these two TFs by ChIP-Seq in SCC cells. We identified 6.8% (1,912/28,293) of TP63/STAT1-shared peaks, 40.8% (11,546/28,293) of STAT1-unique peaks and 52.4% (14,835/28,293) of TP63-unique peaks (Fig. 7A). The majority of TP63 and STAT1 binding peaks were located at intergenic and intron regions, and 8.6-11.2% peaks were enriched at promoters (Fig. 7B). Concordantly, DNA-binding motif sequence analysis identified significant enrichment of both TF-binding motifs in shared peaks; in contrast, in unique peak sets for each TF, only corresponding TF motif was exclusively enriched (Fig. 7C). We next preformed Gene Ontology (GO) enrichment analysis based on either TP63- or STAT1-uniquely occupied regions (Supplementary Fig. 10D, E) and peak-assigning genes (Fig. 7D, E). Uniquely occupied regions by TP63 were predominantly assigned to genes enriched in processes related to epithelial cell proliferation, ERBB signaling pathway, and cell growth pathways. In comparison, those specifically bound by STAT1 were primarily associated with immune response pathways, including IFN signaling, cytokine-mediated signaling, and T cell activation signaling (Fig. 7D, E and Supplementary Fig. 10D, E).

Notably, at the *TP63* locus, we identified a distal peak co-occupied by both TFs (Fig. 7F). Importantly, using 4C-Seq and CRISPR/Cas9 genome editing, we have recently established this peak, which we termed e8, as a functional enhancer promoting the transcription of *TP63*[28]. Reciprocally, in the *STAT1* locus, we observed that TP63 and STAT1 co-occupied both *STAT1* promoter and a distal region, a possible *STAT1* enhancer with both H3K27ac and H3K4me1 marks (Fig. 7G). Therefore, we hypothesized that TP63 and STAT1 mutually suppressed the transcription of each other through regulating these co-occupied regulatory elements. To test this, luciferase reporter assays were performed (Fig. 7F–I and Supplementary Fig. 10F). Consistent with the

increased expression of STAT1 (Fig. 6E, F, H, I), IFNγ stimulation potently enhanced luciferase reporter activities of *STAT1*-promoter and *STAT1*-enhancer (Fig. 7G, H and Supplementary Fig. 10F). Notably, IFNγ administration significantly inhibited the reporter activity of e8 element of *TP63*. Importantly, STAT1 inhibitor-Fludarabine reduced the reporter activities of *STAT1*-promoter and *STAT1*-enhancer, while increased those of e8 (Fig. 7F, H and Supplementary Fig. 10F). Moreover, over-expression of *TP63* inhibited the activity of both *STAT1*-promoter and *STAT1*-enhancer, while significantly increased the activity of e8 (Fig. 7F, G, I). To determine if *TP63*-knockdown and *STAT1*-overexpression (*STAT1*-OE) have similar effects in the SCC mouse models, additional in vivo experiments were performed using either empty vector control or *STAT1*-OE SCC cells. Overexpression of STAT1 in HNM007 cells was verified using qRT-PCR and western blotting assays and then used for syngeneic mouse model experiments (Supplementary Fig. 10G). Importantly, consistent with the results from *Trp63*-knockdown mouse model, *STAT1*-OE significantly decreased tumor growth and tumor weight relative to those in the control group without affecting the body weight of mice (Supplementary Fig. 10H–J). Taken together, these results demonstrate that TP63 and STAT1 mutually suppress the expression of each other by co-occupying and co-regulating their own promoters and enhancers, which determines immune response by regulating CD8⁺ T cell activity (Fig. 8).

## Discussion

A negative correlation between the expression of TP63 and inflammatory/immune response-associated genes in HPV⁺ HNSCC tumors was recently revealed using computational GO enrichment analysis[43]. Here we reveal a tumor-extrinsic role of TP63 in promoting immune evasion of SCC cells by suppressing the IFNγ-STAT1 signaling. Moreover, our data from mouse models and human patient samples demonstrate that high expression of TP63 impedes CD8⁺ T cell infiltration and tumor killing; inhibition of TP63 enhances the efficacy of PD-1 mAb therapy. These results provide insights into how SCC cells escape immunological surveillance, and suggest targeting IFNγ-TP63/STAT1 axis as a potential strategy to improve anti-tumor immunotherapeutic effect and overcome ICB resistance of SCCs.

IFNs are a family of pleiotropic cytokines that have the potential to regulate transcription of hundreds of downstream genes, influencing protein synthesis, autophagy, apoptosis, angiogenesis, innate, and adaptive immunity[30,31,44]. Defective IFNγ signaling is associated with immune evasion and resistance to ICB therapy in melanoma[45–48], colorectal cancer[47,49,50], and breast cancer[51]. In the present study, we propose a reciprocal inhibition between TP63 and STAT1 which dictates the strength of IFNγ signaling, supported by multiple lines of evidence. First, IFNγ treatment resulted in an upregulation of STAT1 and many ISGs, including *IRF1, MHC-I, B2M,* and *BATF2,* as well as downregulation of TP63 expression. Over-expression of TP63 decreased the expression of STAT1 at both mRNA and protein levels. Secondly, TP63 and p-STAT1 physically interacted with each other at

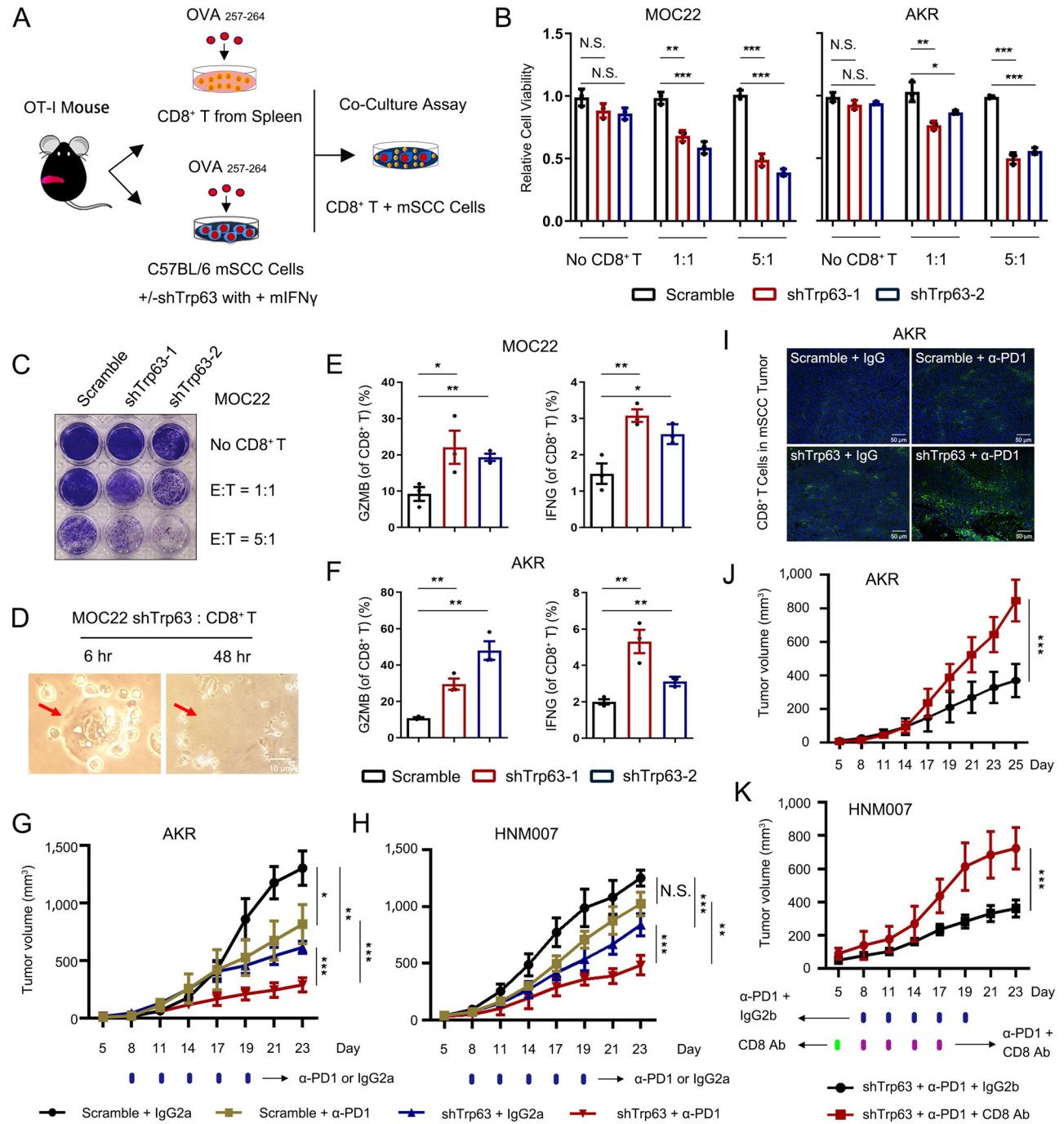

**Fig. 5 | Repression of TP63 results in efficient T cell killing and enhances the efficacy of PD-1 mAb therapy in murine SCC models. A** Flowchart of ex vivo co-culture of murine SCC and OT-I CD8[+] T cells. **B, C** Relative cell viability analysis (**B**) and crystal violet staining (**C**) of SCC cells incubation with or without OT-I CD8[+] T cells at the indicated effector: target (E: T) ratios. *P* value for each comparison from left to right in **B**: 0.1080 (N.S.), 0.0531 (N.S.), 0.0013 (**), 0.0005 (***), 0.0001 (***), 1.88E-05 (***), 0.0988 (N.S.), 0.0846 (N.S.), 0.0059 (**), 0.0249 (*), 5.43E-05 (***), 1.17E-05 (***). **D** Bright field images showing representative co-cultured MOC22 and OT-I CD8[+] T cells. Red arrows indicate a cancer cell killed by activated OT-I CD8[+] T cells following 48 hr co-culture (Scale bar, 10 μm). **E, F** Percent of GZMB and IFNγ production in Scramble or sh*Trp63* MOC22 (**E**) and AKR (**F**) co-cultures. *P* value for each comparison from left to right in **E** and **F**: 0.0467/0.0038, 0.0092/0.0020, 0.0086/0.0072, 0.0281/0.0057. The gating strategy are provided in Supplementary Fig. 6D. **G, H** Plots of AKR (**G**) and HNM007 (**H**) tumor volumes measured every 3 days. C57BL/6J mice were implanted with sh*Trp63* SCC or Scramble cells and

received PD-1 mAb treatment or IgG isotype control (IgG2a). *n* = 5 for each group. For the tumor volume comparison of Scramble + IgG2a vs. Scramble + α-PD1, Scramble + IgG2a vs. sh*Trp63* + IgG2a, sh*Trp63* + IgG2a vs. sh*Trp63* + α-PD1 and Scramble + α-PD1 vs. sh*Trp63* + α-PD1 in AKR and HNM007-derived tumor allografts: *P* = 0.0176/0.0500, *P* = 0.0014/0.0002, 2.1E-05/.00002, 0.0002/0.0013. (**I**) Representative IF staining of CD8α of Scramble and sh*Trp63* AKR allografts, mice were treated with either PD-1 mAb or IgG isotype control (Scale bar, 50 μm). The results were repeated in three biologically independent samples. **J, K** Tumor growth curves of sh*Trp63* AKR-bearing (**J**; *P* = 0.0001) or HNM007-bearing (**K**; *P* = 0.0003) mice treated with PD-1 mAb in combination with CD8 mAb (CD8 T-cell-depletion antibody; *n* = 5) or IgG isotype control (IgG2b; *n* = 5). Bars of **B**–**F** represent mean ± SD of three biologically independent experiments. *P* values were determined using a two-sided *t*-test. The data of (**G, H, J, K**) were analyzed by a one-sided *t*-test. N.S. not significant; **P* < 0.05, ***P* < 0.01, ****P* < 0.001. Source data are provided as a Source Data file.

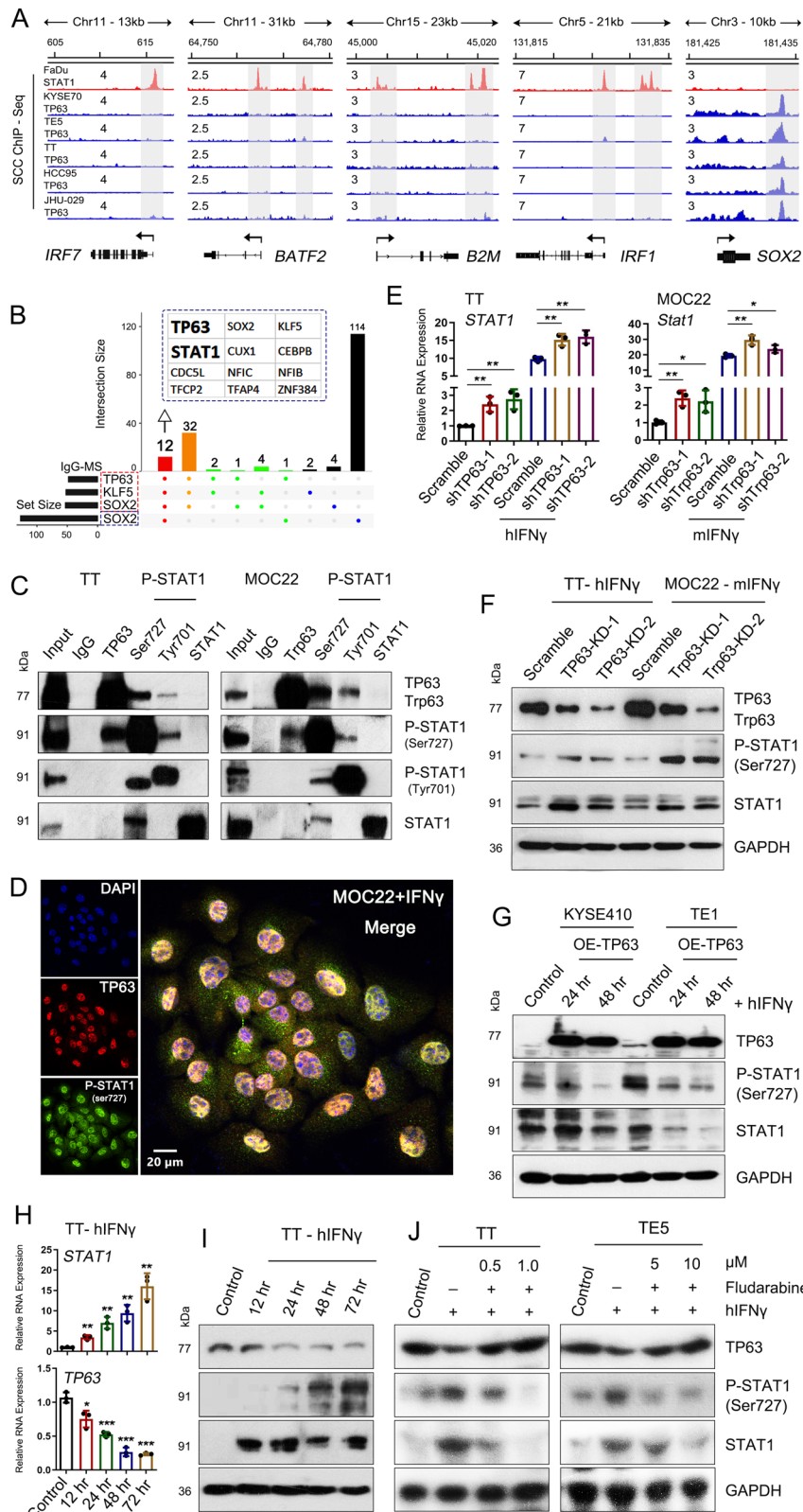

the protein level. Finally, TP63 and STAT1 form a seesaw-like transcription regulation on their own cis-regulatory elements (e8, *STAT1*-promoter, *STAT1*-enhancer) that were co-occupied by STAT1 and TP63 to mutually suppress the expression of each other. The three elements regulated the relative expression of TP63 and STAT1, which determines the strength of IFNγ signaling of SCCs.

Treatment with ICB antibodies against PD-1/PD-L1 has significantly improved outcomes of a subset of advanced SCC patients[3,7–11,52]. However, most patients either do not respond (60-80%) or develop resistance to treatment. Our findings elucidate a mechanism by which SCC cells escape immune surveillance via the function of TP63, which is often over-expressed in SCCs. Our preclinical animal studies showed

**Fig. 6 | TP63 suppresses ISGs by inhibiting STAT1. A** Integrative genomic viewer (IGV) tracks of ChIP-seq revealing binding peaks for STAT1 and TP63 on the promoter or enhancer loci of indicated IFN response genes. The occupancy of TP63 at the *SOX2* locus is shown as a positive control. ChIP-seq data were retrieved from GSE78212, GSE46837, GSE106563, and GSE148920. Gray shadows highlighting promoter region of each gene. **B** TP63-interactome analysis by cross-referencing immunoprecipitation-mass spectrometry (IP-MS). Blue, green, and red dots indicate respectively solo-, duo- or trio-interacted proteins with TP63, KLF5, or SOX2. The 12 common proteins showing interaction with TP63, KLF5, and SOX2 in both datasets were listed. Data in red dotted-line rectangle were from our previous publication[28]; SOX2-MS data was retrieved from Watanabe et al.[26]. **C** Co-IP followed by Western blotting analysis showing the protein interaction between TP63 and phosphorylated STAT1 (Ser727 and Tyr701) in both human TT and murine MOC22 cells. **D** Representative IF staining displaying the localization of TP63 and p-STAT1 (Ser727) in MOC22 cells stimulated with IFNγ (100 ng/mL). Zoom-in view as shown on the right. The similar results were repeated in three biologically independent experiments. **E, F** qRT-PCR (**E**) and Western blotting (**F**) analysis revealing relative mRNA and protein levels of STAT1, TP63 and p-STAT1 in both TT and MOC22 cells expressing Scramble or sh*TP63*/sh*Trp63* ± IFNγ (100 ng/mL) for 48 h. *P* value for the

comparison from left to right in **E**: 0.0081, 0.0099, 0.0061, 0.0049, 0.0064, 0.0314, 0.0056, and 0.0463. **G** Western blotting analysis showing levels of indicated proteins. SCC cells (KYSE410 and TE1) with low expression of *TP63* were transfected with *TP63* over-expression (OE) or empty vector (Control) following 24 and 48 hr stimulation of IFNγ (100 ng/mL). The results were repeated with three biologically independent experiments in two cell lines. **H, I** qRT-PCR (**H**) and Western blotting (**I**) analysis revealing a time-dependent expression of STAT1, TP63, and p-STAT1 in TT cells in responding to the stimulation of IFNγ (100 ng/mL). *P* value for the comparison from left to right in **H**: 0.0017, 0.0020, 0.0021, 0.0013 for *STAT1* expression; 0.0191, 0.0004, 0.0002, 5.08E-05 for *TP63* expression. **J** Western blotting showing expression of STAT1, TP63, and p-STAT1 in both TT and TE5 cells treated with IFNγ (100 ng/mL) or/and Fludarabine, a STAT1-specific inhibitor. The results of panels (**E–H, J**) were repeated with three biologically independent experiments in two cell lines. The results of panels (**C, I**) were repeated in four biologically independent cell lines. **E, H** Data represent mean ± SD of three biologically independent experiments. *P* values were determined using a two-sided *t* test. *\**P* < 0.05, **\*\**P* < 0.01, **\*\*\**P* < 0.001. Source data are provided as a Source Data file.

a markedly improved efficacy of PD-1 mAb treatment following TP63 inhibition. In addition, antibody depletion of CD8+ T cells significantly reversed the reduction of tumor volume caused by sh*TP63* and PD-1 mAb treatment, demonstrating that the function of TP63 in immune response is mediated through CD8+ T cells.

In SCC patient samples, we identified that *TP63* expression is negatively correlated with CD8+ T cell infiltration and the expression of effective cytotoxic markers such as *GZMB*, *GZMK*, and *PRF1*. Supportively, in a phase I clinical trial of ICB therapy (NCT02742935), low expression of *TP63* was shown to be enriched in a subset of "immune modulation" ESCC patients, who were more responsive to single PD-1 mAb therapy[7]. This same study also identified a 28-feature classifier associated with ICB therapy response, with *TP63* expression ranked as the second most significant feature[7]. These independent observations from patient samples underscore the importance of TP63 in regulating anti-tumor immune response, and suggest that TP63 expression might serve as a biomarker associated with immune-cold tumors in SCCs. Inhibition of TP63 may be a potential approach to turn immune-cold SCC tumors into immune-hot ones. However, TP63 has two isoforms (TAp63 and ΔNp63), which often exert opposite functions in cancer biology[53,54]. Therefore, for biomarker studies, it is imperative to discern the relative expression of TAp63 and ΔNp63.

We also find the dual function of IFNγ in both activation of STAT1 and suppression of TP63. Therefore, IFNγ administration may be beneficial for SCC patients with high expression of TP63. IFN therapies have historically been evaluated in clinical trials for cancer treatment, but one of the biggest barriers is the dose-limiting side effects[31]. Nevertheless, in certain patients with breast, melanoma and ovarian cancers as well as in preclinical animal models, encouraging effects have been observed when using IFNα, IFNβ, or IFNγ[31,55–58]. These studies also noted that IFN therapies were most effective in an early, adjuvant treatment setting. Therefore, early use of IFNγ, combined with ICB treatment may represent a potential treatment strategy for SCC patients with high expression of *TP63*.

## Methods
### Patient samples
Human ESCC and operative margin tissues were procured from surgical resection specimens. All of the patients received no treatment prior to surgery and signed separate informed consent forms for sample collection. Formalin-fixed paraffin-embedded (FFPE) samples from 10 ESCC patients were used for H&E and immuno-fluorescence staining analysis. The study was approved by the Ethics Committee of the Cancer Institute (Hospital), Chinese Academy of Medical Sciences (CAMS) & Peking Union Medical College (PUMC) (No. 16-171/1250).

### Mice
C57BL/6 J mice were purchased from Jackson Laboratory or Gem Pharmatech (Nanjing, China). OT-I TCR (C57BL/6-Tg (TcraTcrb) 1100Mjb/J) transgenic mice were purchased from The Jackson Laboratory or provided by Dr. Zhengfan Jiang (Beijing, China) and housed in a specific pathogen-free facility in Cedars-Sinai Medical Center (Los Angeles, USA) or Hefei Institutes of Physical Science, Chinese Academy of Sciences (Hefei, China), respectively. Mice were used between 6 and 12 weeks of age. Animal studies were respectively approved and performed according to the ethical regulations of Cedars-Sinai Institutional Animal Care & Use Committee (CSMC IACUC) and the animal care regulations of Hefei Institutes of Physical Science, Chinese Academy of Sciences.

### Cell lines
TE5 and TT cell lines were kindly provided by Dr. Koji Kono (Cancer Science Institute of Singapore, Singapore). AKR and HNM007[59,60] cell lines were a gift from Dr. Anil K. Rustgi (Columbia University Irving Medical Center, USA). MOC1, MOC22[61] were from the laboratory of Dr. Ravindra Uppaluri (Dana-Farber Cancer Institute, USA). TE5 and TE1 cells were grown in RPMI-1640 (Biowest; Wisent) medium supplemented with 10% FBS (Wisent; Biowest), and 1% penicillin/streptomycin (Gibco). TT, AKR and HNM007 cells were cultured with Dulbecco's modified Eagle's medium (DMEM) (Biowest and Wisent) containing 10% FBS and 1% penicillin/streptomycin. MOC1 and MOC22 cells were cultured in IMDM and Ham's Nutrient Mixture F12 media (at 2:1 v/v, Hyclone) supplemented with 5% FBS, 1% penicillin/streptomycin, 5 μg/mL insulin, 40 ng/mL hydrocoritsone and 5 ng/mL EGF. All cell lines were cultured in a humidified incubator at 37 °C in 5% $CO_2$ and passaged every 2–3 days. All the cell lines were authenticated through short tandem repeat (STR) analysis before using in experiments.

### Construction of expression vectors
To silence TP63 in human cells, two previously verified shRNA vectors were used by respectively targeting all *TP63* isoforms (shRNA-1) and specifically targeting only *ΔNp63α* isoform (shRNA-2)[28]. In murine cell lines and mouse models, a tetracycline-inducible shRNA-expressing vector-Tet-pLKO-puro (Addgene, #21915) was used (Supplementary Table 1). Synthesized forward and reversed oligos were annealed to be double-stranded oligonucleotide shRNAs and then cloned into the AgeI/EcoRI sites of the pLKO.1-TRC (Addgene #10878) or Tet-pLKO-puro vector. pcDNA3.1/hygro and deltaNp63alpha-FLAG (Addgene, # 26979) plasmids were purchased from Addgene (Supplementary Table 1). The siRNA, shRNA, and primer sequences were listed in Supplementary Table 2.

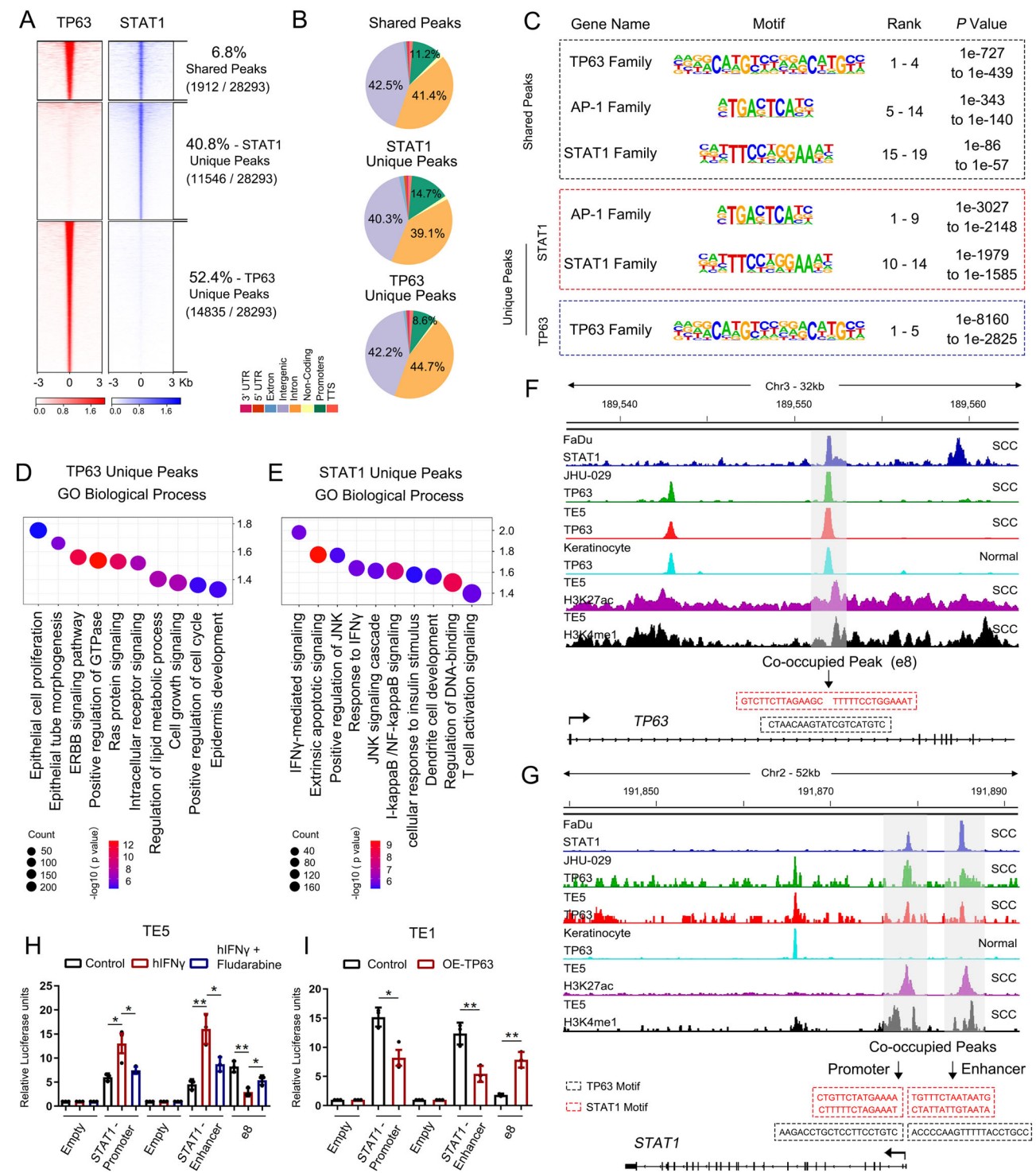

**Fig. 7 | Antagonistic transcription regulation between TP63 and STAT1 in SCCs.**
**A** Density plots of ChIP-seq signals of TP63 and STAT1 at ± 3 kb windows flanking the center of TP63 peaks. Color bars at the bottom show reads-per-million-normalized signals. Data were from GSE46837, GSE106563, and GSE148920. **B** Pie chart showing genome-wide distribution of the regions uniquely- or co-occupied by TP63 and STAT1. **C** Representative top TF motif sequences enriched at TP63 or/ and STAT1 uniquely- or co-occupied loci revealed by de novo motif analysis. **D, E** Gene Ontology (GO) functional categories of TP63 (**D**) or STAT1 (**E**) uniquely occupied peak-assigning genes in **A. F, G** IGV tracks from ChIP-seq data of indicated factors at either *TP63* (**F**) or *STAT1* (**G**) gene loci. Gray shadows highlighting enhancer or promoter elements that are co-occupied by TP63 and STAT1. Black font sequences: TP63 binding motif; Red font sequences: STAT1 binding motif. **H, I** Relative luciferase activity of pGL3-enhancer (1st Empty), pGL3-enhancer

+*STAT1* promoter, pGL3-promoter (2nd Empty), pGL3-promoter+ *STAT1* enhancer, pGL3-promoter+e8 upon stimulation with IFNγ (100 ng/mL) +/− Fludarabine (10 μM) in *TP63*-high TE5 cells (**H**) or over-expression of *TP63* in *TP63*-low TE1 cells (**I**). Relative luciferase activity comparison of Control vs. IFNγ, IFNγ vs. IFNγ + Fludarabine in **H:** *P* = 0.0251 (*) and *P* = 0.0492 (*) for *STAT1*-promoter group, *P* = 0.0037 (**) and *P* = 0.0213 (*) for *STAT1*-enhancer group, *P* = 0.0033 (**) and *P* = 0.0318 (*) for e8 group. Relative luciferase activity comparison of Control vs. OE-TP63 in **I:** *P* = 0.0137 (*) for *STAT1*-promoter group, *P* = 0.0068 (**) for *STAT1*-enhancer group, *P* = 0.0016 for e8 group. **C–E** Statistical analysis was determined by hypergeometric test. **H, I** Data represent mean ± SD of three biologically independent experiments. *P* values were determined using a two-sided *t*-test. Source data are provided as a Source Data file.

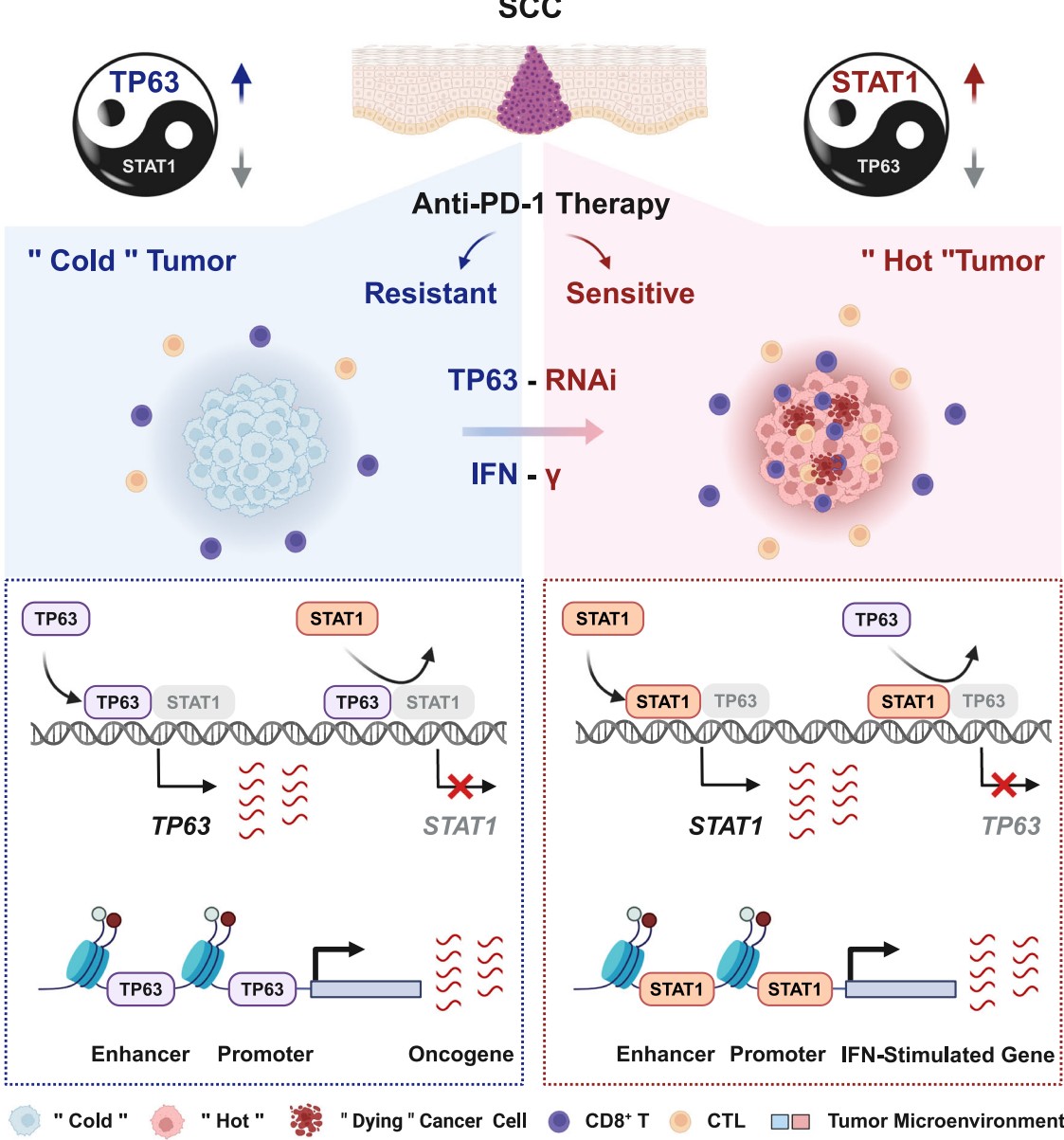

**Fig. 8 | Proposed model of TP63-suppressed immune response in squamous cancer.** Squamous cell master regulator TP63 represses the IFNγ signaling through antagonizing STAT1 in occupying the regulatory elements of themselves. The relative expression between STAT1 and TP63 dictates the strength of IFNγ signaling, which determines immune response by regulating CD8+ T cell activity. Inhibition of TP63 improves anti-tumor effect of PD-1 mAb therapy. The Figure was created with BioRender.com.

## Cell transfection and lentiviral production

Cells were seeded in 6/12-well plates at 40% to 60% confluence. After 16 to 24 h, siRNA or constructed vectors were transfected into cells using Lipofectamine RNAiMAX (Thermo Fisher Scientific) or jetPRIME (Polyplus) according to the manufacturer's instructions. Lentivirus production was performed as described previously[32,62]. 24 hour prior to transfection, cells were plated in a 6-well plate at 70–90% confluence. The next day, the recombinant lentiviral vectors and packaging vectors (psPAX2 and pMD2.G) at 1:1 ratio were co-transfected into HEK-293T cells. Supernatants containing lentiviral particles were collected at 48 h timepoint and filtered through a 0.45 μM filter.

## Generation of inducible shRNA-expressing cell lines

For the development of stable shRNA-expressing cell line models, SCC cell lines were infected with lentiviruses containing either inducible non-targeting shRNA (Scramble) or inducible *Trp63* shRNA and were

then positively selected in 10 μg/mL puromycin (Sigma-Aldrich). The efficient knockdown of *Trp63* was verified by western blot analysis.

## Syngeneic murine SCC models

For the syngeneic murine SCC models and allograft experiments, 6-week-old male C57BL/6 J mice were used. Constructed stable murine SCC cells were subcutaneously to injected into the flanks of mice (5×10⁵–1×10⁶ cells for each injection). Three different murine SCC cell lines derived either from oral cavity (MOC22) or esophagus (AKR and HNM007) were utilized. Once tumor reached an appropriate size (about 100 mm³; 10 days after inoculation of MOC22 and 5 days of AKR and HNM007), doxycycline (Dox, 2 mg/mL) were added in drinking water to silence the expression of *Trp63*. Tumor growth was monitored over 4–6 weeks, and tumor size was measured every 3 days. Mice were euthanized with isoflurane (RWD) followed by cervical dislocation at the end of the experiment (7 days or 20 days after cell injection) or

tumor exceeding the size endpoint of 1500 mm³. Tumors were harvested thereafter for scRNA-seq and flow cytometry analysis. Meanwhile, paraffin-embedded tumor blocks were prepared for further analysis.

### In vivo treatments
For in vivo treatment, indicated SCC cells were injected subcutaneously into C57BL/6 J mice and sh*Trp63*-expressing was induced with Dox as mentioned above. To test whether silencing of *Trp63* enhanced the effect of PD-1 mAb therapy, these tumor-bearing mice were treated with either PD-1 mAb (BE0146, BioXcell) or IgG isotype control (IgG2a, BE0089, BioXcell). To examine whether tumor-suppressing functions of TP63-mediated effects was dependent on CD8$^+$T cells, either CD8 depletion mAb (BE0061, BioXcell) or IgG isotype control (IgG2b, BE0089, BioXcell) was co-treated with PD-1 mAb. All the mAbs were given by intraperitoneal injection (i.p., 100 μg/injection/mouse) once every 3 days for up to 2 weeks. For the CD8 mAb blocking experiments, CD8 mAb was conducted beginning on the 5$^{th}$ day of the transplantation of tumor cells. The specific treatment timepoints were labeled in the main Figures and legends.

### T cell-mediated tumor cell killing assays
**Isolation and in vitro activation of CD8$^+$ T Cells.** Splenocytes were isolated and collected from the spleen of OT-I TCR mice by gently mashing using 1 mL plunger end of the syringe in complete RPMI 1640 medium containing 10% FBS. The red blood cells were lysed with RBC Lysis Buffer (420301). Splenocytes were stimulated by incubation with 300 ng/mL SIINFEKL (OVA$_{257-264}$ peptide) (Sigma-Aldrich) for 24 hr and cultured in media supplemented with IL-2 (100 U/mL (Peprotech) for 2 days. CD8$^+$ T cells were then subsequently isolated using MojoSort™ Mouse CD8 T Cell Isolation Kit (480007) and cultured with IL-2 for an additional 3 days before use in co-culture assays. Successful enrichment and activation of CD8$^+$ T cells were confirmed by flow cytometry using antibodies against CD45 (103112), CD3 (100233) CD8 (100706), and CD69 (104507). All the antibodies and agents for flow cytometry were purchased from BioLegend except for special statement.

**T-cell cytotoxicity assays.** Murine SCC cells transfected with either sh*Trp63* or non-targeting control (Scramble) vector and incubated with 10 ng/mL IFNγ. After 24 hr, SCC cells were pulsed with 1 ng/mL of OVA$_{257-264}$ for 2 hr and plated in 96-well plates. Activated OT-I CD8$^+$ T cells were then added and incubated with tumor cells at different effector: target ratios (E:T = 1:1 and 5:1). After 48 hr, both SCC cancer cells and CD8$^+$ T cells were harvested and subjected to cytotoxicity analysis. Living SCC cells were quantified by a spectrometer at OD (570 nm) followed by crystal violet staining; while the cytotoxicity of CD8$^+$ T cells was measured by flow cytometry staining of GZMB (372212) and IFNγ (505823). The specific schematic view was described in the main Figures and legends.

### Flow cytometry
For mouse samples, single-cell suspension of AKR and HNM007-allograft tumors were obtained by manual dissociation with rapid physical grinding and digestion in 5 mg/mL collagenase IV (Invitrogen) and 0.25% Trypsin-EDTA (Gibco) supplemented with 10 U/mL DNase I (Sigma-Aldrich). Immune cells were enriched using CD45 MicroBeads (Miltenyi Biotec) and blocked with Fc Block™ (anti-mouse CD16/32) antibody (BD Biosciences). Live cells were determined with Fixable Viability Stain 450 (562247, BD Biosciences) then stained with APC-CD45 (103112), BV510-CD3 (100233), FITC-CD8 (100706), PE-CD69 (104507) on ice for 20 min. To quantify intracellular GZMB and IFNγ, PerCP/Cyanine5.5-GZMB (372212) and AF700-IFNγ (505823) antibodies were respectively incubated after fixation and permeabilization by True-Nuclear™ Transcription Factor Buffer Set (424401).

For T-cell cytotoxicity assays, CD8$^+$ T cells were harvested and incubated with PerCP/Cyanine5.5-GZMB antibody (372212). Data were acquired on a CytoFLEX (Beckman, USA) and analyzed using FlowJo v10.8.1 software.

### Western blotting and co-immunoprecipitation
Adherent cells were washed with 1× PBS and dislodged using a cell scraper. Proteins were extracted using RIPA lysis buffer (Merck Millipore; Solarbio) supplemented with protease inhibitors and phosphatase inhibitors (Roche). Protein concentrations were measured by BCA protein assay kit (Thermo Fisher Scientific; Beyotime). Equal amount of protein for each sample was loaded for SDS-PAGE separation and transferred onto the PVDF membrane (Merck Millipore) for determination. Primary antibodies were incubated overnight at 4 °C on a shaker, followed by incubation with secondary antibody for 1–2 hr at room temperature. Membranes were incubated with ECL detection reagent (Vazyme, China) for 1 to 2 min, and the results were visualized on X-ray film in a dark room.

For immunoprecipitation, 500 μg cell lysate was incubated with either indicated antibody or IgG overnight at 4°C. The immunoprecipitated complex were then incubated with Protein A/G (Thermo Fisher Scientific; Beyotime) for 4 hr at 4°C, followed by purification and Western blotting assays. The antibodies used for Western blotting and co-IP were listed in the Supplementary Table 1.

### Immunofluorescence
Sections of formalin-fixed paraffin-embedded tissues were cut into 5 μm sections using a microtome (Leica, Germany). Slides were baked for 2 hr at 65°C, and then deparaffinized with xylene and rehydrated with ethanol (100%-90%-80%-70%). Antigen was retrieved at sodium citrate antigen retrieval solution (Solarbio, China) using microwave and maintained at a sub-boiling temperature for 15 min. The samples were blocked for 1 hr at 37 °C in blocking buffer consisting of 0.3% Triton X-100 (Beyotime) and 3% donkey serum (Kang Yuan Biology, China) in PBS. Primary antibodies including CD8α (1:300; 70306, Cell Signaling Technology) and p63 (1:100; GTX102425, GeneTex) were incubated overnight at 4 °C. The next day, the samples were incubated with fluorescence conjugated secondary antibody (Thermo Fisher Scientific) for 1 hr at 37 °C and mounted with a mounting medium containing DAPI (ZSGB-BIO, China). Immune fluorescence images were acquired using an Olympus SpinSR10 spinning disk confocal super-resolution microscope (Olympus, Japan).

### Dual luciferase reporter assay
The luciferase reporter vectors were purchased from Promega. The DNA fragments of each promoter or enhancer identified by ChIP-seq were amplified using PCR and then inserted into pGL3-enhancer or pGL3-promoter vector, respectively. Transfection was performed as mentioned above. A Renilla luciferase vector was co-transfected as a control for normalization. The luciferase activity was measured with a Dual-Luciferase Assay (Promega; YEASEN) using a SpectraMax i3x Multi-Mode Microplate Reader (Molecular Devices, USA) according to the manufacturer's instructions. The primers used for the amplification of each region were listed in Supplementary Table 2.

### RNA isolation, cDNA synthesis, and quantitative real-time PCR
Total RNA was extracted with RNeasy Mini kit (QIAGEN) or Steady Pure Universal RNA Extraction Kit (Accurate Biology, China), and cDNA was synthesized with Maxima™ H Minus cDNA Synthesis Master Mix with dsDNase (Thermo Fisher Scientific) or Evo M-MLV RT Premix (Accurate Biology, China). The quantitative real-time PCR analyses were conducted with PowerUp™ SYBR™ Green Master Mix (Thermo Fisher Scientific) or SYBR® Green Premix Pro Taq HS qPCR Kit (Accurate Biology, China). The housekeeping gene *GAPDH* was used for

normalization. Primers used in the study were listed in the Supplementary Table 2.

## Mouse tumor dissociation and CD45⁺ immune cells selection

Resected mouse tumors from either non-targeting control or sh*Trp63* C57BL/6 J mice were dissociated into single cells using the Miltenyi Mouse Tumor Dissociation Kit (Miltenyi Biotec) as per the manufacturer's protocol, reducing enzyme R to 20%. Enrichment of CD45⁺ cells was performed using CD45 MicroBeads and the autoMACS Pro Separator (Miltenyi Biotec). Cells positive for CD45 were subsequently used in single-cell RNA-seq analysis.

## scRNA-seq library construction using the 10x genomics chromium platform

scRNA-Seq libraries were prepared per the Single Cell 3′ v3.1 Reagent Kits User Guide (10x Genomics, Pleasanton, California). Barcoded sequencing libraries were quantified by quantitative PCR using the Collibri Library Quantification Kit (Thermo Fisher Scientific, Waltham, MA). Libraries were sequenced on a NovaSeq 6000 (Illumina, San Diego, CA) as per the Single Cell 3′ v3.1 Reagent Kits User Guide, with a sequencing depth of ~40,000 reads/cell.

## Processing of scRNA-seq data of mouse CD45⁺ Cells

Raw reads were aligned to the mm10 reference genome using Cell Ranger (v.7.0.0) with default parameters[63], and the expression matrices were generated per sample. After removing cells that had a total number of molecules less than 800, unique gene counts over 8000 or less than 200, or mitochondrial RNA content greater than 10%, 854 and 1057 cells were retained in Scramble and sh*Trp63* MOC22 tumors, respectively. Then, the Seurat (version 4.1.0) anchoring integration method was applied to correct the technical differences between datasets[64]. We chose 2000 highly variable genes to identify anchors using FindIntegrationAnchors and then integrated two datasets together with IntegrateData. Next, we performed a linear transformation with ScaleData and performed principal component analysis (PCA) using RunPCA. Graph-based Louvain clustering was then performed on the top 10 principal components (PCs) using FindClusters with the option "resolution=0.6". We identified the cluster-specific marker genes using the Wilcoxon test implemented in the FindAllMarkers function. The cell clusters were manually annotated according to these marker genes. Gene expression and clustering results were visualized on a UMAP plot using RunUMAP. Specifically, we reanalyzed CD8⁺ T cells to identify their subclusters by using the same method as described above.

## Annotation of murine immune cell types

Cell types were annotated based on the expression of known markers, i.e., *Cd8a* for CD8⁺ T cells; *Cd4* and *Foxp3* for CD4⁺ T cells; *Il17a* for Th17 cells; *Cd3d, Spry2* and *Bcl2* for NKT cells; *Gzma* and *Klra4* for NK cells; *Cd19, Cd79a,* and *Ly6d* for B cells; *Apoe, Ms4a7,* and *Lyz2* for Macrophage 1; *Ccl7* and *Mrc1* for Macrophage 2; *Plac8* and *Chil3* for Macrophage 3; *S100a9* and *S100a8* for Neutrophils 1; *Cd24a* and *Cyp4f18* for Neutrophils 2; *Fscn1, Ccr7* and *Ccl22* for Dendritic cells. For subclusters of CD8⁺ T cells, *Pdcd1, Ctla4* and *Icos* for exhausted T cell; *Ccl3* and *Jun* for effector T cells; *Tcf7* for naïve T cells.

## Reanalysis of scRNA-seq of human ESCC samples

We downloaded the raw expression matrices of CD45⁻ and CD45⁺ cells in 60 ESCC tumor and 4 adjacent normal tissue samples from GSE160269[20]. After normalizing the expression matrices and finding highly variable genes, we performed graph-based clustering on CD45⁺ and CD45⁻ cells respectively, and labeled the clusters using the cell information provided in GSE160269. For CD45⁺ dataset, *TP63* expression was extracted from each cell, and the average level was calculated for each ESCC tumor. For the CD45⁻ dataset, we extracted all T cells and identified their subclusters by using a similar method described above with the option "dims = 1:30, Resolution=1.5". Immune cell types were identified according to the expression of known markers provided in the original paper. Gene expression and clustering results were visualized on a tSNE plot using RunTSNE. We next calculated the fractions of macrophages, dendritic cells, B cells, CD8 T cells, effector T cells, memory CD8 T/ CD4 T cells, and exhausted T cells for each ESCC tumor and explored the correlation between the expression of *TP63* and the immune cell fraction across 60 ESCC tumors. We further binned ESCC tumors into *TP63*-high and *TP63*-low groups based on the top and bottom 15% of *TP63* expression, and compared their immune cell distribution.

## Hallmark pathway enrichment analysis

Cancer type with *TP63* average FPKM expression >=1 in TCGA project was chosen for Hallmark pathway enrichment analysis. For each cancer type, we binned tumor samples into *TP63*-high and *TP63*-low groups according to the top and bottom 30% of *TP63* expression. Then, the expression fold change of each gene was calculated by a comparison between these two groups. GSEAPrernked was next performed using the fold change as the input and the cancer hallmark gene sets as the library[65]. The enriched hallmarks with FDR < 0.05 were identified.

## ChIP-seq analysis

The following ChIP-seq datasets were collected: TP63 ChIP-seq in the TE5 cell line (GSE148920)[28] and STAT1 ChIP-seq in the FaDu cell line (GSE78212). After alignment to the hg19 reference genome, uniquely mapped reads were retained and sorted by the SAMtools (version 1.7) program[66]. We then removed PCR duplicates by the Picard MarkDuplicates tool (version 1.136) and further subtracted ENCODE blacklist regions using bedtools. ChIP-seq peaks were called by MACS2 (version 2.2.6) with the default parameters[67]. According to the overlap between TP63 and STAT1 peaks, we separated the binding regions into shared, TP63-unique and STAT1-unique peak sets. Sequence motif analyses were then performed through the HOMER findMotifsGenome.pl script using each peak set as the foreground and random regions as the background[68]. The potential TF-binding sequences with *q* value < 0.05 were reserved.

## Quantification and statistical analysis

Significant difference between two groups was analyzed using unpaired student *t*-test. Data was reported as the mean ± SD or independent replicates shown as individual data points, as indicated in the figure legends. Significance was defined as *P* < 0.05. Statistical analysis was carried out using GraphPad Prism 8. Correlation between variables was calculated by Pearson rank correlation coefficient. Details of statistical analysis, the number of samples for each group (n) and *P*-values were described in the results and figure legends.

## Reporting summary

Further information on research design is available in the Nature Portfolio Reporting Summary linked to this article.

# Data availability

Materials and reagents used in this study are listed in the Supplementary Table 1. The scRNA-seq data generated in this study have been deposited in the NCBI-GEO database under accession code GSE221938 and are publicly available. Other accession numbers from publicly available datasets used in this study are listed in Supplementary Table 1. The remaining data are available within the article, Supplementary Information or Source Data file. Source data are provided with this paper.

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

## Acknowledgements

We are very grateful for the technical support of the Applied Genomics, Computation, and Translational (AGCT) Core, Cedars-Sinai Medical Center for the generation of scRNA-seq data. We acknowledge the animal core facility at Hefei Institutes of Physical Science (HFIPS), Chinese Academy of Sciences (CAS), and Cedars-Sinai Medical Center. We thank Dr. Xin Zhang (CAS Key Laboratory of High Magnetic Field and Ion Beam Physical Biology, HFIPS, CAS) for offering the confocal microscope imaging system. We thank Dr. Li Fu (Shenzhen University International Cancer Center) for sharing the mEC25 cell line. This work was supported by grants from the National Natural Science Foundation of China (82103315 and 82273010 to Y.Y.J., 32100476 and 82372697 to Y.J.), the DeGregorio Family Foundation, The Savone Family and the Esophageal Cancer Awareness Association (to Y.Y.J.), Anhui Provincial Natural Science Foundation (2208085J44 to Y.Y.J. and 2108085MC77 to Y.J.), Open Funding of the State Key Laboratory of Molecular Oncology (SKLMO-KF2021-11 to Y.Y.J.), the CASHIPS Director's Fund (BJPY2022B09 to Y.Y.J. and YZJJQY202403 to Y.J.).

## Author contributions

D.C.L. and Y.Y.J. conceived and directed the study. Y.J., Y.Y.J. and D.C.L. designed the experiments. Y.J., Y.Y.J., S.K., J.D., F.W., D.Z., J.Z., S.Z. and J.C. generated in vivo murine SCC models and performed murine surgeries, scRNA-seq, flow cytometry and in vitro experiments. Y.Z. and Y.W.Z. conducted bioinformatic and statistical analysis. D.D.W., B.Z., S.G., J.J.H., C.Y., M.H., Z.L. and W.Q.W. contributed resources or critical feedback on the project. A.H., U.S. and H.P. provided pathology and clinical expertize. Y.J., Y.Y.J., D.C.L. and M.R.W. analyzed the data and generated figures. Y.Y.J., D.C.L. and H.P.K. supervised the research. Y.J. and Y.Y.J. wrote the original draft, together with Y.Z., Y.W.Z., D.C.L. and H.P.K. reviewed and edited the manuscript with inputs from all authors.

## Competing interests

Y.Y.J., Y.J., J.D., S.K., D.Z. and C.Y. are named inventors on patent pending related to this study entitled "Application of Trp63 inhibition in improving the efficacy of immunotherapy for squamous cancer" (identification number: CNIPA. 202311497087.3). Other authors declare that they have no other competing interests.

## Additional information

[1]Institute of Health and Medical Technology, Hefei Institutes of Physical Science, Chinese Academy of Sciences, Hefei 230031, China. [2]University of Science and Technology of China, Hefei 230026, China. [3]Clinical Big Data Research Center, Scientific Research Center, The Seventh Affiliated Hospital of Sun Yat-sen University, Shenzhen 518107, China. [4]Center for Craniofacial Molecular Biology, Herman Ostrow School of Dentistry, and Norris Comprehensive Cancer Center, University of Southern California, Los Angeles, CA 90033, USA. [5]Department of Medicine, Samuel Oschin Cancer Center, Cedars-Sinai Medical Center, Los Angeles, CA 90048, USA. [6]State Key Laboratory of Molecular Oncology, Center for Cancer Precision Medicine, National Cancer Center/National Clinical Research Center for Cancer/Cancer Hospital, Chinese Academy of Medical Sciences and Peking Union Medical College, Beijing 100021, China. [7]Institutes of Physical Science and Technology, Anhui University, Hefei 230601, China. [8]Division of Otolaryngology-Head and Neck Surgery, Department of Surgery, Samuel Oschin Cancer Center, Cedars-Sinai Medical Center, Los Angeles, CA 90048, USA. [9]Department of otolaryngology, Keck School of Medicine, University of Southern California, Los Angeles, CA 90033, USA. [10]Department of Thoracic Surgery, The First Affiliated Hospital of Anhui Medical University, Hefei 230022, China. [11]Hefei Cancer Hospital, Chinese Academy of Sciences, Hefei 230031, China. [12]Cell-Bionomics Research Unit, Innovative Integrated Bio-Research Core, Institute for Frontier Science Initiative, Kanazawa University, Kanazawa, Ishikawa, Japan. [13]Department of Gastrointestinal Oncology, Key Laboratory of Carcinogenesis and Translational Research (Ministry of Education), Peking University Cancer Hospital and Institute, Beijing 100142, China. [14]Department of Cancer Epidemiology, National Cancer Center/National Clinical Research Center for Cancer/Cancer Hospital, Chinese Academy of Medical Sciences and Peking Union Medical College, Beijing 100021, China. [15]These authors contributed equally: Yuan Jiang, Yueyuan Zheng, Yuan-Wei Zhang, Shuai Kong. ✉e-mail: dechenli@usc.edu; yanyij@cmpt.ac.cn

