## [Peer Review File · Nature Communications]

Reciprocal inhibition between TP63 and STAT1 regulates anti-tumor immune response through interferon- γ signaling in squamous cancerREVIEWER COMMENTS

Reviewer #1 (Remarks to the Author):

Yuan Jiang et al. are evaluating whether TP63, the master transcription factor commonly expressed in squamous cell carcinomas (SCC), has immunosuppressive tumor-extrinsic properties that may play a role in immune checkpoint blockade (ICB) resistance. The authors explore whether and how TP63 suppresses IFN signaling, CD8+ T cell infiltration and tumor killing using a variety of in vivo and ex vivo systems. The authors conclude that TP63 and STAT1, mediator of IFN signaling, mutually suppress each other by co-occupying and co-regulating their own promoters and enhancers. Further, they state that TP63 upregulation negatively correlates with CD8+ T cell infiltration and activation, and that silencing of TP63 enhances the anti-tumor efficacy of anti-PD-1 treatments by promoting superior CD8+ T cell infiltration and functionality. While this study is exploring an important topic, data are underdeveloped. The following items need to be addressed:

Major conceptual critiques:

1. The major problem of this study is that it does not clearly set the boundary between constitutive (cell-intrinsic) and exogenous IFN γ /IFN α -induced STAT1 signaling pathways. This blurring of lines between spontaneous and inducible gene signatures is an issue that affects the whole story, and an issue that needs to be carefully addressed in each figure.
2. Lack of translation of ex vivo and in vivo models using human patient specimens (e.g., robustly evaluating correlation between TP63 expression and CD8 infiltrate; p-STAT1 vs TP63 staining in tumor tissues). If access to human tumor tissues is limited, greater reliance on publicly available bulk RNAseq and scRNAseq datasets should be utilized.
3. Numerous figures/images seem to be representative examples of their data (multicolor IF microscopy and, especially, Western blots). The authors need to show robust statistical analyses, where they are evaluating multiple independent experiments before making definitive claims (e.g., for Western blots, they need densitometry analyses). In supplementary figures they could show individual representative images.

Major experimental critiques (in order of appearance in the paper):

1. MOC22 data shown in Figure S1F is not showing significant differences in 23 IFN gene upregulation following TP63 knockdown. TT data looks more convincing, albeit the magnitude of expression is vastly different between these two cell lines. It would be beneficial if the authors could validate these findings with additional human and mouse cell lines. Additionally, there are no statistical analyses shown outside of standard error of mean.
2. At least some of these observations in Figure S1F should be validated on a protein level.
3. Top panel of Figure S1F is mislabeled. It should be TT and not MOC22.
4. In Figure 2, the authors are using a TP63 knockdown mouse tumor model. What is the time-kinetics of gene knockdown? Is TP63 expression suppressed for three weeks (during the duration of an in vivo experiment) or is it a transient knockdown? How quickly does TP63 expression bounce back and how does this affect your experiment outcome?
5. The total number of TIL analyzed by scRNAseq (1,911) is very low and indicates that likely only one mouse was evaluated per test group (scrambled vs. TP63 siRNA). This is not specified in the figure legend but is inferred. Additional experiments are needed to confirm these observations.
6. What genes were used to identify different CD8+ T cell activation states? Also, what are CD8 T cells labeled as "Others?" Nomenclature used is a bit outdated and vague. For example, GZMK+ CD8 effector memory T cells have been associated with favorable response to anti-PD-1 therapy (Caushi, J.X. et al. Nature. 2021).
7. Figure 2I lacks the gating strategy and staining examples. Also, were these cells analyzed after stimulation or without?
8. Figure 3 needs a citation of cell annotations (or show the genes used to identify them).
9. Interpretation (or wording) of Figure 3B is incorrect. It is stated that "the abundance of T cells" negatively correlated with TP63 expression levels. Due to the nature of this dataset, CD45+ and CD45- cells were enriched prior to sequencing. Consequently, we do not know the abundance of infiltrate, but do know the frequency/representation of total T cells. The authors need to validate these claims by using IHC or mIF staining experiments.
10. In Figures 4A -> 4B, the authors are relying on tSNE figures to infer CD8 T cell activity in TP63low and TP63high tumors. This should be done in a statistical way. Also, these data optimally need to be validated with suitable in vitro killing assays.
11. What is Figure 4C showing? Are dots representing individual patients?

12. In Figure 4D and 4E, the authors look like they are showing representative images of multicolor IF. They need to summarize and present the data in a statistically relevant way. Also, how are they choosing TP63 low and high samples? They need to have at least 4-6 patients in each cohort. How are tumor sections chosen? How were tumor areas analyzed chosen? Are data normalized somehow? For example, are they only looking at tumor beds or tumor stroma?
13. For Figure 5, the authors do not address a critical question. Does TP63 overexpression drive MHC class I downregulation? Or, does TP63 upregulation increase PD-L1 expression on tumors? Or is there a different mechanism by which this transcription factor hampers CD8 T cell-mediated killing of tumors? These questions need to be addressed.
14. Figures 5J and 5K are showing the same data, except one shows average and other individual tumor growth curves. What about HNM007? These data should be added.
15. Description of Figure 6 suggests that TP63 knockdowns lead to constitutive STAT1 phosphorylation. That is not visible. The effect of TP63 knockdown is unclear as it is measured by concurrent IFN γ treatment. These data suggest synergistic action of TP63 knockdown and IFN γ treatment.
16. In Figure 6G, the authors need an experiment where cells are either untreated or treated with IFN γ over the same time course. This is needed to clearly distinguish between TP63 and IFN γ -driven effects.
17. In Figure 6J, effect of IFN γ inhibitor on TP63 expression is ambiguous. Data needs to be normalized for GAPDH and multiple experiments performed and plotted.
18. One of the more interesting pieces of data is buried in Figure S7H. In this figure, gene-set enrichment analysis is performed on TP63 unique peaks. Here, the authors show that TP63 may be regulating the WNT signaling pathway. WNT/beta-catenin signaling pathway has been reported to drive immune exclusion (i.e., cold tumors) across different cancers (Luke, J, et al. Clin Cancer Res. 2019). The authors should focus on exploring this potential role of TP63.
19. For your discussion, loss of p63 expression based on IHC was reported to be a biomarker for aggressive cancers (Stefan Stevrer, et al. Biomarker Res. 2021). How do you reconcile your conclusions with these observations?
20. The idea that TP63 may be driving immunologically cold tumors is not completely novel. Glathar, A.R., et al. Frontiers Oncol. 2022. have shown that the p63^{low} expression signature

was enriched with genes associated with the immune response in HPV+ tumors. Perhaps a sentence or two discussing novelties of this study is warranted.

21. The last sentence of the discussion states that early use of IFN-g with ICB may represent a potential treatment strategy for SCC patients with high expression of TP63. Systemic IFN γ therapies have been associated with toxicities, which is why they have not been pursued more aggressively as immunotherapeutic agents. Additionally, IFN γ drives upregulation of multiple checkpoints (e.g., PD-1 and PD-L1), which would hamper ICB therapy. Are there ways of enhancing IFN γ (or IFN α) locally, within tumors.

Minor critiques:

1. Legends are underdeveloped, making it sometimes difficult to understand what specimens are being analyzed. There is a lack of basic information (e.g., number of experiments performed, statistical analyses used, etc.).
2. Figures are hard to follow. Panel organization in many figures is not straightforward (e.g., Figure 5). Instead, figures jump all over the page. Font size is pretty small, making it hard to decipher certain figures.
3. Statistics are not always shown in figures. This needs to be addressed for each panel.
4. Occasional grammar issues.

Reviewer #2 (Remarks to the Author):

This study by Jiang et al. addresses the immunomodulatory abilities of TP63, a transcription factor known from previous reports to perform oncogene functions specifically in squamous cell carcinoma, and the key mechanisms associated with these abilities. In this study, the authors confirm that TP63 expression is specific and highly expressed in SCC cancers (specifically, ESCC, HNSC, and LUSC) in a TCGA-based comparison of various cancers. The authors identified a negative correlation between TP63 expression and the IFN signaling pathway and also confirmed that the IFN response pathway was highly enriched by TP63 knockdown in various SCC-derived cell lines. Furthermore, they found that 23 ISG genes in the IFN signaling pathway were closely associated with TP63 expression in ESCC, HNSC, and LUSC carcinomas, and confirmed that shTP63 knockdown increased the expression of these

ISG genes in human and mouse SCC cell lines. Based on the correlation between TP63 expression and IFN response, the authors identified increased CD8 T cell infiltration and activity into the TME by TP63 silencing in a syngeneic mouse SCC tumor model, leading to a reduction in tumor growth and an increase in the effectiveness of ICB treatment with anti-PD1. The inverse correlation between TP63 expression and CD8 T cell (especially cytotoxic effector/memory, exhausted CD8 T subsets) infiltration into the TME was confirmed in both public scRNA-seq data derived from SCC patients and actual ESCC patient cancer tissues.

Based on these findings, the authors hypothesized that TP63 negatively regulates the IFN signaling pathway and showed that, as a key mechanism, it transcriptionally inhibits the expression of STAT1, a key transcription factor in the IFN signaling pathway. Furthermore, the authors showed that STAT1 also transcriptionally represses TP63 expression. Indeed, the authors showed that phosphorylated STAT1 physically binds to TP63 after IFN-g treatment and that STAT1 can competitively interact with TP63 at the distal regulatory element (e8) of the TP63 gene to repress TP63 expression.

Overall, this study highlights a tumor-extrinsic role of TP63 in SCCs in negatively regulating the IFN-STAT signaling axis and provides a mechanistic insight into how cancers such as SCC can evade immune surveillance and attack by CD8 T cells and become resistant to ICB therapy.

The interesting and novel results of this study appear to be presented relatively convincingly, with a series of elegant and well-organized experiments and extensive RNA-seq data analysis. Nevertheless, this reviewer had some concerns and would like to strengthen the manuscript with a few points below.

Major concerns:

In the paper, the authors focused on the mechanism of negative regulation of the IFN-STAT1 signaling axis by TP63. Although the authors demonstrated the physical interaction of TP63 and STAT1 at the protein level as well as their shared occupancy at promoter and/or enhancer regions for reciprocal transcriptional regulation of each respective gene

expression using various approaches (Co-IP, Chip-Seq, luciferase assays, etc), it is still difficult to be certain that the observed changes in various immunological and functional aspects associated with different levels of TP63 expression are truly STAT1-dependent effects mediated by direct IFN signaling.

Therefore, in this reviewer's view, the most important but unanswered question in this study is to clarify whether the immunophenotypic and functional differences observed with differences in TP63 expression levels (CD8 T cell infiltration, activation, cytotoxic activity, tumor control, etc) are entirely dependent on physiological IFN exposure and concurrent activation of the key downstream transcription factor STAT1. To this end, the authors should clearly validate the direct physiological relevance of the IFN/STAT1 signaling axis in at least some key experiments, if not all experimental conditions presented in the study.

1) The authors need to demonstrate that the phenomena observed in the syngeneic mouse tumor model using shTP63-treated SCCs (i.e., the increased proportion of CD8 T cells and their enhanced activation features shown in Fig. 2A,D,E,H,I; and reduced tumor growths and increased CD8 T cell infiltration shown in Fig. 5G-I) actually decrease or disappear under IFN or IFNR blocking conditions.

2) In addition, the authors should be able to determine if shTP63-SCC and STAT1 overexpressing (OE)-SCC have similar effects in the mouse tumor models mentioned above (or if the phenomena observed with shTP63-SCC can be offset by introducing shSTAT1).

Other comments:

1) The authors analyzed TP63 gene expression in several different cancers using TCGA data and showed that TP63 expression is highest in SCCs (Figure S2A). In the same analysis, what about STAT1 gene expression? Did the authors observe a negative correlation between STAT1 and TP63 gene expression?

2) The authors observed that transduction of shTP63 into SCC cell lines (both TT and MOC22) resulted in increased expression of 23 ISG genes (Fig. 1E and S1F). This was

observed upon IFN γ treatment, but curiously, it was also observed without IFN γ treatment. If there are any internal or external factors that can promote the increase in ISG genes - which is likely dependent on the transcriptional activity of IFN-induced p-STAT1 - even in the absence of exogenous IFN γ , the authors need to clarify this.

3) Since the authors used a variety of SCC cell lines in this study, it would be helpful to interpret the data by checking to what extent TP63 and STAT1 expression differs between these cell lines at the protein level in at least the most commonly used cell lines (MOC22, AKR, HNM007), if not all, and if there is a negative correlation between these expressions.

4) In Fig. S4C and S4D, the authors separated the TP63^{high} and TP63^{low} (top/bottom 15%) groups based on public scRNA-seq data of 60 ESCC patient specimens, showing a clear difference in CD8 T cell frequency between them. Here, did the authors investigate CD8 T cell frequency in patients with median TP63 expression levels, and if so, did the values fall somewhere in between those observed in the TP63^{low} and TP63^{high} groups?

5) The authors observed in Figs. 5G and 5H that tumor growth was inhibited in the shTP63 group compared to the scramble group, and further showed that this inhibition was further amplified by anti-PD1 treatment. This reviewer has two questions here. First, the reduction in tumor growth by shTP63 was only observed in AKR and not in HNM007 (Scramble + IgG2a vs shTrp63 + IgG2a in Fig. 5G and 5H), did the authors check whether this is related to the difference in TP63 expression levels between these two cell lines? Second, did the authors observe the effect of CD8 T cell depletion in the absence of anti-PD1 treatment? In other words, did the authors check whether the reduction in AKR tumor growth observed in the shTP63 group (Scramble + IgG2a vs shTrp63 + IgG2a in Fig. 5G) was also CD8 T cell-dependent?

6) In Fig. 6, the authors described the physical interaction of TP63 and phospho-STAT1 (specifically on Ser727 but not on Tyr701) and the reciprocal transcriptional regulation between them. Although p-STAT1 (Ser727) can be induced in an IFN signaling-dependent manner together with p-STAT1 (Tyr701), it was shown that the former can also be formed by IFN-independent signaling by various stress-induced MAP kinases (ERK, P38, JNK, etc.). In

addition, even in the case of IFN-induced p-STAT1 (Ser727), previous studies have shown that it requires the transcriptional activity of p-STAT1 (Tyr701) (Sadzak et al., PNAS, 105:8944, 2008).

Therefore, the authors need to check TP63-p-STAT1(Ser727) or possible TP63-p-STAT1(Tyr701) interactions very early after IFN treatment (~0.5-1 hr), as the possibility of interaction between p-STAT1(Tyr701) and TP63, which is expected to appear and disappear quickly after IFN treatment, cannot be ruled out. Furthermore, rather than limiting these conditions to IFN γ , a type II IFN (mainly induce p-STAT1/STAT1 homodimer that binds GAS element), a cross-comparison with IFN α/β , a type I IFN (predominantly induce p-STAT1/STAT2 heterodimer that combines IRF9 and binds ISRE regions to induce ISG gene expressions) treatment will be very useful to clearly interpret and generalize the IFN/STAT-related mechanism of TP63 in this study.

Minor corrections:

In line 208-209, "~observed in TP63-low tumor samples (median=40.07%) than TP63-high tumors (median= 52.29%)" should be corrected to "~observed in TP63-low tumor samples (median=52.29%) than TP63-high tumors (median= 40.07%)"

Reviewer #3 (Remarks to the Author):

Jiang et al. report on a negative regulator of IFN γ -signalling via TP63, a transcription factor which is found to be ubiquitously overexpressed in several human squamous cell cancer types. The authors explored their own transcriptomics data and publicly available datasets to analyze the relationship between TP63 expression and IFN γ signalling in squamous tumors. They further used in vitro and in vivo experiments to characterize the co-regulation between TP63 and Stat1 expression, and the relevance of this crosstalk. Subcutaneous in vivo SCC murine models with TP63 knockdown by shRNA show increased CD8+ T cell infiltration, and the combination of knockdown plus checkpoint blockade in vivo appears to slow tumour growth. The manuscript is well-written, the coregulation between TP63 and STAT1 is novel, and the data is compelling for the most part. Below are some comments that

will improve the manuscript.

Main comments:

1- The overall observations would greatly benefit from addressing whether TP63 high tumours are a result of loss of IFN signalling, or vice versa, whereby TP63 overexpression results in desensitization of tumors towards IFNg. This is important for understanding whether TP63 expression is dysregulated, leading to resistance of IFNg is simply a biomarker/readout of the state of the immune response. The fact that the authors observe an inverse correlation between CD8 T cell proportion and TP63 expression (Fig.3E) suggests the latter. To answer this, the authors could explore whether TP63 high tumours are always associated with an overall immune cold phenotype or CD8 excluded phenotype. Similarly, are PD-1 unresponsive tumours TP63 high in humans? Can TP63 be used to stratify patient response to checkpoint blockade? At the very least, the discussion should be a little more nuanced and reflect the possibility that TP63 in human datasets might just be a marker of cold tumors.

2- The fact that TP63 downregulation in tumours is enough to observe an effect on cytotoxicity during in vitro co-culture of CD8 T cells and tumours is interesting. Providing some type of possible mechanism would greatly increase the value of the paper. For example, assessing the expression and relevance of MHC-I, PDL-1 or CXCL9/10 could help understand what is going on. In quite a few places, the authors investigate mRNA expression, but it is important to demonstrate that protein expression is also altered. In the same line, how is T cell priming affected by TP63 overexpression or downregulation?

3- The authors suggested that TP63 can directly inhibit STAT1 activity through binding of the transcription factor with phosphorylated protein (Figure 6). We suggest that authors show the experimental condition where although pSTAT1 can be detected, downstream signalling effects such as MHC-I/II upregulation are abrogated in the presence of TP63. In figure 6C, IP-pulldown blots are missing a total STAT1 control, and negative control such as a non-related, non-interacting protein which is non-nuclear. In Figure 6J, data is not obvious just by looking at the blot, it would be beneficial to quantify the bands and normalize to GAPDH band intensity.

4- Usually, it is required to treat T cells with Brefeldin A or Monensin to detect IFNg production by T cells. The authors don't seem to use anything like this, but this is important to reproduce that experiment with Brefeldin to confidently detect IFNg in their in vitro

system (Fig.5E,F).

5- Finally, to confirm the reciprocal relationship between STAT1 and TP63 interaction by knocking down STAT1 – do these cells have a corresponding increase in TP63?

Minor comments:

- In figure 2, why are Dox timing different between the scramble or TP63 shRNA?
- Figure 2I and 5E&F: Representative FACS plots should be shown in the supplementary to support bar graphs, especially for intratumoral cytokine-related stains and gating of positive populations for both tumors.
- The authors focused on CD8 T-cells based on Figure 2, but there are other changes that occur, especially a diminution in neutrophils and macrophage 3. The authors should elaborate on this.
- What is the expression of IFNGR1 and IFNGR2 in the cell lines used?
- In line 266, the authors don't know whether that's the infiltration or expansion that's increased in TP63 shRNA tumors.
- Figure 5A: schematic suggests that IFN γ was added to change antigen presentation capability of tumour cells but corresponding data is not shown.
- In figure 5J, the colors and legend below the graph don't match the graph itself.

Reviewer #4, (Remarks to the Author):

Reviewer #5 (Remarks to the Author):

Jiang et al applied scRNA-seq, immune profiling, and ChIP-seq to squamous cell carcinoma cell lines and mouse models to show that the SCC master regulator TP63 suppresses the interferon signaling through repressing STAT1 expression. They also showed that silencing of TP63 increased CD8+ T cell infiltration, which enhanced anti-tumor efficacy of immune

checkpoint blockade therapy. The experiments were designed and performed properly, and the manuscript was well written. The findings of the paper are of potential interest to the field of squamous cell carcinoma and oncogene-related immune suppression. That being said, I suggest the following experiments and analyses that in my view would improve the rigor of the work.

1. Given the potential off-targets effects of shRNAs, I would suggest performing rescue experiments for key findings of the manuscript, at least for the in vitro part. For instance, does ectopic expression of TP63 rescue the expression of STAT1 and IFN genes up-regulated upon TP63 silencing? And does expressing TP63 rescue the increased T-cell killing in the ex vivo co-culturing system?

2. The authors did a good job incorporating different datasets to support their conclusions. However, some of the key experiments were performed in only one or two cell lines. I would suggest adding STAT1 ChIP-seq in more lines, especially lines that already have TP63 ChIP-seq data. I would also suggest assessing the repressive role of TP63 on STAT1 expression in more cell lines. Also, although most of the in vitro experiments were done in TT cells, this cell line seems missing ChIP-seq of STAT1/TP63 data and luciferase reporter data.

3. The negative regulation of STAT1 on TP63 expression is interesting. However, IFN γ may trigger several other TFs in addition to STAT1. Knock-out or knock-down of STAT1 may provide more direct evidence regarding its role in TP63 regulation.

4. The antagonistic role of TP63 and STAT1 is of great interest. The authors should perform a more thorough analysis assessing how genes adjacent to TP63-unique, STAT1-unique, and shared sites respond to TP63 or STAT1 perturbations. Also, the authors should discuss potential mechanisms of how these two TFs act as repressors for each other.

Revised NCOMMS-23-28296-T by Yuan Jiang *et al.*

Title: Reciprocal inhibition between TP63 and STAT1 regulates anti-tumor immune response through interferon- γ signaling in squamous cancer

Dear Reviewers,

We are truly delighted to note that all Reviewers are interested in our work. We are also appreciative of the valuable comments raised by the Reviewers, which have helped improve further the quality of our work.

After careful and extensive experimentation, analysis and revision, we hereby provide our responses point by point (marked in blue font) and indicate where and how we have revised the manuscript taking into account all the critiques. All changes have been highlighted by underline in blue in the revised manuscript.

Thank you for your interest in and evaluation of our work.

REVIEWER COMMENTS

Reviewer #1 (Remarks to the Author):

Yuan Jiang et al. are evaluating whether TP63, the master transcription factor commonly expressed in squamous cell carcinomas (SCC), has immunosuppressive tumor-extrinsic properties that may play a role in immune checkpoint blockade (ICB) resistance. The authors explore whether and how TP63 suppresses IFN signaling, CD8+ T cell infiltration and tumor killing using a variety of in vivo and ex vivo systems. The authors conclude that TP63 and STAT1, mediator of IFN signaling, mutually suppress each other by co-occupying and co-regulating their own promoters and enhancers. Further, they state that TP63 upregulation negatively correlates with CD8+ T cell infiltration and activation, and that silencing of TP63 enhances the anti-tumor efficacy of anti-PD-1 treatments by promoting superior CD8+ T cell infiltration and functionality. While this study is exploring an important topic, data are underdeveloped. The following items need to be addressed:

Authors' reply:

We are very grateful for the Referee's positive comment that our study is exploring an important topic. We are also appreciative of your important critiques, addressing of which has helped improve further the quality and the strength of our study.

Major conceptual critiques:

1. The major problem of this study is that it does not clearly set the boundary between constitutive (cell-intrinsic) and exogenous IFN γ /IFN α -induced STAT1 signaling pathways. This blurring of lines between spontaneous and inducible gene signatures is an issue that affects the whole story, and an issue that needs to be carefully addressed in each figure.

Authors' reply:

We understand the Referee's concern and apologize for any conflation of exogenous and endogenous IFN signaling. To be clear, our study intends to focus on the exogenous IFN γ -induced STAT1 pathway due to the following considerations:

(1) Our data of Figures 1E and S1F in the original manuscript have shown much more significantly upregulated expression of ISGs with *exogenous* addition of IFN γ

relative to TP63-knockdown alone, suggesting strong synergistic effect of exogenous IFN γ and TP63-depletion on the stimulation of STAT1 signaling pathway (Figures 1E in the revised manuscript).

(2) The prominent and indispensable role of *exogenous* IFN γ signaling pathway has been well documented in both immune response and anti-tumor immunotherapy in multiple cancer types (Gao et al., 2016; Liao et al., 2019; Zaretsky et al., 2016).

As suggested, during the revision, we have carefully addressed this issue in each figure and corresponding paragraph (Figures 1D-1F, 5A-5F, 6F-6J, 7H, 7I, S1F, S6, S9 and S10) and now only include data on *exogenous* IFN γ signaling.

2. Lack of translation of *ex vivo* and *in vivo* models using human patient specimens (e.g., robustly evaluating correlation between TP63 expression and CD8 infiltrate; p-STAT1 vs TP63 staining in tumor tissues). If access to human tumor tissues is limited, greater reliance on publicly available bulk RNAseq and scRNAseq datasets should be utilized.

Authors' reply:

As suggested by the Referee, during the revision we performed correlational analysis using publicly available datasets from human tumor tissues across various cancer types. Clearly, multiple TME deconvolution tools, including TIMER, CIBERSORT, EPIC, QUANTISEQ and XCell, consistently revealed that CD8⁺ T cell infiltration was negatively correlated with the expression of TP63, while positively (as expected) with STAT1 in three types of SCC (ESCC, HNSC and LUSC) (**Figures L1A-L1B**). Moreover, the negative correlation between expression of TP63 and the abundance of CD8⁺ T cells was further verified using both ESCC scRNA-seq and TCGA bulk RNA-seq data from LUSC and HNSC patient samples (**Figure 3E** in our manuscript, which was pasted here only for review convenience). In addition, as a validation, we performed immunofluorescence (IF) staining using human tissues from an independent cohort of 10 ESCC patients (TP63-low patients, n=5 vs. TP63-high patients, n=5). Cells were counted under 20 \times field of view by selecting 5 fields randomly for each slide. The prominent accumulation of CD8⁺ T cells was observed in SCC tumors with low/no expression of TP63. In contrast, while TP63-high tumors had only a few scattered CD8⁺ T cells at the edge of cancer lesions (**Figures 4D-4E** in our

manuscript, which was pasted here only for review convenience).

To investigate the correlation between TP63 and STAT1 in tumor cells, we interrogated RNA-seq data of both TCGA and CCLE from various cancer types. Results from TCGA dataset did not reveal a significant negative correlation (data not shown) likely due to the confounding STAT1 signals from immune cells (which do not express TP63) in patient tumor samples. However, a significant anti-correlation between the expression of TP63 and STAT1 was found in the CCLE RNA-seq dataset, including ESCC and LUSC (**Figure L1C**).

In conclusion, these results suggest that the TP63 level is negatively correlated with the expression of STAT1 and CD8⁺ T cell infiltration in SCCs.

Figure L1. (A and B) Heatmap showing either negative or positive correlation between the expression of TP63 (A) or STAT1 (B) and the infiltration of CD8+ T and its subtype populations using different TME analysis tools, including TIMER, CIBERSORT, EPIC, QUANTISEQ and XCell in various types of cancer. For each cancer type, we obtained gene expression data from the TCGA project and download T cell fractions from TIMER2 websites (<http://timer.cistrome.org/>). Subsequently, we performed correlation analyses between T cell fractions and the expression of TP63 or STAT1. The significant correlations were labeled accordingly: *P < 0.05, **P < 0.01, ***P < 0.001, ****P < 0.0001. (C) Cancer types that show negative or positive correlation between the expression of TP63 and STAT1. For each cancer type, we obtained gene expression data of cancer cell lines from the CCLE project. Then we conducted correlation analyses between the expression level of TP63 and STAT1.

Figure 3E in original manuscript

Figure 4E in original manuscript

Figure 4D in original manuscript

3. Numerous figures/images seem to be representative examples of their data (multicolor IF microscopy and, especially, Western blots). The authors need to show robust statistical analyses, where they are evaluating multiple independent experiments before making definitive claims (e.g., for Western blots, they need densitometry analyses). In supplementary figures they could show individual representative images.

Authors' reply:

At least 3 independent experimental or biological replicates were performed for all of the experiments, and we apologize for not making this clear in the initial submission. Following the Reviewer's valid comment, we now have presented statistical analyses and individual replicates in each Figure, Supplementary Figures as well as the Source Data file.

Here are two examples:

(1) For the multicolor IF microscopy of Figures 4D-4E in revised original manuscript, each group contained 5 slides, each of which was from 5 individual ESCC patients. Quantification of Figure 4D was shown in Figure 4E (see the details on page 20 in this Response Letter). The 'n' number and cell count methods have been clearly described in the figure legend of Figure 4.

(2) For Western blot results such as Figure 6, we have shown one representative image in main figure (**Figure L2**), and provided another 2-3 independent replicates and statistical densitometry analyses in supplementary figures and the Source Data file (**Figures L2A-L2D** showing the replicates of Figures 6F and 6G in manuscript; **Figures L2E-L2H** showing the replicates of Figures 6I and 6J in original manuscript, respectively).

Figure L2. (A, C, E, G) Independent biological or experimental replicates of western blotting analysis showing levels of TP63, STAT1 and p-STAT1 in the indicated cells and conditions. (B, D, F, H) statistical densitometry analyses of (A, C, E, G). Data represent means \pm SD, $n \geq 3$. * $p < 0.05$, ** $p < 0.01$, *** $p < 0.001$. Figures L2A-L2D are related to Figures 6F and 6G, while Figures L2E-L2H are related to Figures 6I and 6J in the original manuscript.

Major experimental critiques (in order of appearance in the paper):

1. MOC22 data shown in Figure S1F is not showing significant differences in 23 IFN gene upregulation following TP63 knockdown. TT data looks more convincing, albeit the magnitude of expression is vastly different between these two cell lines. It would be beneficial if the authors could validate these findings with additional human and mouse cell lines. Additionally, there are no statistical analyses shown outside of standard error of mean.

Authors' reply:

Following the Referee's suggestion, we have now added the statistical results outside of standard error of mean for both TT and MOC22 cell lines in the revised manuscript (**Figure L3A and L3C**). Moreover, we examined the expression of 23 IFN signature genes in another two cell lines (human TE5 and murine MOC1 cell line) (**Figures L3B and L3D**). Consistently, in the presence of IFN γ , most of ISG genes were significantly elevated following the TP63 knockdown in these two new cell lines, validating our original results.

As mentioned earlier, this study has now been revised to focus on the exogenous IFN γ -STAT1 signaling, and thus we have now removed the non-IFN γ group and only retained the IFN γ treatment group in the revised manuscript, Figure 1E and Supplementary Figure 1F.

Figure L3. (A-D) qRT-PCR analysis showing relative mRNA levels of the 23 ISGs in human (TT, TE5) and murine SCC (MOC22 and MOC1) cells expressing non-targeting control (Scramble) or TP63-targeting shRNA (shTP63) pulsed with IFN γ (100 ng/mL) for 48 hours. Data represent means \pm SD, n=3. *p < 0.05, **p < 0.01, *** P < 0.001.

2. At least some of these observations in Figure S1F should be validated on a protein level.

Authors' reply:

Following the Referee's suggestion, during the revision we measured the protein levels of four ISGs (MHC-I, B2M, IRF1 and BATF2) by western blot assays using corresponding antibodies. First, verifying our original finding, in the presence of IFN γ treatment, TP63 knockdown significantly increased the protein levels of both total

STAT1 and p-STAT1. Importantly, conspicuous upregulation of the 4 ISG targets was detected (**Figure L4**), again validating our original results at the mRNA level.

Figure L4. Western blotting analysis showing the protein levels of TP63, STAT1, p-STAT1 and some ISGs in human (TT) and murine SCC (MOC22) cells expressing non-targeting control (Scramble) or TP63-targeting shRNA (shTP63) pulsed with IFN γ (100 ng/mL) for 48 hours.

3. Top panel of Figure S1F is mislabeled. It should be TT and not MOC22.

Authors' reply:

We apologize for the confusion which was not because of typo but likely because of the dense figure layout. Indeed, both upper and lower panels showed the results from MOC22 cell line; the difference was the administration of IFN γ in the lower but not upper panel (all the results were normalized to Scramble). Results from TT cells were shown in Figure 1E.

As mentioned earlier, since this study has now been revised to focus on the exogenous IFN γ -STAT1 signaling, we have now removed the non-IFN γ group and only retained the IFN γ treatment group in the revised manuscript, Figure 1E and Supplementary Figure S1F (see page 10 in this Response Letter).

4. In Figure 2, the authors are using a TP63 knockdown mouse tumor model. What is the time-kinetics of gene knockdown? Is TP63 expression suppressed for three weeks (during the duration of an in vivo experiment) or is it a transient knockdown? How quickly does TP63 expression bounce back and how does this affect your experiment outcome?

Authors' reply:

In Figure 2, we employed a tetracycline inducible shRNA system by using a Tet-pLKO-puro vector to stably silence Trp63 expression. Specifically, in this system, an shRNA targeting Trp63 was turned on only in the presence of the antibiotic tetracycline or its derivatives, such as doxycycline. In this experiment, after the tumor allografts reached $\sim 100 \text{ mm}^3$, we added and maintained doxycycline in drinking water. **Thus, the knockdown of *Trp63* was not transient, but was lasting during the entire process following the addition of doxycycline.** This inducible system is commonly employed to knockdown gene expression in various mouse tumor models, for example knocking-down *Emsy* in lung cancer mouse model (Marzio et al., 2022) and *G9a* in breast cancer mouse model (Tu et al., 2018).

To further address the Referee's query, during the revision we examined the expression of Trp63 during the entire inducible period. Indeed, the protein level of Trp63 was significantly decreased upon the addition of doxycycline and lasted to the

end of the *in vivo* experiment (20 days) (**Figures L5A-L4B**).

The details regarding the construction of tetracycline inducible *Trp63* knockdown system have been described in the section of Methods. Briefly, we firstly established stable shRNA-expressing SCC cell lines with either non-targeting shRNA (Scramble) or *Trp63* shRNA (sh*Trp63*)-containing Tet-pLKO-puro vector. The knockdown of *Trp63* in each cell line was induced by supplement of doxycycline and was verified by western blot analysis. These stable SCC cells were then subcutaneously injected into the flanks of C57BL/6J mice to establish syngeneic murine SCC tumor models. After 10 days, doxycycline (Dox, 2 mg/mL) was added and maintained in the drinking water to silence the expression of *Trp63*.

Figure L5. (A) Schematic graph of the *in vivo* syngeneic experiments. MOC22 cells expressing shRNA were subcutaneously injected into the flanks of C57BL/6J mice to allow the formation of tumor allografts. After 10 days, doxycycline (Dox, 2 mg/mL) were added in drinking water to silence the expression of target shRNA. (B) Western blotting analysis showing the protein levels of TP63 in MOC22 tumors across the whole inducible process of doxycycline.

5. The total number of TIL analyzed by scRNAseq (1,911) is very low and indicates that likely only one mouse was evaluated per test group (scrambled vs. TP63 siRNA). This is not specified in the figure legend but is inferred. Additional experiments are needed to confirm these observations.

Authors' reply:

Indeed, we agree that the total number of TILs analyzed by scRNA-seq is low. However, in this experiment, each group included 4 tumors from 4 mice, which were combined and dissociated into single cells capture. Due to budget constraint at the time of this experiment, we did not capture large number of single cells for sequencing.

Nevertheless, despite the low number of TIL cells, both the increased proportion and activity of CD8⁺ T cells were detected in *Trp63*-knockdown tumors relative to scrambled control (corresponding to **Figures 2D, 2E and 2H** in the manuscript).

To validate these observations, we performed orthogonal flow cytometry analyses on either *Trp63*-knockdown or scrambled mouse tumors. Consistently, both increased proportion and activation of CD8⁺ T cells were detected in the sh*Trp63* group compared with the scramble group in HNM007 tumor samples, as measured by CD69, GZMB, IFN γ . In a different syngeneic model (AKR), we also observed increased proportion of activated CD8⁺ T population (CD69 and IFN γ) in *Trp63*-knockdown tumors relative to those in scramble group (**Figure 2I** in the manuscript).

Moreover, the above findings were also confirmed in human samples by re-analyzing scRNA-seq data (GSE160269) from 60 ESCC tumors. Firstly, among major immune subsets, the expression of TP63 was specifically anti-correlated with the T cell fraction (**Figure 3B** in our manuscript). Secondly, a higher proportion of CD8⁺ T cells as well as activated CD8⁺ T cells were observed in *TP63*-low tumor samples than *TP63*-high tumors (**Figures 3C, 3D S4D and S5** in our manuscript). Moreover, expression of *TP63* was negatively correlated with the cellular proportion of CD8⁺ T cells, as analyzed using both ESCC scRNA-seq and TCGA bulk RNA-seq data from LUSC and HNSC patient samples (**Figure 3E** in our manuscript).

6. What genes were used to identify different CD8⁺ T cell activation states? Also, what are CD8 T cells labeled as “Others?” Nomenclature used is a bit outdated and vague. For example, GZMK⁺ CD8 effector memory T cells have been associated with favorable response to anti-PD-1 therapy (Caushi, J.X. et al. Nature. 2021).

Authors’ reply:

As we have shown in Figure S2G, the mean expression of activation markers (*Pdcd1*, *Ctla4*, *Lag3*, *Havcr2/Tim-3*, *Ifng*, *Tnfrsf9*, *Cd69*), cytotoxic markers (*Gzmb*, *Gzma* and *Prf1*) and IFN signature genes (*Isg15*, *Isg20* and *Ifit1*) were used to define CD8⁺ T cell activation states based on curated gene signatures from recent publications (Andreatta et al., 2021; Daniel et al., 2022; Dong et al., 2021; Hao et al., 2021; Zhang et al., 2020a). As expected, we detected high expression of cytotoxic markers (*Gzmb*, *Gzma* and *Prf1*) in both exhausted and effector memory CD8⁺ T subsets. Between these

two subsets, the effector population (CD8_Tem) showed higher activation features of *Cd69*, *Ifng*, *Fos*, *Jun* and chemokines *Ccl3* and *Ccl4*, while the exhausted cluster (CD8_Tex) was characterized by the higher expression of exhausted markers including *Pdcd1* and *Ctla4* (**Figures L6A-L6B**, corresponding to Figures 2G, S2G and S3A in our manuscript).

The Reviewer mentioned that GZMK⁺ CD8 effector memory T cells have been associated with favorable response to anti-PD-1 therapy in non-small cell lung cancer (Caushi et al., 2021). Consistently, our study showed that TP63 knockdown led to the increased proportion of *Ccl4*⁺ effector memory T cells expressing *Gzmk*. Moreover, in human ESCC samples, we also observed increased proportion of GZMK population in TP63-low tumors relative to TP63-high counterparts (see **Figures L8B-L8D** on page 19 in Response Letter). The potential regulation of GZMK⁺ CD8⁺ effector memory T cells by TP63 warrants further investigations.

Regarding the “other” subgroup of CD8⁺ T cells, they had high expression of MHCII molecules, including *H2-Ab1*, *H2-Aa*, *H2-Eb1* and *Cd74*, which were found by (Caushi et al., 2021) and (Holling et al., 2004). According to the nomenclature used in these two reports, we have now annotated this group as “CD8_MHCII”.

Finally, as suggested by the Referee, we have now updated the nomenclature, and defined the subsets of CD8 populations as exhausted (CD8_Tex), effector memory (CD8_Tem), naïve (Tn) and CD8_MHCII in the revised manuscript (Page 8, paragraph 2) (Figures S2G and S3A).

Figure L6. (A) Dot plot showing the expression levels of representative marker genes across each CD8⁺ T cell subgroup. (B) UMAP visualization of CD8⁺ T cells with color-coded for the expression levels of representative marker genes to identify subsets.

7. Figure 2I lacks the gating strategy and staining examples. Also, were these cells analyzed after stimulation or without?

Authors' reply:

These cells were derived directly from *in vivo* allografts of syngeneic mouse models from either control or Trp63-knockdown tumors. The staining and gating strategies are shown below (**Figure L7**) and have now been added as Supplementary Figure 3B in our revised manuscript.

Figure L7. The gating strategy and a representative staining example for immune profiling of murine SCC tumors. AKR and HNM007-allograft tumors were collected after 7 days induction of Doxycycline. Tumors were dissociated into single cell suspension; immune cells were enriched using CD45 MicroBeads. Single cells were first gated and followed by Fixable Viability Stain 450 selection for live cells. Immune cell populations were then identified by the sequential gating strategy using specific markers: total immune cells (CD45⁺), total T cells (CD3⁺), CD8⁺ T cells. In addition, the expression of activation and cytotoxic protein markers (CD69, GZMB, IFN γ) in CD8 T cells was analyzed.

8. Figure 3 needs a citation of cell annotations (or show the genes used to identify them).

Authors' reply:

As suggested, the paper of Zhang, et al. (Zhang et al., 2021)²⁰ has been cited in the main text (Page 9, paragraph 3) and the legend of Figure 3 (Page 32, paragraph 2) in the revised manuscript.

9. Interpretation (or wording) of Figure 3B is incorrect. It is stated that “the abundance of T cells” negatively correlated with TP63 expression levels. Due to the nature of this dataset, CD45⁺ and CD45⁻ cells were enriched prior to sequencing. Consequently, we do not know the abundance of infiltrate, but do know the frequency/representation of total T cells. The authors need to validate these claims by using IHC or mIF staining experiments.

Authors' reply:

Yes, the Referee is correct that CD45⁺ and CD45⁻ cells were enriched prior to sequencing. Following your comment, we have now revised the description to “Among major immune subsets, the expression of TP63 was specifically anti-correlated with T cell fraction”.

Indeed, this anti-correlation was validated using IF staining experiments (**Figures 4D-4E** in our manuscript).

10. In Figures 4A -> 4B, the authors are relying on tSNE figures to infer CD8 T cell activity in TP63^{low} and TP63^{high} tumors. This should be done in a statistical way. Also, these data optimally need to be validated with suitable in vitro killing assays.

Authors' reply:

To address this, we performed statistical analysis of the expression of CD8 markers shown in Figures 4A-4B. We confirmed the increased proportion of CD8, PRF, GZMK and GZMB populations in TP63^{-low} tumors relative to TP63^{-high} counterparts (**Figures L8A-L8D**). As a validation, we observed strongly augmented tumor cell killing by CD8⁺ T cells upon downregulation of TP63 in both MOC22 and AKR cells, as revealed by *in vitro* tumor killing assays (**Figure L8E**, corresponding to Figures 5C-5D of in our manuscript). Consistently, in this co-culture system, markedly increased proportions of GZMB⁺ and IFN γ ⁺ cells were detected in *Trp63*-knockdown group in

comparison with those in the scramble group (**Figures L8F-L8G**, corresponding to **Figures 5E-5F** of in our manuscript). These results confirm that downregulation of TP63 promotes the antigen-dependent cytotoxicity of CD8⁺ T cells against SCC cells. As suggested by the Reviewer, the statistical analysis has been added in the revised manuscript (**Figure S5D**).

Figure L8. (A and B) t-SNE plots showing the expression levels of CD8 (A) and cytotoxic marker genes (B) of CD8⁺ T cells. Cells with the expression level of cytotoxic marker genes (TPM) high than 1 was labeled. (C) Bar plots showing the proportion of cytotoxic lymphocytes that express cytotoxic and activation markers including GZMB, GZMK, PRF1 and IFN γ in TP63-high/low expressing ESCC tumors. (D) Box plots showing the statistical analysis of (C) according to the cell

proportion of T cells with the expression of indicated marker genes in TP63-low/high expressing patient tumor samples. For each gene of interest, we defined target gene-positive cells if the corresponding TPM ≥ 1 . Subsequently, we calculated the fraction of target gene-positive T cells in TP63-low and TP63-high tumors. One-tailed Wilcoxon test was used to compare these two groups. *P < 0.05, **P < 0.01. (E) Crystal violet staining (left) of SCC cells incubated with or without OT-I CD8⁺ T cells at the indicated effector: target (E: T) ratios. Right panel: Bright field images showing representative co-cultured MOC22 and OT-I CD8⁺ T cells. Red arrows indicate a cancer cell killed by activated OT-I CD8⁺ T cells following 48 hr co-culture. (F and G) Percent of GZMB⁺ cells in the above Scramble or shTrp63 MOC22 (F) and AKR (G) co-cultures.

11. What is Figure 4C showing? Are dots representing individual patients?

Authors' reply:

Figure 4C shows that in SCC patient tumors, the TP63 level was negatively correlated with the expression of cytotoxic marker genes including GZMB, GZMK and PRF1, and positively correlated with SOX2 (as a positive control). Yes, each dot represents each individual patient sample. The number of patients was labeled under the dot plots (data from TCGA).

12. In Figure 4D and 4E, the authors look like they are showing representative images of multicolor IF. They need to summarize and present the data in a statistically relevant way. Also, how are they choosing TP63 low and high samples? They need to have at least 4-6 patients in each cohort. How are tumor sections chosen? How were tumor areas analyzed chosen? Are data normalized somehow? For example, are they only looking at tumor beds or tumor stroma?

Authors' reply:

Figure 4D shows the representative images of H&E and IF staining of TP63 and CD8. Figure 4E shows the statistical quantification of infiltrated CD8 T cells from an independent cohort of 10 ESCC patients with either TP63-high (n=5) or -low (n=5) expression. Specifically, one slide per patient was used for analysis. IF images were acquired at the same exposure time. The staining results were scored by 2 different researchers according to the fluorescence intensity of TP63; five cases with highest

TP63 scores and five with the lowest scores were defined based on the median score. Five fields representing tumor regions were randomly selected for each slide, and the number of CD8⁺ T cell was counted under 20× field of view which was then averaged across 5 fields. Statistical analysis was performed by comparing the average CD8⁺ T cell number of each slide from five TP63-high vs. five TP63-low patients. The details have been added into the legend of Figure 4.

13. For Figure 5, the authors do not address a critical question. Does PT63 overexpression drive MHC class I downregulation? Or, does PT63 upregulation increase PD-L1 expression on tumors? Or is there a different mechanism by which this transcription factor hampers CD8 T cell-mediated killing of tumors? These questions need to be addressed.

Authors' reply:

To address the Referee's query, during the revision we first performed correlation analysis between the expression of TP63 and MHC (human HLA) and CD274 (PD-L1) in SCC patient samples, including ESCC, HNSC and LUSC. Notably, the TP63 level was negatively correlated with the expression of HLA-A, HLA-B and HLA-C (except in ESCC), as well as the MHC-I component-B2M (**Figure L9**). There was weak or no correlation between the expression of TP63 and CD274 in these patient samples (**Figure L9**).

Next, we analyzed their correlation in CD45⁻ populations using scRNA-seq data from 60 ESCC patients. In line with the above results, a significantly negative correlation between TP63 and HLA/B2M, but not CD274, was found (**Figure L10A**). More importantly, our new western blot assays revealed increased protein levels of MHC-I and B2M upon downregulation of TP63 under stimulation with IFN γ (**Figure L10B**). These additional data and our original findings together suggest that TP63 hampers CD8 T cell-mediated tumor killing likely through suppressing MHC-I and B2M, key downstream targets of the IFN γ -STAT1 signaling (**Figure L10C**).

Figure L9. Correlation analysis between the expression of TP63 and HLA, B2M and CD274 in ESCC, HNSC and LUSC. The immune cell fractions were predicted by TIMER2 using TCGA expression datasets. Pearson Correlation Coefficient was calculated.

Figure L10. (A) Correlation analysis between the expression of TP63 and HLA, B2M and CD274 in CD45 negative cells of 60 ESCC patient tumors. Data from scRNA-seq dataset (GSE160269). Pearson Correlation Coefficient was calculated. (B) Western blotting analysis showing the protein levels of TP63, MHCI, B2M, with GAPDH as a positive control in human (TT) and murine (MOC22) SCC cells expressing either non-targeting control (Scramble) or TP63-targeting shRNA

(shTP63) pulsed with IFN γ (100 ng/mL) for 48 hours. (C) A work model of TP63 suppressed IFN γ -STAT1 immune regulatory pathway in SCC tumor microenvironment.

14. Figures 5J and 5K are showing the same data, except one shows average and other individual tumor growth curves. What about HNM007? These data should be added.

Authors' reply:

The Reviewer is correct that Figures 5J and 5K respectively shows statistical average values and individual tumor growth curves, two complementary presentations of the same data.

As requested by the Referee, we performed additional mouse model experiments to evaluate the functional contribution of CD8 T cells using the HNM007 model. Importantly, consistent with the AKR tumors, the anti-tumor effect exerted by shTrp63 and PD-1 mAb treatment was markedly reversed by the depletion of CD8 T cells with CD8 blocking mAb (**Figures L11A-L11C**). These additional data and our original findings together suggest that inhibiting TP63 synergistically promotes the anti-tumor effect of PD-1 mAb by enhancing CD8 T cell infiltration and activity. As suggested by the Reviewer, the new results from HNM007 tumors have been added into the revised manuscript (Figure 5K).

Figure L11. (A-C) Tumor growth curves (A), Tumor weights at completion of the study (B) and Body weights (C) of shTrp63 HNM007-bearing mice treated with PD-1 mAb in combination with CD8 mAb (CD8 T-cell-depletion antibody) (n=5) or IgG isotype control (IgG2b) (n=5). All the mAbs were given by intraperitoneal injection (i.p., 100 µg/injection/mouse) once every 3 days for up to 2 weeks. CD8 mAb was conducted beginning on the 5th day of the transplantation of tumor cells. The significance was determined by *t* test. ** P < 0.01, *** P < 0.001.

15. Description of Figure 6 suggests that TP63 knockdowns lead to constitutive STAT1 phosphorylation. That is not visible. The effect of TP63 knockdown is unclear as it is measured by concurrent IFN γ treatment. These data suggest synergistic action of TP63 knockdown and IFN γ treatment.

Authors' reply:

Knockdown of TP63 increased the mRNA expression of constitutive STAT1, which was shown in Figure 6E. However, the Reviewer is correct that we did not examine the constitutive STAT1 phosphorylation due to extremely low level of p-STAT1 without stimulation of IFN γ (See the next point). As mentioned earlier, this manuscript has now been revised to focus on the exogenous IFN γ -STAT1 signaling, and we agree with the Reviewer's interpretation that these results suggest the synergistic action of TP63-knockdown and IFN γ treatment. We have revised the description of Figure 6 accordingly as below (Page 13, paragraph 1):

“Notably, TP63 knockdown and IFN γ stimulation elevated both mRNA and protein levels of STAT1 as well as p-STAT1 (Figures 6E, 6F and S9C).”

16. In Figure 6G, the authors need an experiment where cells are either untreated or treated with IFN γ over the same time course. This is needed to clearly distinguish between TP63 and IFN γ -driven effects.

Authors' reply:

As requested by the Referee, we performed western blot assays during the revision in the absence of IFN γ treatment over the same time course as in original Figure 6G (Figure L12A; Note the Figure L12A is the same as Figure 6G, which was pasted here for review convenience). We found that ectopic expression of TP63 reduced slightly

the protein level of total STAT1 over 48 hr, but the phosphorylation level of STAT1 was extremely low without stimulation of IFN γ (**Figure L12B**). Indeed, IFN γ treatment induced markedly expression of both STAT1 and p-STAT1, as expected (**Figure L12A**). Importantly, IFN γ -induced levels of both STAT1 and p-STAT1 were diminished by ectopic expression of TP63 (**Figure L12A**), confirming our original finding that TP63 suppresses IFN γ -induced STAT1 signaling.

Figure L12. (A-B) Western blotting analysis showing levels of indicated proteins at different exposure time. SCC cells (KYSE410 and TE1) with low expression of TP63 were transfected with TP63 over-expression (OE) or empty vector (Control) following IFN γ (100 ng/mL) treatment (A) or without IFN γ treatment (B). Exposure time were denoted at right.

17. In Figure 6J, effect of IFN γ inhibitor on TP63 expression is ambiguous. Data needs to be normalized for GAPDH and multiple experiments performed and plotted.

Authors' reply:

As we have responded to the Reviewer's 3rd point earlier (refer to page 8 in this Response Letter), two independent biological and three experimental replicates in two cell lines were performed for Figure 6J. The results confirmed consistently reduced levels of STAT1 and p-STAT1 treated with STAT1-specific inhibitor, Fludarabine, in both TT and TE5 cells. Importantly, STAT1-inhibition by Fludarabine rescued TP63 levels decreased by IFN γ in all experiments (**Figures L2G-L2H**). We now have shown both statistical analyses and individual replicates in the revised manuscript.

18. One of the more interesting pieces of data is buried in Figure S7H. In this figure, gene-set enrichment analysis is performed on TP63 unique peaks. Here, the authors show that TP63 may be regulating the WNT signaling pathway. WNT/beta-catenin signaling pathway has been reported to drive immune exclusion (i.e., cold tumors) across different cancers (Luke, J, et al. Clin Cancer Res. 2019). The authors should focus on exploring this potential role of TP63.

Authors' reply:

We thank the Referee for identifying this interesting point. Actually, a few years ago, our group reported that TP63 induced the expression of β -catenin in SCC cells (Jiang et al., 2018), which was first seen in head and neck squamous cancer by Patturajan, et al. (Patturajan et al., 2002) (**Figures L13A- L13B**).

To pursue this further, in another two SCC cell lines, we performed western blotting experiments and also found reduced protein levels of β -catenin upon TP63 knockdown in the presence of IFN γ (**Figure L13C**). Interestingly, we observed the occupancy of TP63 at the *cis*-regulatory element-enhancer of β -catenin gene (*CTNNB1*) (**Figure L13D**). As the Referee mentioned, activation of WNT/ β -catenin signaling pathway is associated with immune exclusion (i.e., cold tumors) (Luke et al., 2019). We thus hypothesize that in SCC, β -catenin-mediated immune exclusion might partially underly the immune regulatory function of TP63, which warrants future investigations.

Figure L13. (A) Upper panel: Western blotting assays showing protein levels of TP63 in nuclear and cytosolic fractions, with DNA topoisomerase II α (TOPO II, nuclei) or α -tubulin (cytoplasm) as fractionation loading controls. Nuclear and cytoplasmic fractions were isolated from HEK-293 cells transfected (~60% efficiency) with 2 μ g empty pCEP4 vector (- Δ Np63 α) or pCEP4- Δ Np63 α (+ Δ Np63 α). Lower panel: Nuclear colocalization of Δ Np63 α and β -catenin. HNSCC 013 cells were stained with antibody to β -catenin (green) or antibody to Δ Np63 α (red) and were counterstained with DAPI for nuclear DNA (blue). Merging of the two stainings shows colocalization of the two proteins (Δ Np63 α and β -catenin) in the nucleus (yellow). (B and C) Western blotting analysis showing levels of TP63 and β -catenin in SCC cells (TE5, KYSE140, TT, MOC22) following IFN γ (100 ng/mL) treatment (B) or not (C). (D) Integrative genomic viewer (IGV) tracks of ChIP-seq and ATAC-seq revealing the occupancy of TP63, RNAPII, H3K4me1, H3K27ac and H3K4me3 as

well as chromatin accessibility on the promoter/enhancer loci of β -catenin (*CTNNB1*) gene. ChIP-seq and ATAC-seq data were retrieved from GSE46837, GSE106563 and GSE148920. Figure L13A and L13B were published figures from Patturajan, et al. 2002 and Jiang, et al. Nat Commun. 2018, respectively.

19. For your discussion, loss of p63 expression based on IHC was reported to be a biomarker for aggressive cancers (Stefan Stevrer, et al. Biomarker Res. 2021). How do you reconcile your conclusions with these observations?

Authors' reply:

Indeed, distinct molecular functions of TP63 have been reported in different human cancers. Supporting our results, upregulation and oncogenic property of TP63 has been frequently reported in various types of squamous cell carcinomas (SCCs) (Cancer Genome Atlas, 2015; Cancer Genome Atlas Research et al., 2017). In addition, high expression of TP63 has been associated with low responsiveness to single PD-1 mAb therapy in a phase I clinical trial of ICB therapy in ESCC patients (NCT02742935) (Liu et al., 2023).

TP63 is a transcription factor of the p53 gene family, regulating the activity of a multitude of genes involved in growth and development of the ectoderm and derived structures and tissues. TP63 has two isoforms-the transactivation domain (TAp63) and amino-deleted TP63 isoform (Δ Np63), which exert different (often opposite) functions on stemness, cycle arrest, mobility and invasion (epithelial-mesenchymal transition, EMT) and senescence (Melino, 2011). While full length TAp63 activates p53 target genes such as p21 or BAX, the shorter transcript Δ Np63 exerts oncogenic properties and is generally over-expressed in cancer, such as SCC, which is our focus.

In the published paper referred to by the Referee (Stevrer, et al. Biomarker Res. 2021), the authors used an in-house antibody of TP63 (clone 7B4) without specifying the epitope. As mentioned above, TAp63 is known to suppress cancer metastasis by decreasing cell mobility and invasion. Thus, we speculate that loss of the TA isoform (TAp63) might be correlated with aggressive urothelial carcinomas (Stevrer et al., 2021).

As suggested, the following comments have been added to the discussion section in the revised manuscript (**Page 16, paragraph 2**):

“However, TP63 has two isoforms (TAp63 and Δ Np63), which often exert opposite functions in cancer biology^{53,54}. Therefore, for biomarker studies, it is imperative to discern the relative expression of TAp63 and Δ Np63.”

20. The idea that TP63 may be driving immunologically cold tumors is not completely novel. Glathar, A.R., et al. *Frontiers Oncol.* 2022. have shown that the p63^{low} expression signature was enriched with genes associated with the immune response in HPV⁺ tumors. Perhaps a sentence or two discussing novelties of this study is warranted.

Authors’ reply:

Thank you for your comment. Accordingly, we carefully read this paper published by Glathar et al. (Glathar et al., 2022), which predominantly investigated the oncogenic role of p63 in regulating HPV⁺ HNSCC carcinogenesis by driving subtype-specific gene expression partially through PI3K and mTOR signaling pathways. There was only one piece of data in this paper linking TP63 to immune pathways, which was a computational GO enrichment analysis based on differentially expressed genes. This enrichment analysis revealed a negative correlation between the expression of p63 and inflammatory/immune response-associated genes in the HPV⁺ HNSCC samples from published datasets. However, no functional experiments were performed to pursue this negative correlation.

As suggested by the Referee, the following discussion on the novelties of our study has been incorporated into the revised manuscript (**Page 15, paragraph 2**):

“A negative correlation between the expression of TP63 and inflammatory/immune response-associated genes in HPV⁺ HNSCC tumors was recently revealed using computational GO enrichment analysis⁴³. Here we reveal a novel tumor-extrinsic role of TP63 in promoting immune evasion of SCC cells by suppressing the IFN γ -STAT1 signaling, which has hitherto not been appreciated.”

21. The last sentence of the discussion stats that early use of IFN-g with ICB may represent a potential treatment strategy for SCC patients with high expression of TP63. Systemic IFN γ therapies have been associated with toxicities, which is why they have not been pursued more aggressively as immunotherapeutic agents. Additionally, IFN γ

drives upregulation of multiple checkpoints (e.g., PD-1 and PD-L1), which would hamper ICB therapy. Are there ways of enhancing IFN γ (or IFN α) locally, within tumors.

Authors' reply:

Yes, several approaches to locally enhance intratumoural IFN γ (or IFN α) have been reported, including direct injection of IFNs, nonreplicating adenoviral vector-encoding IFN γ expression and engineered intratumorally retained IFNs. For example, combination of intratumoral injections of mouse IFN α and intraperitoneal injections of anti-CD137 mAb showed a synergistic effect in the MC38 colon carcinoma model (Dubrot et al., 2011). Intratumoral application of adenovirus-IFN γ (TG1042) has been tested in a phase II clinical trial of patients with advanced cutaneous T-cell lymphomas and cutaneous B-cell lymphomas. This trial resulted in local tumor regression in about half of treated patients (Dummer et al., 2010). In addition, alum (aluminum-hydroxide)-anchored intratumoral retention of IFNs greatly improved the tolerability and efficacy of both IFN α and IFN β (Lutz et al., 2022). Therefore, locally enhancing intratumoural IFN is a potentially promising strategy.

Minor critiques:

1. Legends are underdeveloped, making it sometimes difficult to understand what specimens are being analyzed. There is a lack of basic information (e.g., number of experiments performed, statistical analyses used, etc.).

Authors' reply:

Thanks for Referee's critique. During the revision, we have added the necessary basic information on each figure legend.

2. Figures are hard to follow. Panel organization in many figures is not straightforward (e.g., Figure 5). Instead, figures jump all over the page. Font size is pretty small, making it hard to decipher certain figures.

Authors' reply:

We apologize for the organization of figure presentation. We have now re-organized the panels in the revised manuscript.

3. Statistics are not always shown in figures. This needs to be addressed for each panel.

Authors' reply:

As suggested by the Referee, we have carefully addressed this issue for each figure panel.

4. Occasional grammar issues.

Authors' reply:

The revised manuscript has been edited throughout by two additional English-speaking scientists.

Reviewer #2 (Remarks to the Author):

This study by Jiang et al. addresses the immunomodulatory abilities of TP63, a transcription factor known from previous reports to perform oncogene functions specifically in squamous cell carcinoma, and the key mechanisms associated with these abilities. In this study, the authors confirm that TP63 expression is specific and highly expressed in SCC cancers (specifically, ESCC, HNSC, and LUSC) in a TCGA-based comparison of various cancers. The authors identified a negative correlation between TP63 expression and the IFN signaling pathway and also confirmed that the IFN response pathway was highly enriched by TP63 knockdown in various SCC-derived cell lines. Furthermore, they found that 23 ISG genes in the IFN signaling pathway were closely associated with TP63 expression in ESCC, HNSC, and LUSC carcinomas, and confirmed that shTP63 knockdown increased the expression of these ISG genes in human and mouse SCC cell lines. Based on the correlation between TP63 expression and IFN response, the authors identified increased CD8 T cell infiltration and activity into the TME by TP63 silencing in a syngeneic mouse SCC tumor model, leading to a reduction in tumor growth and an increase in the effectiveness of ICB treatment with anti-PD1. The inverse correlation between TP63 expression and CD8 T cell (especially cytotoxic effector/memory, exhausted CD8 T subsets) infiltration into the TME was confirmed in both public scRNA-seq data derived from SCC patients and actual ESCC patient cancer tissues.

Based on these findings, the authors hypothesized that TP63 negatively regulates the IFN signaling pathway and showed that, as a key mechanism, it transcriptionally inhibits the expression of STAT1, a key transcription factor in the IFN signaling pathway. Furthermore, the authors showed that STAT1 also transcriptionally represses TP63 expression. Indeed, the authors showed that phosphorylated STAT1 physically binds to TP63 after IFN- γ treatment and that STAT1 can competitively interact with TP63 at the distal regulatory element (e8) of the TP63 gene to repress TP63 expression.

Overall, this study highlights a tumor-extrinsic role of TP63 in SCCs in negatively regulating the IFN-STAT signaling axis and provides a mechanistic insight into how cancers such as SCC can evade immune surveillance and attack by CD8 T cells and become resistant to ICB therapy.

The interesting and novel results of this study appear to be presented relatively convincingly, with a series of elegant and well-organized experiments and extensive

RNA-seq data analysis. Nevertheless, this reviewer had some concerns and would like to strengthen the manuscript with a few points below.

Authors' reply:

We truly appreciate the Referee's positive comments! We are also grateful for your important critiques, addressing of which has helped strengthen further our study.

Major concerns:

In the paper, the authors focused on the mechanism of negative regulation of the IFN-STAT1 signaling axis by TP63. Although the authors demonstrated the physical interaction of TP63 and STAT1 at the protein level as well as their shared occupancy at promoter and/or enhancer regions for reciprocal transcriptional regulation of each respective gene expression using various approaches (Co-IP, Chip-Seq, luciferase assays, etc), it is still difficult to be certain that the observed changes in various immunological and functional aspects associated with different levels of TP63 expression are truly STAT1-dependent effects mediated by direct IFN signaling.

Therefore, in this reviewer's view, the most important but unanswered question in this study is to clarify whether the immunophenotypic and functional differences observed with differences in TP63 expression levels (CD8 T cell infiltration, activation, cytotoxic activity, tumor control, etc) are entirely dependent on physiological IFN exposure and concurrent activation of the key downstream transcription factor STAT1. To this end, the authors should clearly validate the direct physiological relevance of the IFN/STAT1 signaling axis in at least some key experiments, if not all experimental conditions presented in the study.

1) The authors need to demonstrate that the phenomena observed in the syngeneic mouse tumor model using shTP63-treated SCCs (i.e., the increased proportion of CD8 T cells and their enhanced activation features shown in Fig. 2A,D,E,H,I; and reduced tumor growths and increased CD8 T cell infiltration shown in Fig. 5G-I) actually decrease or disappear under IFN or IFNR blocking conditions.

Authors' reply:

Thanks for the Referee's valuable suggestion. During the revision, we performed additional mouse model experiments using shTrp63 SCCs, which were treated with

either IFN γ blocking antibody or the IgG isotype control antibody, as suggested by the Reviewer. Under IFN γ blocking condition, tumors continued to grow and the tumor weight was significantly higher in comparison with that of the control group treated with IgG isotype antibody (**Figures L14A-L14C**). These results demonstrate that the reduced tumor growth by shTP63 in the syngeneic mouse tumor model was mediated at least partially by the IFN γ signaling. The new generated data has been incorporated into our revised manuscript as **Figure S7D**.

Figure L14. (A-C) Tumor growth curves (A), Tumor weights at completion of the study (B) and Body weights (C) of shTrp63 HNM007-bearing mice treated with IFN γ blocking antibody (n=5) or IgG isotype control (IgG1) (n=5). mAbs were given by intraperitoneal injection (i.p., 400 μ g/injection/mouse) once every 2 days for up to 2 weeks. The significance was determined by *t* test. * $P < 0.05$.

2) In addition, the authors should be able to determine if shTP63-SCC and STAT1 overexpressing (OE)-SCC have similar effects in the mouse tumor models mentioned above (or if the phenomena observed with shTP63-SCC can be offset by introducing shSTAT1).

Authors' reply:

As suggested by the Referee, during the revision, we performed additional mouse model experiments using either empty vector control or STAT1-overexpressing (OE) SCC cells. Overexpression of STAT1 in HNM007 cells was verified using qRT-PCR and western blotting assays and then used for *in vivo* experiments (**Figures L15A-L15B**). Importantly, consistent with the results from TP63-knockdown mouse model, STAT1-overexpression significantly decreased tumor growth and tumor weight relative to those in the control group without affecting the body weight of mice (**Figures L15C-L15E**). These new data have been incorporated into our revised manuscript as **Figures S10G-S10J**.

Figure L15. (A and B) qRT-PCR (A) and western blotting (B) analysis showing mRNA and protein levels of STAT1 in STAT1-overexpressing HNM007 cells. Data represent means \pm SD, $n=3$. (C-E) Tumor growth curves (C), Tumor weights at completion of the study (D) and Body weights (E) of HNM007-bearing mice with either STAT1-overexpression ($n=5$) or empty vector control ($n=5$). The significance of (A, C and D) was determined by *t* test. * $P < 0.05$, *** $P < 0.001$.

Other comments:

1) The authors analyzed TP63 gene expression in several different cancers using TCGA data and showed that TP63 expression is highest in SCCs (Figure S1A). In the same analysis, what about STAT1 gene expression? Did the authors observe a negative correlation between STAT1 and TP63 gene expression?

Authors' reply:

These questions were also raised by *Referee #1* (see pages 5 in this Response Letter). As replied earlier, we interrogated RNA-seq data of both TCGA and CCLE from various cancer types. Results from TCGA dataset did not reveal a significant negative correlation likely due to the confounding STAT1 signals from immune cells (which do not express TP63) in patient tumor samples. However, a strong anti-correlation between the expression of TP63 and STAT1 was found in the CCLE RNA-seq dataset, including ESCC and LUSC (**Figure L1C**; Note that Figures L1C is a duplicate of the same figure in page 5 of this Response Letter. We have pasted it here for review convenience).

Figure L1C. Cancer types that show negative or positive correlation between the expression of TP63 and STAT1. For each cancer type, we obtained gene expression data of cancer cell lines from the CCLE project. Then we conducted correlation analyses between the expression level of TP63 and STAT1. The significant correlations were colored.

2) The authors observed that transduction of shTP63 into SCC cell lines (both TT and MOC22) resulted in increased expression of 23 ISG genes (Fig. 1E and S1F). This was observed upon IFN γ treatment, but curiously, it was also observed without IFN γ treatment. If there are any internal or external factors that can promote the increase in ISG genes - which is likely dependent on the transcriptional activity of IFN-induced p-STAT1 - even in the absence of exogenous IFN γ , the authors need to clarify this.

Authors' reply:

Thank you for commenting on this, which was also asked by *Reviewer #1* (page 2). To be clear, our study intends to focus on the exogenous IFN γ -induced STAT1 pathway because:

(1) In the absence of IFN γ stimulation, STAT1, and in particular p-STAT1, were barely expressed in SCC tumor cells (see **Figure L12** on page 26 in this Response Letter).

(2) Our data of Figures 1E and S1F in the original manuscript have shown much more significantly upregulated expression of ISGs with exogenous addition of IFN γ relative to TP63-knockdown alone, suggesting strong synergistic effect of exogenous IFN γ and TP63-depletion on the stimulation of STAT1 signaling pathway (Figures 1E in the revised manuscript).

(3) The prominent and indispensable role of exogenous IFN γ signaling pathway has been well documented in immune response and anti-tumor immunotherapy in multiple cancer types (Gao et al., 2016; Liao et al., 2019; Zaretsky et al., 2016).

As suggested by both Referees, during the revision, we have carefully addressed this issue in each figure and corresponding paragraphs and now only include data on *exogenous* IFN γ signaling.

3) Since the authors used a variety of SCC cell lines in this study, it would be helpful to interpret the data by checking to what extent TP63 and STAT1 expression differs between these cell lines at the protein level in at least the most commonly used cell lines (MOC22, AKR, HNM007), if not all, and if there is a negative correlation between these expressions.

Authors' reply:

To address this query, during the revision we examined the baseline levels of TP63 and STAT1 in those murine SCC cell lines used in this study, including MOC1, MOC22, AKR and HNM007. While the expression of TP63 was comparable in these cells, the levels of STAT1 were different (**Figure L16**). This is not surprising given that STAT1 expression is strongly regulated by IFN signaling. Most importantly, downregulation of TP63 increased the expression of STAT1 and p-STAT1 (**Figures 6E-6F** in our manuscript). On the other hand, STAT1 inhibition with siRNA or inhibitor-Fludarabine

significantly elevated TP63 level (**Figures 6J and L24**, see page 54 in this response letter).

Figure L16. qRT-PCR analysis showing relative mRNA levels of Trp63 and STAT1 in 4 murine SCC cells.

4) In Fig. S4C and S4D, the authors separated the TP63^{high} and TP63^{low} (top/bottom 15%) groups based on public scRNA-seq data of 60 ESCC patient specimens, showing a clear difference in CD8 T cell frequency between them. Here, did the authors investigate CD8 T cell frequency in patients with median TP63 expression levels, and if so, did the values fall somewhere in between those observed in the TP63^{low} and TP63^{high} groups?

Authors' reply:

To address this question, we investigated CD8 T cell frequency in patients with median TP63 expression levels and compared with TP63-high and TP63-low groups. Yes, as expected, the fraction of CD8 T cells in TP63-median ESCC specimens was between those observed in the TP63-low and TP63-high groups (**Figure L17**).

Figure L17. (A) Beeswarm plot showing TP63 expression in each CD45- cell of each sample. Patient samples are ranked by the median expression level of TP63 highlighted with red bars. The top/middle/bottom 15% patient samples according to TP63 expression are labeled with blue, black and red dotted-line rectangle. TP63-low samples, n=9; TP63-medium samples, n=9; TP63-high samples, n=9. (B) Box plots showing the cell proportion of CD8 T cells in total T cells of TP63-low/medium/high expressing samples. The significance was determined by t test. *P < 0.05. n.s., not significant.

5) The authors observed in Figs. 5G and 5H that tumor growth was inhibited in the shTP63 group compared to the scramble group, and further showed that this inhibition was further amplified by anti-PD1 treatment. This reviewer has two questions here. First, the reduction in tumor growth by shTP63 was only observed in AKR and not in HNM007 (Scramble + IgG2a vs shTrp63 + IgG2a in Fig. 5G and 5H), did the authors check whether this is related to the difference in TP63 expression levels between these two cell lines? Second, did the authors observe the effect of CD8 T cell depletion in the absence of anti-PD1 treatment? In other words, did the authors check whether the reduction in AKR tumor growth observed in the shTP63 group (Scramble + IgG2a vs shTrp63 + IgG2a in Fig. 5G) was also CD8 T cell-dependent?

Authors' reply:

Thank you for your questions. For the first question, in fact, the significant reduction in tumor growth caused by shTP63 was observed in both AKR and HNM007 tumors (black line of Scramble + IgG2a vs blue line of shTrp63 + IgG2a in Figure 5G and 5H). However, the PD-1 mAb treatment-induced decrease in tumor growth was only observed in AKR tumors (Figures 5G and 5H in the revised manuscript). The variation in these SCC models may be caused by: (1) distinct genetic background: AKR and HNM007 were established from esophageal cell lines transformed by *cyclin-D1* overexpression+ loss of *p53*, and mutant *HRAS*^{G12V} + loss of *p53*, respectively (Opitz et al., 2002; Predina et al., 2011); (2) different immunogenicity of these cells; and (3) the amount of internal TP63 levels in these cell lines (AKR has higher TP63 expression than its in HNM007, see mRNA data shown before) (**Figure L16**).

Regarding the second query that whether the reduction in AKR tumor growth observed in the shTP63 group was also CD8 T cell-dependent, additional mouse model

experiments were performed in the absence of anti-PD1 treatment during the revision. As shown in Figure L18A and L18B, depletion of CD8 T cell with CD8 mAb resulted in continued tumor growth and elevated tumor weight (**Figure L18**). These results demonstrate that the reduction in AKR tumor growth observed in the shTP63 group is also CD8 T cell-dependent. The new generated data has been incorporated into our revised manuscript as **Figure S7E**.

Figure L18. (A-C) Tumor growth curves (A), Tumor weights at completion of the study (B) and Body weights (C) of sh*Trp63* AKR-bearing mice treated with CD8 mAb (CD8 T-cell-depletion antibody) (n=5) or IgG isotype control (IgG2b) (n=5). mAbs were given by intraperitoneal injection (i.p., 100 μ g/injection/mouse) once every 2 days for up to 2 weeks. The significance of was determined by *t* test. * $P < 0.05$, ** $P < 0.01$.

6) In Fig. 6, the authors described the physical interaction of TP63 and phospho-STAT1 (specifically on Ser727 but not on Tyr701) and the reciprocal transcriptional regulation between them. Although p-STAT1 (Ser727) can be induced in an IFN signaling-dependent manner together with p-STAT1 (Tyr701), it was shown that the former can also be formed by IFN-independent signaling by various stress-induced MAP kinases (ERK, P38, JNK, etc.). In addition, even in the case of IFN-induced p-STAT1 (Ser727), previous studies have shown that it requires the transcriptional activity of p-STAT1

(Tyr701) (Sadzak et al., PNAS, 105:8944, 2008).

Therefore, the authors need to check TP63-p-STAT1(Ser727) or possible TP63-p-STAT1(Tyr701) interactions very early after IFN treatment (~0.5-1 hr), as the possibility of interaction between p-STAT1(Tyr701) and TP63, which is expected to appear and disappear quickly after IFN treatment, cannot be ruled out. Furthermore, rather than limiting these conditions to IFN γ , a type II IFN (mainly induce p-STAT1/STAT1 homodimer that binds GAS element), a cross-comparison with IFN α /b, a type I IFN (predominantly induce p-STAT1/STAT2 heterodimer that combines IRF9 and binds ISRE regions to induce ISG gene expressions) treatment will be very useful to clearly interpret and generalize the IFN/STAT-related mechanism of TP63 in this study.

Authors' reply:

We thank the Reviewer for these insightful comments. In fact, we indeed observed a detectable p-STAT1 (Tyr701) in the TP63-immunoprecipitates upon long exposure time. Reciprocally, very weak TP63 bands were detected in the p-STAT1 (Tyr701) interactors (**Figure L19**). Therefore, we do not rule out the interaction between p-STAT1 (Tyr701) and TP63. As suggested by the Referee, we have cited the paper (Sadzak et al., 2008) and revised our interpretation as below (Page 12, paragraph 3):

“Following co-immunoprecipitation (co-IP) assays, we confirmed the physical interaction between endogenous TP63 and p-STAT1 (particularly Ser727 and weakly Tyr701) proteins in both of human and murine SCC cells using either TP63 or p-STAT1 antibodies for reciprocal immunoprecipitation.”

Figure L19. Co-IP followed by Western blotting analysis showing the protein interaction between TP63 and phosphorylated STAT1 (Ser727 and Tyr701) in both human TT/TE5 and murine MOC1/MOC22 cells with long and short exposure time, respectively.

With respect to your suggestion on a cross-comparison of IFN γ with IFN α/β treatment, we performed additional experiments accordingly. Unlike the result of IFN γ exposure which caused downregulation of TP63 expression (**Figures L20A-L20B**, corresponding to Figures 6H, 6I, S9E and S9F), administration of IFN α did not affect TP63 expression at either mRNA or protein level (**Figures L20C-L20D**), suggesting that it is IFN γ but not IFN α which plays a suppressive role in regulating TP63 expression in SCCs.

Figure L20. (A and B) qRT-PCR (A) and Western blotting (B) analysis revealing a time-dependent expression of TP63 in both TT and MOC22 cells in responding to the stimulation of IFN γ (100 ng/mL). (C and D) qRT-PCR (C) and Western blotting (D) analysis revealing the mRNA and protein level of TP63 in both TT and MOC22 cells in responding to the stimulation of IFN α . Data of (A and C) represent means \pm SD, n=3. The significance was determined by *t* test. *P < 0.05, **P < 0.01; N.S., not significant.

Minor corrections:

In line 208-209, "~observed in TP63-low tumor samples (median=40.07%) than TP63-high tumors (median= 52.29%)" should be corrected to "~observed in TP63-low tumor samples (median=52.29%) than TP63-high tumors (median= 40.07%)"

Authors' reply:

Thank you for spotting the error, which has now been corrected in the revised manuscript (Page 9, paragraph 3).

Reviewer #3 (Remarks to the Author):

Jiang et al. report on a negative regulator of IFN γ -signalling via TP63, a transcription factor which is found to be ubiquitously overexpressed in several human squamous cell cancer types. The authors explored their own transcriptomics data and publicly available datasets to analyze the relationship between TP63 expression and IFN γ signalling in squamous tumors. They further used in vitro and in vivo experiments to characterize the co-regulation between TP63 and Stat1 expression, and the relevance of this crosstalk. Subcutaneous in vivo SCC murine models with TP63 knockdown by shRNA show increased CD8 $^+$ T cell infiltration, and the combination of knockdown plus checkpoint blockade in vivo appears to slow tumour growth. The manuscript is well-written, the coregulation between TP63 and STAT1 is novel, and the data is compelling for the most part. Below are some comments that will improve the manuscript.

Authors' reply:

It is with great pleasure to find that the Referee considers our manuscript is well-written, the coregulation between TP63 and STAT1 is novel, and the data is compelling for the most part. We are also appreciative of your important comments below, which have helped improved the manuscript.

Main comments:

1- The overall observations would greatly benefit from addressing whether TP63 high tumours are a result of loss of IFN signalling, or vice versa, whereby TP63 overexpression results in desensitization of tumors towards IFN γ . This is important for understanding whether TP63 expression is dysregulated, leading to resistance of IFN γ is simply a biomarker/readout of the state of the immune response. The fact that the authors observe an inverse correlation between CD8 T cell proportion and TP63 expression (Fig.3E) suggests the latter. To answer this, the authors could explore whether TP63 high tumours are always associated with an overall immune cold phenotype or CD8 excluded phenotype. Similarly, are PD-1 unresponsive tumours TP63 high in humans? Can TP63 be used to stratify patient response to checkpoint blockade? At the very least, the discussion should be a little more nuanced and reflect

the possibility that TP63 in human datasets might just be a marker of cold tumors.

Authors' reply:

We thank the Referee for this valuable suggestion. Due to a limited number of in-house samples, we could not determine the association between the expression of TP63 and responsiveness to ICB treatment and desensitization towards IFN γ . However, the negative correlation between TP63 expression and CD8 T cell fraction was revealed in both 60 published scRNA-seq and over 1,000 bulk RNA-seq data from SCC patient samples (Figure 3E in our manuscript). Meanwhile, our immunofluorescence staining indeed confirmed the prominent accumulation of CD8⁺ T cells in ESCC tumors with low/no expression of TP63, but only a few scattered CD8⁺ T cells at the edge of cancer lesions in TP63-high tumors using human ESCC slides samples from an independent cohort of 10 ESCC patients (Figures 4D-4E in our manuscript). In addition, low expression of TP63 has been found in a subset of “immune modulation” ESCC patients (**Figure L21A**), who were more responsive to single PD-1 mAb therapy in a phase I clinical trial of ICB therapy (NCT02742935) (Liu et al., 2023). This study also identified a 28-feature classifier associated with ICB therapy response, with TP63 expression being ranked as the second most significant feature (**Figure L21B**; Note Figures L21A-L21B were published data by Liu, et al. 2023, which was pasted here only for review convenience). These independent observations from patient samples underscore the importance of TP63 in regulating anti-tumor immune response.

As suggested by the Referee, the following discussion has now been added into the revised manuscript (Page 16, paragraph 2):

“These independent observations from patient samples underscore the importance of TP63 in regulating anti-tumor immune response, and suggest that TP63 expression might serve as a biomarker associated with immune-cold tumors in SCC. Inhibition of TP63 may be a potential approach to turn immune-cold SCC tumors into immune-hot ones.”

Figure L21. (A) Heatmap showing the 28 features and their normalized levels within each sample grouped by the four ESCC subtypes in ECGEA. Features of mRNA expression, copy number alterations, and methylation are colored as indicated. (B) Bar graph of feature importance of the classifier. The mean decrease accuracy axis represents the percentage that the accuracy of the classifier will decrease if the indicated feature is removed. Features of gene expression, copy number alterations, and differentially methylated regions are colored as red, yellow, and green, respectively. Asterisk on the right of each bar represents the significance level of each feature. *P < 0.05, **P < 0.01. Permutation test of 100 iterations using the “rfPermute” R package. Figures L23A-B were published data by Liu, et al. 2023.

2- The fact that TP63 downregulation in tumours is enough to observe an effect on cytotoxicity during in vitro co-culture of CD8 T cells and tumours is interesting. Providing some type of possible mechanism would greatly increase the value of the paper. For example, assessing the expression and relevance of MHC-I, PDL-1 or CXCL9/10 could help understand what is going on. In quite a few places, the authors investigate mRNA expression, but it is important to demonstrate that protein expression is also altered. In the same line, how is T cell priming affected by TP63 overexpression or downregulation?

Authors' reply:

Thank you for this important comment, which was also raised by Referee #1 in the 13th point (see page 21 in this Response Letter). To address this query, during the revision we first performed correlation analysis between the expression of TP63 and

MHC-I (human HLA), B2M and CD274 (PD-L1) in SCC patient samples, including ESCC, HNSC and LUSC. Notably, the TP63 level was negatively correlated with the expression of HLA-A, HLA-B and HLA-C (except in ESCC), as well as the MHC-I component-B2M (**Figure L9**). There was very weak or no correlation between the expression of TP63 and CD274 in these patient samples (**Figure L9**).

Next, we analyzed their correlation in CD45⁺ populations using scRNA-seq data from 60 ESCC patients. In line with the above results, a significantly negative correlation between TP63 and HLA/B2M, but not CD274, was found (**Figure L10A**). More importantly, our new western blot assays revealed higher protein levels of MHC-I and B2M upon downregulation of TP63 under stimulation by IFN γ (**Figure L10B**). These additional data and our original findings together suggest that TP63 hampers CD8 T cell-mediated tumor killing likely through suppressing MHC-I and B2M, key downstream targets of the IFN γ -STAT1 signaling (**Figure L10C**).

Figure L9. Correlation analysis between the expression of TP63 and HLA, B2M and CD274 in ESCC, HNSC and LUSC. The immune cell fractions were predicted by TIMER2 using TCGA

expression datasets. Pearson Correlation Coefficient was calculated. Note the Figure L9 is a duplication of the same figure in page 22 of this Response Letter. We have also pasted it here for review convenience.

Figure L10. (A) Correlation analysis between the expression of TP63 and HLA, B2M and CD274 in CD45 negative cells of 60 ESCC patient tumors. Data from scRNA-seq dataset (GSE160269). Pearson Correlation Coefficient was calculated. (B) Western blotting analysis showing the protein levels of TP63, MHCI, B2M, with GAPDH as a positive control in human (TT) and murine (MOC22) SCC cells expressing either non-targeting control (Scramble) or TP63-targeting shRNA

(shTP63) pulsed with IFN γ (100 ng/mL) for 48 hours. (C) A work model of TP63 suppressed IFN γ -STAT1 immune regulatory pathway in SCC tumor microenvironment. Note the Figure L10 is a duplication of the same figure in page 23 of this Response Letter. We have also pasted it here for review convenience.

3- The authors suggested that TP63 can directly inhibit STAT1 activity through binding of the transcription factor with phosphorylated protein (Figure 6). We suggest that authors show the experimental condition where although pSTAT1 can be detected, downstream signalling effects such as MHC-I/II upregulation are abrogated in the presence of TP63. In figure 6C, IP-pulldown blots are missing a total STAT1 control, and negative control such as a non-related, non-interacting protein which is non-nuclear. In Figure 6J, data is not obvious just by looking at the blot, it would be beneficial to quantify the bands and normalize to GAPDH band intensity.

Authors' reply:

Regarding the first question on the suppression of MHC I by TP63, it has been addressed above (**Figure L10B**).

Regarding the IP-pulldown blots, as suggested by the Reviewer, we have now added the total STAT1 control (**Figure L22A**) and a negative control protein B2M (**Figure L22B**). As anticipated, no interaction was detected between TP63 and B2M (**Figure L22B**). In addition, slight to moderate interaction between TP63 and total STAT1 (due to extensive affinity of total STAT1 antibody to both p-STAT1 and STAT1) was observed at long exposure time (**Figure L19**, page 42 in this Response Letter), which is consistent with previous reports and our finding of the nuclear localization of phosphorylated STAT1 and its interaction with TP63.

For the last question on Figure 6J, as we have responded to the *Referee#1's 3rd point earlier (please refer to page 8 in this Response Letter)*, three independent experimental replicates in two cell lines were performed for Figure 6J. The results confirmed consistently reduced levels of STAT1 and p-STAT1 treated with STAT1-specific inhibitor, Fludarabine, in both TT and TE5 cells. Importantly, STAT1-inhibition by Fludarabine rescued TP63 levels reduced by IFN γ treatment in all experiments (**Figures L2G-L2H**). We now have shown both statistical analyses and individual replicates in the revised manuscript.

Figure L22. (A) Co-IP followed by Western blotting analysis showing the protein interaction between TP63 and phosphorylated STAT1 (Ser727 and Tyr701), total STAT1 in both human TT/TE5 and murine MOC1/MOC22 cells. (B) Co-IP-pulldown blots showing a TP63-non-interacting protein B2M, as a negative control.

4- Usually, it is required to treat T cells with Brefeldin A or Monensin to detect IFN γ production by T cells. The authors don't seem to use anything like this, but this is important to reproduce that experiment with Brefeldin to confidently detect IFN γ in their in vitro system (Fig.5E,F).

Authors' reply:

As suggested by the Referee, during the revision we treated T cells with Brefeldin A in *in vitro* tumor killing assays and detected IFN γ production, confirming our original results. Indeed, conspicuous enhancement of IFN γ production was observed upon the treatment of Brefeldin A. Importantly, in line with our original results, TP63-

knockdown significantly increased the IFN γ production. Moreover, overexpression of TP63 in the shTrp63 group reversed the IFN γ production to its original level (**Figure L23**). The new data has been added in the revised manuscript (**Figure S6G**).

Figure L23. (A and B) Representative flow cytometry results (A) and Percent of IFN γ ⁺ production (B) from (A) in Scramble, shTrp63 and TP63-overexpressing MOC22 and OT-I T cell co-cultures. MOC22 cells were transfected with siRNA targeting TP63 or non-targetable control (Scramble) and ectopically expressed TP63 or empty vector (Control) following 24 hr stimulation of IFN γ (10 ng/mL), and then incubated with OT-I CD8 T cells for 48 hr at 5:1 effector: target (E: T) ratio. Brefeldin A was added for the last 4 hr before harvest of cells. Trp63-KD: Trp63 knockdown; OE-Trp63: ectopic expression of TP63. Data represent means \pm SD, n=3. The significance was determined by *t* test. *P < 0.05; N.S., not significant.

5- Finally, to confirm the reciprocal relationship between STAT1 and TP63 interaction by knocking down STAT1 – do these cells have a corresponding increase in TP63?

Authors' reply:

To answer this question, we first silenced STAT1 expression with 4 independent

siRNAs targeting STAT1. We found that siSTAT1-3 and siSTAT1-4 exhibited the greatest knockdown efficiency, which were selected for further experimentation (**Figure L24A**). We next transfected siSTAT1-3 and siSTAT1-4 into TT and TE5 cells and examined the effect of STAT1-knockdown on the expression of TP63. Importantly, consistent with our results obtained by the treatment of STAT1 inhibitor-Fludarabine, knockdown of STAT1 significantly increased the protein levels of TP63 (**Figure L24B**). These results confirm our original finding that STAT1 inhibits the expression of TP63. The new data has been added into the revised manuscript (**Figure S10C**).

Figure L24. (A) qRT-PCR analysis showing mRNA of STAT1 upon downregulation of STAT1 with 4 independent siRNA target. Data represent means \pm SD, n=3. *P < 0.05, **P < 0.01, *** P < 0.001. (B) Western blotting assays showing protein levels of indicated proteins upon knockdown of STAT1 (with #3 and #4 siRNA target) or not.

Minor comments:

- In figure 2, why are Dox timing different between the scramble or TP63 shRNA?

Authors' reply:

In Figure 2, the doxycycline was in fact given at the same time, on day-10, for both the scramble and the TP63 shRNA group of MOC22 tumors. We apologize if our experimental description was not entirely clear in our initial submission: for consistency across different SCC-derived models, we started doxycycline

administration based on the tumor growth (when reaching 100 mm³). Because AKR and HNM007 cells proliferate faster and are more aggressive than MOC22 cells, administration of doxycycline was from day-10 for MOC22, and from day-5 for AKR and HNM007 after the inoculation, respectively. However, again **for the scramble and the TP63 shRNA group within each of these SCC models, doxycycline was given at the same time.** The details have been described in the section of Methods in the revised manuscript (Page 19, paragraph 2).

- Figure 2I and 5E&F: Representative FACS plots should be shown in the supplementary to support bar graphs, especially for intratumoral cytokine-related stains and gating of positive populations for both tumors.

Authors' reply:

As requested, for the Figure 2I, the gating strategy and a representative staining example have been shown as Figure L7 on page 17 of this Response letter. For Figures 5E and 5F, below is the gating strategy (**Figure L25**):

Figure L25. The exemplification of the gating strategy for intracellular cytokine stains in *ex vivo* co-culturing system (related to Figure 5E and 5F). After 48 hr of co-culturing SCC and CD8 T cells, adherent living cancer cells were stained crystal violet, while suspended CD8 T cells were collected for flow cytometry analysis of GZMB and IFN γ . Single cells were first gated. The expression of activation and cytotoxic protein markers (GZMB, IFN γ) in the co-culturing system was analyzed.

- The authors focused on CD8 T-cells based on Figure 2, but there are other changes that occur, especially a diminution in neutrophils and macrophage 3. The authors should elaborate on this.

Authors' reply:

Yes, the Referee is correct that diminution of macrophage 2/3 and neutrophil populations were also seen upon TP63 downregulation. Accordingly, the following comments have been added to the revised manuscript (Page 8, paragraph 1):

“Tumor associated macrophages and neutrophils can promote tumor cell survival and immune evasion (Xue et al., 2022; Zhang et al., 2020b)^{33,34}. It is interesting to note that macrophage 2/3 and neutrophil populations were also decreased upon TP63 knockdown. Further investigations on potential regulation on macrophages and neutrophils by TP63 are thus necessary to further understand the complexity of tumor microenvironment and immune response of SCCs.”

- What is the expression of IFN γ R1 and IFN γ R2 in the cell lines used?

Authors' reply:

To address this query, during the revision we first analyzed the expression of IFN γ R1 and IFN γ R2 in 24 human ESCC cell lines using CCLE dataset and found that both of the receptors are highly expressed (**Figure L26A**). We further performed quantitative RT-PCR analysis in murine SCC cell lines. Consistent with the results from CCLE data, IFN γ R1 and IFN γ R2 had high to moderate expression levels across these murine cells. Among cell lines used in our study, MOC22 had highest expression of IFN γ R1 while in HNM007 had the highest IFN γ R2 level (**Figure L26B**).

Figure L26. (A) Bar chart showing the expression of IFNGR1 and IFNGR2 in 24 human SCC cell lines. Data from CCLE dataset. (B) qRT-PCR analysis revealing mRNA levels of IFNGR1 and IFNGR2 in 7 murine SCC cell lines.

- In line 266, the authors don't know whether that's the infiltration or expansion that's increased in TP63 shRNA tumors.

Authors' reply:

Yes, at this point, we are unable to ascertain whether the increased CD8 T cell population is due to increased T cell infiltration or expansion in TP63-knockdown tumors relative to control group. According to our new finding that TP63 can suppress

MHC-I and B2M (**Figure L10C**), the increased CD8 T cells are more likely to be caused by the infiltration in TP63 shRNA tumors. However, future investigations are required to confirm this mechanism.

- Figure 5A: schematic suggests that IFN γ was added to change antigen presentation capability of tumour cells but corresponding data is not shown.

Authors' reply:

We apologize if the schematic description was not immediately clear. Here, we showed the tumor killing results in the presence of IFN γ in Figures 5B-5F. The comparison data was shown in Figure S5D that without IFN γ treatment in our original manuscript (**Figure S6A** now in our revised manuscript), only slight tumor cell killing was observed upon knocking down of TP63 at 5:1 ratio (E: T). However, pretreatment with IFN γ and TP63-knockdown strongly augmented tumor cell killing by CD8 T cells in both MOC22 and AKR cell lines, at different effector: target ratios.

- In figure 5J, the colors and legend below the graph don't match the graph itself.

Authors' reply:

We apologize for the confusion. Figures 5J and 5K used the same color legend, wherein black curves represent the sh*Trp63*+ α -PD-1 mAb+IgG2b and the red denotes sh*Trp63*+ α -PD-1 mAb+CD8 mAb group.

Reviewer #4, ECR co-reviewed with Reviewer #3(Remarks to the Author):

Authors' reply:

Thank you for your note.

Reviewer #5 (Remarks to the Author):

Jiang et al applied scRNA-seq, immune profiling, and ChIP-seq to squamous cell carcinoma cell lines and mouse models to show that the SCC master regulator TP63 suppresses the interferon signaling through repressing STAT1 expression. They also showed that silencing of TP63 increased CD8+ T cell infiltration, which enhanced anti-tumor efficacy of immune checkpoint blockade therapy. The experiments were designed and performed properly, and the manuscript was well written. The findings of the paper are of potential interest to the field of squamous cell carcinoma and oncogene-related immune suppression. That being said, I suggest the following experiments and analyses that in my view would improve the rigor of the work.

Authors' reply:

We are very appreciative for the Referee's positive comments. We also thank you for offering the valuable suggestions below!

1. Given the potential off-targets effects of shRNAs, I would suggest performing rescue experiments for key findings of the manuscript, at least for the in vitro part. For instance, does ectopic expression of TP63 rescue the expression of STAT1 and IFN genes up-regulated upon TP63 silencing? And does expressing TP63 rescue the increased T-cell killing in the ex vivo co-culturing system?

Authors' reply:

As suggested by the Referee, during the revision we ectopically overexpressed TP63 in the TP63-knockdown cells and repeated the *ex vivo* co-culture and tumor killing assays. Validating our original results, knockdown of TP63 increased the levels of both STAT1 and p-STAT1, as well as downstream IFN γ -response genes, such as B2M and IRF1. More importantly, this upregulation was rescued by ectopic expression of TP63 (**Figure L27A**). Consistently, in the *ex vivo* co-culturing assay, ectopic expression of TP63 rescued T-cell killing caused by TP63-knockdown (**Figure L27B**). These results confirm our finding that TP63 suppresses IFN γ /STAT1 signaling and CD8 T-cell killing. These new data have been added into the revised manuscript (**Figures S6E and S6F**).

Figure L27. (A) Western blotting analysis showing the protein levels of TP63, p-STAT1 and STAT1 in the indicated conditions in MOC22 cells. MOC22 cells were transfected with siRNA targeting TP63 or non-targetable control (Scramble) and ectopically expressed TP63 or empty vector (Control) in the presence of IFN γ (100 ng/mL). Trp63-KD: Trp63 knockdown; OE-Trp63: ectopic expression of TP63. (B) Crystal violet staining of MOC22 cells pretreated with IFN γ (10 ng/mL) and incubation with OT-I CD8 T cells for 48 hr at 5:1 effector: target (E: T) ratio.

2. The authors did a good job incorporating different datasets to support their conclusions. However, some of the key experiments were performed in only one or two cell lines. I would suggest adding STAT1 ChIP-seq in more lines, especially lines that already have TP63 ChIP-seq data. I would also suggest assessing the repressive role of TP63 on STAT1 expression in more cell lines. Also, although most of the in vitro experiments were done in TT cells, this cell line seems missing ChIP-seq of STAT1/TP63 data and luciferase reporter data.

Authors' reply:

Thank you for the important suggestion. Accordingly, during the revision we first attempted the ChIP-seq assay using either p-STAT1 (Ser727) or STAT1 antibody in both TT and TE5 cell lines. Although high mapping rate was achieved in each ChIP assay, unfortunately, the peak number was too low (5-288 peaks) to analyze (Figure

L28A). We next decided to search for public datasets and found only 1 ChIP-seq data of STAT1 from SCC cells (Fadu, **Figure L28A**).

Nevertheless, despite the lack of ChIP-seq data, we have studied the repressive role of TP63 on STAT1 expression in multiple SCC cell lines, showing that i) knockdown of TP63 increased STAT1 and p-STAT1 levels in TT, TE5 and MOC22 cell lines, and importantly, ii) ectopic overexpressing TP63 rescued the repression of STAT1 induced by TP63-knockdown in KYSE410, TE1 and MOC22 cells (**Figure L27** and Figure 6G in manuscript).

For the genomic occupancy of TP63 in TT cells, we have now re-analyzed our TP63 ChIP-seq data. A total of 6,038 TP63-binding peaks were obtained in TT cells through quality control and normalization. We then integrated TP63-binding peaks in TT cells with those obtained in TE5 cells, and found that the majority (74.8%, 4,483/5,993) were overlapped in these two SCC cell lines (**Figure L28B**). These 4,483 overlapped peaks were then used for analyzing the TP63/STAT1-unique and -shared peaks, as well as associated GO biological process. Importantly, consistent with our original data, we found that: (i) the majority of TP63 and STAT1 binding peaks were located at intergenic and intron regions, and 5.7%-15.2% peaks were enriched at promoters (**Figures L28C and L28D**); (ii) both TP63 and STAT1 binding motifs were observed in shared peaks; while in unique peak sets for each TF, only corresponding individual motif was exclusively enriched (**Figure L28E**); (iii) uniquely-occupied regions by TP63 were predominantly assigned to genes associated with epithelial cell development, epithelial cell proliferation, and Wnt signaling pathway. In comparison, those specifically bound by STAT1 were primarily associated with immune response pathways, including IFN γ -mediated signaling, JNK cascade, and T cell activation signaling (**Figure L28F**).

Finally, as suggested by the Referee, we performed luciferase reporter assays in TT cells. Consistent with the results from TE5 and TE1 cell lines in our initial submission, IFN γ treatment significantly increased reporter activities of *STAT1*-promoter/enhancer while inhibited those of e8 in TT cells. Importantly, these changes in reporter activities were reversed by the STAT1 inhibitor - Fludarabine (**Figure L28G**). These concordant new results together our original findings demonstrate that TP63 and STAT1 mutually suppress the expression of each other by co-occupying and co-regulating their own promoters and enhancers.

A Schematic diagram of the strategy to identify corresponding TP63 and STAT1 ChIP-seq

ID	Sample	Mapped reads	Mapping rate	Peak number	FRIP
TE5_p727_STAT1vslgG	TE5_IgG	46,542,905	98.71%	5	0.00%
	TE5_p727_STAT1	41,146,956	91.04%		
TE5_STAT1vslgG	TE5_IgG	46,542,905	98.71%	21	0.21%
	TE5_STAT1	64,452,906	98.67%		
TT_p727_STAT1vslgG	TT_IgG	177,039,367	92.35%	288	1.03%
	TT_p727_STAT1	29,736,124	89.14%		
TT_STAT1vslgG	TT_IgG	177,039,367	92.35%	41	1.67%
	TT_STAT1	32,803,070	98.42%		

E

Gene Name	Motif	Rank	P Value
Shared Peaks	TP63 Family	1 - 4	1e-659 to 1e-419
	AP-1 Family	5 - 14	1e-189 to 1e-96
	STAT1 Family	15 - 21	1e-57 to 1e-36
STAT1	AP-1 Family	1 - 9	1e-3237 to 1e-2278
	STAT1 Family	10 - 14	1e-1993 to 1e-1592
TP63	TP63 Family	1 - 5	1e-4449 to 1e-392

Figure L28. (A) Strategy to identify binding peaks of TP63 or STAT1 using ChIP-seq data from our home-made and various datasets. (B) Integrative analysis of TP63 occupied peaks in both TE5 and TT cell lines. (C) Density plots of ChIP-seq signals of TP63 and STAT1 at ± 3 kb windows flanking the center of TP63 peaks. Color bars at the bottom show reads-per-million-normalized signals. Data were from GSE46837, GSE106563 and GSE148920. (D) Pie chart showing genome-wide distribution of the regions uniquely- or co-occupied by TP63 and STAT1. (E) Representative top TF motif sequences enriched at TP63 or/and STAT1 uniquely- or co-occupied loci revealed by de novo motif analysis. (F) Gene Ontology (GO) functional categories of TP63 or STAT1 uniquely occupied peak-assigning genes in (C). (G) Relative luciferase activity of pGL3-enhancer (1st Empty), pGL3-enhancer+STAT1 promoter, pGL3-promoter (2nd Empty), pGL3-promoter+ STAT1 enhancer, pGL3-promoter+e8 upon stimulation with IFN γ (100 ng/mL) +/- Fludarabine (1 μ M) in TT cells. Data represent means \pm SD, n=3. * P < 0.05, ** P < 0.01, *** P < 0.001.

3. The negative regulation of STAT1 on TP63 expression is interesting. However, IFN γ may trigger several other TFs in addition to STAT1. Knock-out or knock-down of STAT1 may provide more direct evidence regarding its role in TP63 regulation.

Authors' reply:

Yes, we agree the Referee's comment, which was also suggested by *Referee #3* in his/her 5th point (see pages 53 in this Response Letter). To address this query, we first silenced STAT1 expression with 4 independent siRNAs targeting STAT1. We found that siSTAT1-3 and siSTAT1-4 exhibited the greatest knockdown efficiency, which were selected for further experimentation (**Figure L24A**). We then transfected siSTAT1-3 and siSTAT1-4 into TT and TE5 cells and examined the effect of STAT1-knockdown on the expression of TP63. Consistent with our results obtained by the treatment of STAT1 inhibitor-Fludarabine, knockdown of STAT1 significantly increased the protein level of TP63 (**Figure L24B**). These results further confirm our original finding that STAT1 inhibits the expression of TP63. The new data has been added into the revised manuscript (**Figure S10C**).

Figure L24. (A) qRT-PCR analysis showing mRNA of STAT1 upon downregulation of STAT1 with 4 independent siRNA target. Data represent means \pm SD, n=3. *P < 0.05, **P < 0.01, *** P < 0.001. (B) Western blotting assays showing protein levels of indicated proteins upon knockdown of STAT1 (with #3 and #4 siRNA target) or not. Note the Figure L24 is a duplicate from page 54, we pasted it here only for review convenience.

4. The antagonistic role of TP63 and STAT1 is of great interest. The authors should perform a more thorough analysis assessing how genes adjacent to TP63-unique, STAT1-unique, and shared sites respond to TP63 or STAT1 perturbations. Also, the authors should discuss potential mechanisms of how these two TFs act as repressors for each other.

Authors' reply:

We appreciate that the Referee's interest in the antagonistic relationship between TP63 and STAT1. As suggested by the Referee, we first analyzed genes whose expression were affected by the either TP63 or STAT1 using RNA-seq data upon knockdown of either factor. Consistent with their known functions, silencing of TP63 primarily suppressed genes involved in cancer proliferation and survival, such as *SOX2*, *CDK7*, *ERFG* and *MYC*. In comparison, knockdown of STAT1 mainly decreased expression of genes participating in immune response, for example, *HLA-A*, *CD47*, *B2M* and *CXCL10* (**Figures L29A and L29B**).

To further assess how genes assigned to either TP63-unique, STAT1-unique, or

shared sites respond to TP63 and STAT1 perturbation, we integrated RNA-seq from TP63- or STAT1-knockdown with their corresponding ChIP-seq data (**Figures L29C-L29E**). Indeed, genes respectively associated with TP63-unique and STAT1-unique peaks were specifically (while modestly) downregulated upon silencing of TP63 and STAT1 (**Figures L29D and L29E**). In contrast, genes assigned to TP63/STAT1-shared peaks did not respond to either TP63-knockdown or STAT1-knockdown.

Finally, as suggested, discussion on the reciprocal repression between TP63 and STAT1 has been added in the revised manuscript (Page 15, paragraph 3), pasted as below:

“The present study uncovers a reciprocal inhibition between TP63 and STAT1 which dictates the strength of IFN γ signaling, supported by multiple lines of evidence. First, IFN γ treatment resulted in an upregulation of STAT1 and many ISGs, including IRF1, MHC-I, B2M and BATF2, as well as downregulation of TP63 expression. Over-expression of TP63 decreased the expression of STAT1 at both mRNA and protein levels. Secondly, TP63 and p-STAT1 physically interacted with each other at the protein level. Finally, TP63 and STAT1 form a seesaw-like transcription regulation on their own cis-regulatory elements (e8, STAT1-promoter, STAT1-enhancer) that were co-occupied by STAT1 and TP63 to mutually suppress the expression of each other. The three elements regulated the relative expression of TP63 and STAT1, which determines the strength of IFN γ signaling of SCCs.”

Figure L29. (A and B). Rank order of differentially expressed genes based on log₂ fold change between wildtype and TP63-knockdown cell line (A) or STAT1-knockdown cell line (B). (D and E) Boxplots of mRNA expression of genes associated with indicated peaks in wildtype and TP63-knockdown cell line (D) or STAT1-knockdown cell line (E). The expression data for TE5 wildtype and TP63 knockdown cell lines in A and D from GSE106564; while the expression data for SCC61 wildtype and STAT1 knockdown cell lines in B and E from GSE15848. P value was determined by one-tail wilcox test. **P < 0.01; n.s., not significant.

References

- Andreatta, M., Corria-Osorio, J., Muller, S., Cubas, R., Coukos, G., and Carmona, S.J. (2021). Interpretation of T cell states from single-cell transcriptomics data using reference atlases. *Nat Commun* *12*, 2965.
- Cancer Genome Atlas, N. (2015). Comprehensive genomic characterization of head and neck squamous cell carcinomas. *Nature* *517*, 576-582.
- Cancer Genome Atlas Research, N., Analysis Working Group: Asan, U., Agency, B.C.C., Brigham, Women's, H., Broad, I., Brown, U., Case Western Reserve, U., Dana-Farber Cancer, I., Duke, U., *et al.* (2017). Integrated genomic characterization of oesophageal carcinoma. *Nature* *541*, 169-175.
- Caushi, J.X., Zhang, J., Ji, Z., Vaghasia, A., Zhang, B., Hsiue, E.H., Mog, B.J., Hou, W., Justesen, S., Blosser, R., *et al.* (2021). Transcriptional programs of neoantigen-specific TIL in anti-PD-1-treated lung cancers. *Nature* *596*, 126-132.
- Daniel, B., Yost, K.E., Hsiung, S., Sandor, K., Xia, Y., Qi, Y., Hiam-Galvez, K.J., Black, M., C, J.R., Shi, Q., *et al.* (2022). Divergent clonal differentiation trajectories of T cell exhaustion. *Nat Immunol* *23*, 1614-1627.
- Dong, L., Chen, C., Zhang, Y., Guo, P., Wang, Z., Li, J., Liu, Y., Liu, J., Chang, R., Li, Y., *et al.* (2021). The loss of RNA N(6)-adenosine methyltransferase Mettl14 in tumor-associated macrophages promotes CD8(+) T cell dysfunction and tumor growth. *Cancer Cell* *39*, 945-957 e910.
- Dubrot, J., Palazon, A., Alfaro, C., Azpilikueta, A., Ochoa, M.C., Rouzaut, A., Martinez-Forero, I., Teijeira, A., Berraondo, P., Le Bon, A., *et al.* (2011). Intratumoral injection of interferon-alpha and systemic delivery of agonist anti-CD137 monoclonal antibodies synergize for immunotherapy. *Int J Cancer* *128*, 105-118.
- Dummer, R., Eichmuller, S., Gellrich, S., Assaf, C., Dreno, B., Schiller, M., Dereure, O., Baudard, M., Bagot, M., Khammari, A., *et al.* (2010). Phase II clinical trial of intratumoral application of TG1042 (adenovirus-interferon-gamma) in patients with advanced cutaneous T-cell lymphomas and multilesional cutaneous B-cell lymphomas. *Mol Ther* *18*, 1244-1247.
- Gao, J.J., Shi, L.Z., Zhao, H., Chen, J.F., Xiong, L.W., He, Q.M., Chen, T.H., Roszik, J., Bernatchez, C., Woodman, S.E., *et al.* (2016). Loss of IFN-gamma Pathway Genes in Tumor Cells as a Mechanism of Resistance to Anti-CTLA-4 Therapy. *Cell* *167*, 397-+.
- Glathar, A.R., Oyelakin, A., Gluck, C., Bard, J., and Sinha, S. (2022). p63 Directs Subtype-Specific Gene Expression in HPV+ Head and Neck Squamous Cell Carcinoma. *Front Oncol* *12*, 879054.
- Hao, Y., Hao, S., Andersen-Nissen, E., Mauck, W.M., 3rd, Zheng, S., Butler, A., Lee, M.J., Wilk, A.J., Darby, C., Zager, M., *et al.* (2021). Integrated analysis of multimodal single-cell data. *Cell* *184*, 3573-3587 e3529.
- Holling, T.M., Schooten, E., Langerak, A.W., and van den Elsen, P.J. (2004). Regulation of MHC class II expression in human T-cell malignancies. *Blood* *103*, 1438-1444.
- Jiang, Y., Jiang, Y.Y., Xie, J.J., Mayakonda, A., Hazawa, M., Chen, L., Xiao, J.F., Li, C.Q., Huang, M.L., Ding, L.W., *et al.* (2018). Co-activation of super-enhancer-driven CCAT1 by TP63 and SOX2 promotes squamous cancer progression. *Nat Commun* *9*, 3619.
- Liao, W.T., Overman, M.J., Boutin, A.T., Shang, X.Y., Zhao, D., Dey, P., Li, J.X., Wang, G.C.,

- Lan, Z.D., Li, J., *et al.* (2019). KRAS-IRF2 Axis Drives Immune Suppression and Immune Therapy Resistance in Colorectal Cancer. *Cancer Cell* *35*, 559-+.
- Liu, Z., Zhao, Y., Kong, P., Liu, Y., Huang, J., Xu, E., Wei, W., Li, G., Cheng, X., Xue, L., *et al.* (2023). Integrated multi-omics profiling yields a clinically relevant molecular classification for esophageal squamous cell carcinoma. *Cancer Cell* *41*, 181-195 e189.
- Luke, J.J., Bao, R., Sweis, R.F., Spranger, S., and Gajewski, T.F. (2019). WNT/beta-catenin Pathway Activation Correlates with Immune Exclusion across Human Cancers. *Clin Cancer Res* *25*, 3074-3083.
- Lutz, E.A., Agarwal, Y., Momin, N., Cowles, S.C., Palmeri, J.R., Duong, E., Hornet, V., Sheen, A., Lax, B.M., Rothschilds, A.M., *et al.* (2022). Alum-anchored intratumoral retention improves the tolerability and antitumor efficacy of type I interferon therapies. *Proc Natl Acad Sci U S A* *119*, e2205983119.
- Marzio, A., Kurz, E., Sahni, J.M., Di Feo, G., Puccini, J., Jiang, S., Hirsch, C.A., Arbini, A.A., Wu, W.L., Pass, H.I., *et al.* (2022). EMSY inhibits homologous recombination repair and the interferon response, promoting lung cancer immune evasion. *Cell* *185*, 169-183 e119.
- Melino, G. (2011). p63 is a suppressor of tumorigenesis and metastasis interacting with mutant p53. *Cell Death Differ* *18*, 1487-1499.
- Opitz, O.G., Harada, H., Suliman, Y., Rhoades, B., Sharpless, N.E., Kent, R., Kopelovich, L., Nakagawa, H., and Rustgi, A.K. (2002). A mouse model of human oral-esophageal cancer. *J Clin Invest* *110*, 761-769.
- Patturajan, M., Nomoto, S., Sommer, M., Fomenkov, A., Hibi, K., Zangen, R., Poliak, N., Califano, J., Trink, B., Ratovitski, E., *et al.* (2002). DeltaNp63 induces beta-catenin nuclear accumulation and signaling. *Cancer Cell* *1*, 369-379.
- Predina, J.D., Judy, B., Aliperti, L.A., Fridlender, Z.G., Blouin, A., Kapoor, V., Laguna, B., Nakagawa, H., Rustgi, A.K., Aguilar, L., *et al.* (2011). Neoadjuvant in situ gene-mediated cytotoxic immunotherapy improves postoperative outcomes in novel syngeneic esophageal carcinoma models. *Cancer Gene Ther* *18*, 871-883.
- Sadzak, I., Schiff, M., Gattermeier, I., Glinitzer, R., Sauer, I., Saalmuller, A., Yang, E., Schaljo, B., and Kovarik, P. (2008). Recruitment of Stat1 to chromatin is required for interferon-induced serine phosphorylation of Stat1 transactivation domain. *Proc Natl Acad Sci U S A* *105*, 8944-8949.
- Steurer, S., Riemann, C., Buscheck, F., Luebke, A.M., Kluth, M., Hube-Magg, C., Hinsch, A., Hofmayer, D., Weidemann, S., Fraune, C., *et al.* (2021). p63 expression in human tumors and normal tissues: a tissue microarray study on 10,200 tumors. *Biomark Res* *9*, 7.
- Tu, W.B., Shiah, Y.J., Lourenco, C., Mullen, P.J., Dingar, D., Redel, C., Tamachi, A., Ba-Alawi, W., Aman, A., Al-Awar, R., *et al.* (2018). MYC Interacts with the G9a Histone Methyltransferase to Drive Transcriptional Repression and Tumorigenesis. *Cancer Cell* *34*, 579-595 e578.
- Xue, R., Zhang, Q., Cao, Q., Kong, R., Xiang, X., Liu, H., Feng, M., Wang, F., Cheng, J., Li, Z., *et al.* (2022). Liver tumour immune microenvironment subtypes and neutrophil heterogeneity. *Nature* *612*, 141-147.
- Zaretsky, J.M., Garcia-Diaz, A., Shin, D.S., Escuin-Ordinas, H., Hugo, W., Hu-Lieskovan, S., Torrejon, D.Y., Abril-Rodriguez, G., Sandoval, S., Barthly, L., *et al.* (2016). Mutations Associated with Acquired Resistance to PD-1 Blockade in Melanoma. *New Engl J*

Med 375, 819-829.

Zhang, H., Christensen, C.L., Dries, R., Oser, M.G., Deng, J., Diskin, B., Li, F., Pan, Y., Zhang, X., Yin, Y., *et al.* (2020a). CDK7 Inhibition Potentiates Genome Instability Triggering Anti-tumor Immunity in Small Cell Lung Cancer. *Cancer Cell* 37, 37-54 e39.

Zhang, L., Li, Z., Skrzypczynska, K.M., Fang, Q., Zhang, W., O'Brien, S.A., He, Y., Wang, L., Zhang, Q., Kim, A., *et al.* (2020b). Single-Cell Analyses Inform Mechanisms of Myeloid-Targeted Therapies in Colon Cancer. *Cell* 181, 442-459 e429.

Zhang, X., Peng, L., Luo, Y., Zhang, S., Pu, Y., Chen, Y., Guo, W., Yao, J., Shao, M., Fan, W., *et al.* (2021). Dissecting esophageal squamous-cell carcinoma ecosystem by single-cell transcriptomic analysis. *Nat Commun* 12, 5291.

REVIEWERS' COMMENTS

Reviewer #2 (Remarks to the Author):

In this revised manuscript, the authors have added a number of key data addressing all my concerns. Thank you for taking the time to provide a scholarly response. Overall, this study makes valuable contributions to our understanding of TP63-IFN-STAT1 axis in anti-cancer immunity, and is appropriate for publication in this journal.

Reviewer #3 (Remarks to the Author):

The authors made good-faith efforts to address the reviewers' concerns, clarifying unclear conclusions and most importantly adding new data that strengthen their conclusions. The new knowledge warrants publication in Nature Communications.

Reviewer #4 (Remarks to the Author):

Reviewer #5 (Remarks to the Author):

The authors have largely addressed my previous concerns. In my view, the manuscript now meets the standards for acceptance by Nature Communications.

Revised NCOMMS-23-28296B by Yuan Jiang *et al.*

Title: Reciprocal inhibition between TP63 and STAT1 regulates anti-tumor immune response through interferon- γ signaling in squamous cancer

Dear Reviewers,

We are truly delighted to note that all Reviewers consider that our revision has comprehensively and adequately addressed all the comments, and support its publication in Nature Communications as is. The valuable comments have helped cemented our conclusions and improve further the quality of our work.

Thank you for your interest in and evaluation of our work.